# Activation of IRF3 in cardiomyocytes impairs mitochondrial oxidative function through PGC-1α inhibition and drives heart failure

Manju Kumari [1,2,3,15] ✉, Ioannis Evangelakos [2], Anushka Deshpande [1,3], Kirstie A. De Jong[4], Nesrin Schmiedel[5], Nicolàs Palacio-Escat [6], Adriano de Britto Chaves-Filho [7], Roberto Carlos Frias-Soler [7], Glynis Klinke [8], Hyun Cheol Roh [9], Luisa Lange[1], Michael Berlin[10], Marceline M. Fuh[2], Jakob Johannes Buss [1,3], Tian Tian[2], Carolin Lerchenmüller [1,3,11], Marc Freichel [10], Almut Schulze [7], Viacheslav O. Nikolaev [4], Thomas Eschenhagen [12], Evan D. Rosen [13,14], Ludger Scheja[2], Ashraf Yusuf Rangrez [1,3], Joerg Heeren [2,15] ✉ & Norbert Frey [1,3,15] ✉

Heightened sterile inflammation and mitochondrial metabolic dysfunction drives the pathophysiology of heart failure in ischemic cardiomyopathy. Yet, the transcriptional regulators within cardiomyocytes driving crosstalk between inflammation and energy metabolism remain ill-defined. Here we identify elevated Ser396/Ser398 phosphorylation of the type I interferon (IFN) response regulating transcription factor IRF3 in the myocardium of patients and male mice with ischemic cardiomyopathy. Cardiomyocyte-specific IRF3 deficiency attenuates ischemia induced contractile dysfunction. Conversely, IRF3 activation in cardiomyocytes through a phosphomimetic IRF3 mutant represses *Ppargc1α* expression leading to dysfunctional mitochondrial oxidative phosphorylation, altered metabolic flux in the pentose phosphate pathway/TCA cycle, impaired NAD metabolism and an excessive type I IFN activation, collectively detrimental for cardiac function. Restoring cardiomyocyte-specific *Ppargc1α* expression in IRF3-overexpressor male mice attenuates contractile dysfunction by augmenting a metabolic shift towards fatty acid oxidation and decreasing inflammatory fibrotic responses. These findings identify IRF3 activation in cardiomyocytes as a transcriptional nexus between cardiac inflammation and metabolic fuel switch contributing to heart failure progression.

Elevated inflammation in cardiac disease is often accompanied by maladaptive cellular metabolism with a concomitant risk for morbidity and mortality. Averting heart failure (HF) in patients surviving acute and chronic ischemic cardiomyopathy remains a challenge despite recent therapeutic advances[1]. Myocardial infarction (MI) is associated with a heightened sterile inflammatory state and impaired mitochondrial oxidative phosphorylation machinery in the affected myocardium[2,3]. At the onset of myocardial ischemia, cardiomyocytes

are deprived of oxygen and nutrient supply with adverse impact on mitochondrial function[4]. To combat cardiac injury, the activated innate immune response provides a short term adaptive response[5]. On the other hand, an exuberant state of inflammation in cardiac tissue injury is created by a sustained or excessive innate inflammatory response along with altered energy metabolism leading to a deleterious process and resulting in HF[6]. Traditionally, transcriptional activation of myeloid cells has been studied as a major driver of innate immune signaling; however, results from targeting the immune response in these cells as a strategy to revert cardiac dysfunction are ambivalent. On the other hand, the potential of cardiomyocyte to regulate inflammation has only recently been recognized[7–9]. Yet, our understanding of transcription factors that regulate innate signaling within cardiomyocytes along with the potential to alter mitochondrial oxidative capacity and cardiac metabolic communication remains elusive.

Over the past decades, nuclear factor-κB (NFκB) has been identified as a major transcription factor that regulates ischemic cardiomyopathy and possesses diverse and dichotomous actions in the heart[10,11]. The role of NFκB has been highly debated and described as both adaptive and maladaptive in cardiac remodeling. During acute hypoxia and reperfusion injury, cardiac-specific expression of IκBα (NFκB inhibitor) increases infarct size with increased apoptosis, indicating a cardioprotective role of NFκB[12]. To the contrary, prolonged activation of NFκB in ischemia is detrimental and promotes HF by executing elevated TNFα, IL-1β and IL6 mediated ER stress and cell death[13]. These studies showed that differential effects of cardioprotective vs cardiotoxic roles of NFκB are dependent on the timing and cellular context of NFκB activation. Furthermore, the outcome of randomized controlled trials targeting chronic inflammation in HF patients yielded mixed results[14,15]. These seemingly conflicting data reflect our limited understanding of the inherent complexity of the transcriptional landscape of innate immunity to modulate inflammation upon cardiac injury.

It is evident from experimental studies that pro-inflammatory signaling in the heart plays an important role in mitochondrial dysfunction while mitochondrial physiology can also serve as the arbiter of cardiomyocyte fate and alter pro-inflammatory response to ischemic injury[16–19]. Within the heart, cardiomyocyte metabolism is intensively active with mitochondria primarily involved in energy production. Major alterations in mitochondrial energy metabolism contribute to the severity of cardiac injury. In mammals, the principal energy source changes from glycolysis in the fetal heart to mitochondrial oxidative phosphorylation in the adult postnatal heart[20]. Peroxisome proliferator-activated receptor γ coactivator-1α (PGC-1α) is involved in the postnatal induction of nuclear genes encoding fatty acid (FA) oxidation enzymes. PGC-1α functions as a master regulator of mitochondrial biogenesis, OXPHOS gene expression and energy production by interacting with NRF1/2 and TFAM in the adult heart[21,22]. Deregulation of cardiac PGC-1α expression has been associated with adverse effect on heart function in mice and humans[23–25]. However, the transcriptional partners of PGC-1α upon activation of innate immune signaling in cardiomyocytes remain undefined.

Recently, three interferon (IFN) stimulated genes (*IFIT2*, *IFIT3* and *IFI44L*), have been identified as biomarkers of ischemic cardiomyopathy in humans[26]. In this regard, interferon-regulatory factors (IRFs), a family of transcription factors (IRF1-IRF9) are major inducers of IFN-signaling, regulating several aspects of innate and adaptive immune responses, including, but not limited to, driving pro-inflammatory signaling in response to toll-like receptors (TLR) 3/4 activation by exogenous virus or pathogens or by endogenous signals (e. g. dsRNA, mtDNA, matricellular proteins, S100A8)[27,28]. Among IRFs, IRF3 is a major pro-inflammatory type I IFN signaling transducer[29]. IRF3 is activated by phosphorylation, which promotes its dimerization, nuclear translocation, and subsequent association with the co-activator CREB-

binding protein (CBP) and binding to canonical IFN response element sequence (IRES) to act as a transcriptional regulator of IFN-α and -β[30,31]. We and others have earlier shown that activation of IFN signaling by IRF3 is associated with metabolic dysfunction in obese and diabetic humans and diet-induced obese mouse models[32,33]. Recently, global IRF3-deficient mice have been shown to exhibit reduced levels of chemokines and inflammatory cell infiltration associated with improved survival after MI[34]. While myeloid cells are well known effectors of immune response, here we unravel the potential of cardiomyocytes in regulating IRF3-driven innate immune signaling and how this alters cardiac immunometabolism towards the pathogenesis of HF. In this study, we identify elevated phospho-IRF3 in the left ventricle of HF patients along with impaired mitochondrial OXPHOS machinery. Our findings in cardiomyocyte-specific IRF3 deficient (CMI3KO) mice show that selective inhibition of IRF3-driven immune response in cardiomyocytes attenuates contractile dysfunction in ischemic cardiomyopathy. We further show that cardiomyocyte-specific activation of IRF3 using IRF3-2D phosphomimetic mutant transgene (CMI3OE) emulates an in vivo model of type I IFN stimulated sterile inflammation that reveals a bidirectional relationship between innate inflammation and mitochondria function. Mechanistically, we identify that pathological activation of IRF3 in cardiomyocytes down-regulates PGC-1α, enforces glycolysis imbalance and impairs mitochondrial oxidative phosphorylation. Finally, we show that AAV-mediated moderate expression of *Ppargc1α* sets a break on the deleterious pro-inflammatory signaling driven by IRF3 activation in the myocardium and elevates FA oxidation attenuating cardiac dysfunction.

## Results

### Activation of IRF3 and downregulation of mitochondrial oxidative phosphorylation in ischemic cardiomyopathy

An excessive innate immune response and mitochondrial abnormality is a characteristic of failing hearts. Yet, the transcriptional control of these processes remains poorly understood. IRF3 is a major transcription factor regulating type I IFN-signaling mediated innate immune response. Phosphorylation of IRF3 (Ser 396 and Ser 398) is essential for IRF3 transcriptional activity and activation of IRF3-dependent pro-inflammatory IFN-signaling. To assess the role of IRF3 in ischemic heart failure, we determined the expression and phosphorylation level of IRF3 in cardiomyopathy due to ischemic injury. First, we found increased phosphorylated IRF3 (Ser 396, Ser 398) levels in the left ventricle tissue of humans with ischemic cardiomyopathy (Fig. 1a, b). IRF3-dependent induced IFN stimulated genes (*IFIT2 and IFIT3*) were recently identified as biomarkers of ischemic cardiomyopathy[26]. We found increased *IRF3* mRNA levels along with significantly increased *IFIT2* and a trend to increased *IFIT3* levels in left ventricle tissue of humans with ischemic cardiomyopathy (Fig. 1c, and Fig. S1a). Furthermore, in order to determine the expression of IRF3 in heart failure patients, we obtained and analyzed single-cell RNA-seq dataset of heart failure and control patients from publicly available Gene Expression Omnibus (GEO) accession no GSE109816 and GSE121893. We applied t-distributed stochastic neighbor embedding (t-SNE) to visualize the different cell type populations in 9,248 single cells isolated from both healthy and heart failure ventricle samples as annotated in the original publication[35] namely, cardiomyocytes (CMs), endothelial cells (ECs), fibroblasts (FBs), macrophages (MPs), and smooth muscle cells (SMCs) (Fig. 1d). We found *IRF3* expression to be upregulated in CMs of heart failure patients compared to the controls (Fig. 1e). In addition, *IRF3* expression was also increased in ECs, FBs and SMCs but was reduced in MPs in these patients (Fig. S1b). Second, we performed ligation of the left anterior descending coronary artery (LAD) in mice to generate a model of myocardial infarction associated with damage induced pro-inflammatory signaling. This resulted in robust induction of phosphorylated IRF3 (Ser 388, Ser 390) levels

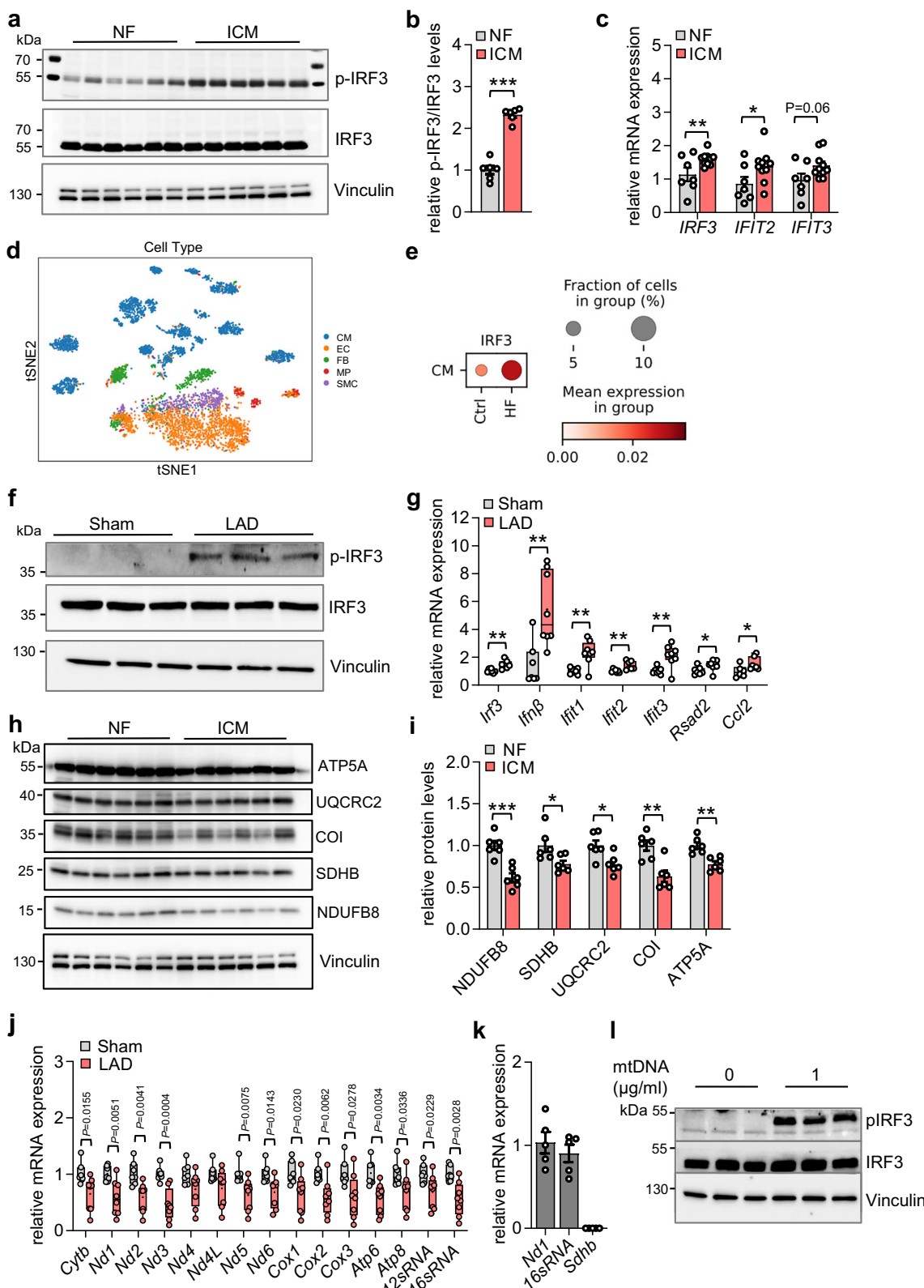

upon occlusion in mice with LAD compared to sham controls (Fig. 1f). Furthermore, immunoblot analysis showed higher levels of phosphorylated IRF3 in the nuclear fraction of cardiomyocytes isolated from mice with myocardial infarction suggesting physiological activation of endogenous IRF3 (Fig. S1c). Consistent with these findings, expression of several *Irf3*-dependent target genes (e.g., *Ifit1, Ifit2, Ifit3*, etc.) was upregulated in left ventricle of mice with myocardial

infarction (Fig. 1g). Of note, we found a marked increase in IRF3 phosphorylation compared to the effect on mRNA expression in mice with ischemic cardiomyopathy. Cardiomyocytes isolated from lipopolysaccharide (LPS) stimulated adult mouse and neonatal rat cardiomyocytes stimulated in presence of polyinosinic:polycytidylic acid (poly I:C) also showed increased expression of *Irf3* and IFN stimulated genes indicating a cell-autonomous key role for type I IFN activation in

**Fig. 1 | IRF3 is activated in ischemic cardiomyopathy in humans and mice.**
**a** Immunoblot showing expression of p-IRF3 and IRF3 levels in the left ventricle tissue of humans with ischemic cardiomyopathy (ICM) compared to non-failing (NF) hearts, NF ($n = 6$) and ICM ($n = 6$), biological replicates. Vinculin was used as a loading control. **b** Quantification of the immunoblot shown in Fig. 1a. Statistical significance was calculated by unpaired two-tailed Student's t test, NF ($n = 6$) and ICM ($n = 6$), ***$P = 5.1 \times 10^{-7}$. **c** mRNA expression of *IRF3* and target genes in the left ventricle tissue of human patients with ICM compared to NF hearts determined by qPCR. Statistical significance was calculated by unpaired two-tailed Student's t test, samples from NF ($n = 7$) and ICM patients ($n = 10$) are biological replicates, *IRF3*: **$P = 0 0.0230$; *IFIT2*: *$P = 0.0534$. **d** UMAP showing single-cell analysis of genes regulated in the left ventricle tissue of patients with heart failure compared to healthy control hearts. **e** Gene expression of *IRF3* in cardiomyocytes isolated from patients with heart failure. **f** Immunoblot showing expression of p-IRF3 and IRF3 levels in the left ventricle tissue of mice with LAD induced myocardial infarction (MI) compared to Sham operated αMHC Cre control. Vinculin was used as a loading control. N = 3 per group, biological replicates. **g** Box plot showing gene expression of *Irf3* and target genes regulating type I IFN signaling in the left ventricle tissue of mice with LAD induced MI determined by qPCR. Sham ($n = 7$) and LAD ($n = 8$), biological replicates. Box and whiskers plot showing all minimum to maximum points. Significance was calculated by unpaired two-tailed Student's t test, *Irf3*:

**$P = 0.0047$; *Ifnβ*: **$P = 0.0041$; *Ifit1*: **$P = 0.0022$; *Ifit2*: **$P = 0.0099$; *Ifit3*: **$P = 0.0046$; *Rsad2*: *$P = 0.0438$; Ccl2: *$P = 0.0311$. **h** Immunoblot showing expression of OXPHOS proteins [NADH: ubiquinone oxidoreductase (NDUFB8), Succinate dehydrogenase (SDHB), Cytochrome c oxidase I (CoI), Ubiquinol Cytochrome c oxidoreductase (UQCRC2), ATP synthase (ATP5A)] in the left ventricle tissue of humans with ICM compared to NF hearts, n = 6 per group, biological replicates. **i** Quantification of the immunoblot shown in Fig. 1h. Statistical significance was calculated by unpaired two-tailed Student's t test, $n = 6$ per group, biological replicates, NDUFB8: ***$P = 0.0005$; SDHB: *$P = 0.0269$; UQCRC2: *$P = 0.0162$; COI: **$P = 0.0024$; ATP5A: **$P = 0.0015$. **j** Gene expression of mitochondrial OXPHOS marker genes by qPCR in the left ventricle tissue of mice with LAD induced MI. Box and whiskers plot showing all minimum to maximum points. Sham ($n = 7$) and LAD ($n = 8$), biological replicates. Significance was calculated by unpaired two-tailed Student's t test and P values are shown in the box plot. **k** *Nd1*, *16sRNA* (mitochondrial) and *Sdhb* (nuclear) levels in mtDNA isolated from rat left ventricle was quantified by qPCR. Data points represent mtDNA samples ($n = 4$), biological replicates. **l** Immunoblot showing pIRF3 levels in primary cardiomyocytes treated with 1 μg/ml of freshly isolated mtDNA for 6 h. N = 3 independent replicates per group. Data in all panels are represented as mean ± SEM. Source data are provided as a Source file.

---

cardiomyocyte inflammatory signaling (Fig. S1d, S1e). In congruous to the activated inflammatory state, a maladaptive response with mitochondrial dysfunction often occurs in patients with HF. The primary function of mitochondria in cardiac cells is to generate energy in the form of ATP through oxidative phosphorylation (OXPHOS) machinery[17]. Impaired OXPHOS can lead to a decline in the bioenergetics capacity of the myocardium. In this regard, the five multimeric protein complexes of the OXPHOS system, namely, NADH: ubiquinone oxido reductase (NDUFB8), Succinate dehydrogenase (SDHB), Cytochrome c oxidase (CoI), Ubiquinol Cytochrome c oxidoreductase (UQCRC2) and ATP synthase (ATP5A) were found to be reduced in the left ventricle of humans with ischemic cardiomyopathy (Fig. 1h, i). Mitochondrial DNA encodes 13 core genes of the OXPHOS using 22 transfer RNAs and 2 ribosomal RNAs. We found severely impaired expression of marker genes of the OXPHOS system (*Cytb, Nd1, Nd2, Nd3, Nd4, Nd4L, Nd5, Nd6, Cox1, Cox2, Cox3, Atp5, Atp6, 12sRNA, 16sRNA*) indicating inefficiency in electron transport chain (ETC) in mice with ischemic cardiomyopathy (Fig. 1j). mtDNA is a widely studied damage-associated molecular pattern (DAMP) that is also known to trigger the inflammatory response during myocardial damage by ischemia[36–38]. We treated primary cardiomyocytes with freshly isolated mtDNA from rat ventricle tissue to determine the impact on IRF3 activation within cardiomyocytes. The purity of mtDNA was validated by *Nd1* and *16sRNA* expression compared to nuclear DNA encoded *Sdhb* (Fig. 1k). We found a robust increase in pIRF3 levels along with elevated type I IFN signaling markers in the presence of mtDNA (Fig. 1l, and Fig. S1f). Together, these results indicate that elevated phosphorylation of IRF3 in ischemic cardiomyopathy is associated with detrimental cardiac mitochondrial function.

### Cardiomyocyte-specific IRF3 deficiency attenuates ischemic cardiomyopathy
To investigate the cardiomyocyte-specific role of IRF3 deficiency in vivo, we utilized mice with an allele in which *Irf3* exon three to six were flanked by loxP sites (Fig. 2a). *Irf3* floxed mice were crossed with αMHC-Cre mice (Jax: 011038)[39], to create cardiomyocyte-specific IRF3-knockouts, hereafter designated as CMI3KO mice. To compare the effects of CMI3KO mice on cardiac physiology, αMHC-Cre littermate mice were used as controls in all experiments. Cardiomyocytes were isolated from 12wk old CMI3KO mice using Langendorff technique and IRF3 deficiency was confirmed at mRNA and protein level (Fig. 2b, c; and Fig. S2a, S2b). To study the effect of IRF3 deficiency on cardiac function upon injury, MI was induced by left anterior descending

artery ligation in 10wk old CMI3KO and αMHC-Cre control mice (Fig. 2d). Contractile function as determined by echocardiography measurement of ejection fraction and fractional shortening was severely impaired in αMHC-Cre mice post MI compared to sham control (Fig. 2e–g, and Fig. S2d). In contrast, CMI3KO mice subjected to MI showed partial yet significantly attenuated cardiac dysfunction despite intact IRF3 levels in other cardiac cell types (Fig. 2e–g, and Fig. S2b). Cardiomyocyte-specific IRF3 deficiency had no significant effect on body weight, heart weight/body weight or lung weight/body weight ratio (Fig. 2h, i; and Fig. S2c) following MI. Furthermore, IRF3 deficiency in cardiomyocytes led to reduced fibrosis in CMI3KO mice compared to the αMHC-Cre control mice as quantified with Masson-Trichrome immunostaining (Fig. 2j, k). To further investigate the cell-autonomous effect of IRF3 deficiency in cardiomyocytes, we performed IRF3 knockdown in cardiomyocytes isolated from neonatal rats (Fig. S2e, S2f). IRF3 knockdown did not alter the expression of other IRFs with the exception of IRF7 which is a known transcriptional target of IRF3[40] (Fig. S2g). IRF3 deficiency in cardiomyocytes treated in the absence or presence of LPS attenuated the cellular inflammatory state by reducing the expression of *Irf3* target genes (*Ifit3, Rsad2, Ccl2, Isg15* etc.) in a cell-autonomous manner (Fig. S2h). These results show that cardiomyocyte-specific IRF3 deficiency attenuates excessive inflammatory signaling and thereby reduces detrimental effects on cardiac function in response to myocardial injury.

### Cardiomyocyte-specific IRF3 activation impairs cardiac function and leads to HF
As we found increased phosphorylated IRF3 in ischemic cardiomyopathy in humans and mice, we hypothesized that aberrant IRF3 activation in cardiomyocytes is sufficient to create local and systemic pro-inflammatory milieu and is detrimental to cardiac function in vivo. To note, in human IRF3, mutations of Ser396 and Ser398 to Asp results in a phosphomimetic constitutively active allele that enables modeling IRF3 activation without a need for external stimuli such as lipopoly-saccharide or poly I:C[30,32,41]. Our previous studies have shown that an analogous double mutant murine IRF3 allele in which Ser388 and Ser390 are mutated to Asp, hereafter designated as IRF3-2D, is constitutively active in vitro[32]. To study the role of IRF3 activation on cardiac function, we developed a knock-in mouse line, in which IRF3-2D allele was flag-tagged and placed within the Rosa26 locus behind a loxP-stop-loxP cassette, allowing Cre-dependent expression of IRF3-2D (Fig. 3a). Floxed IRF3-2D mice were bred with αMHC-MerCreMer mice (αMHCMCM, JAX: 005657)[42] to generate cardiomyocyte-specific IRF3-

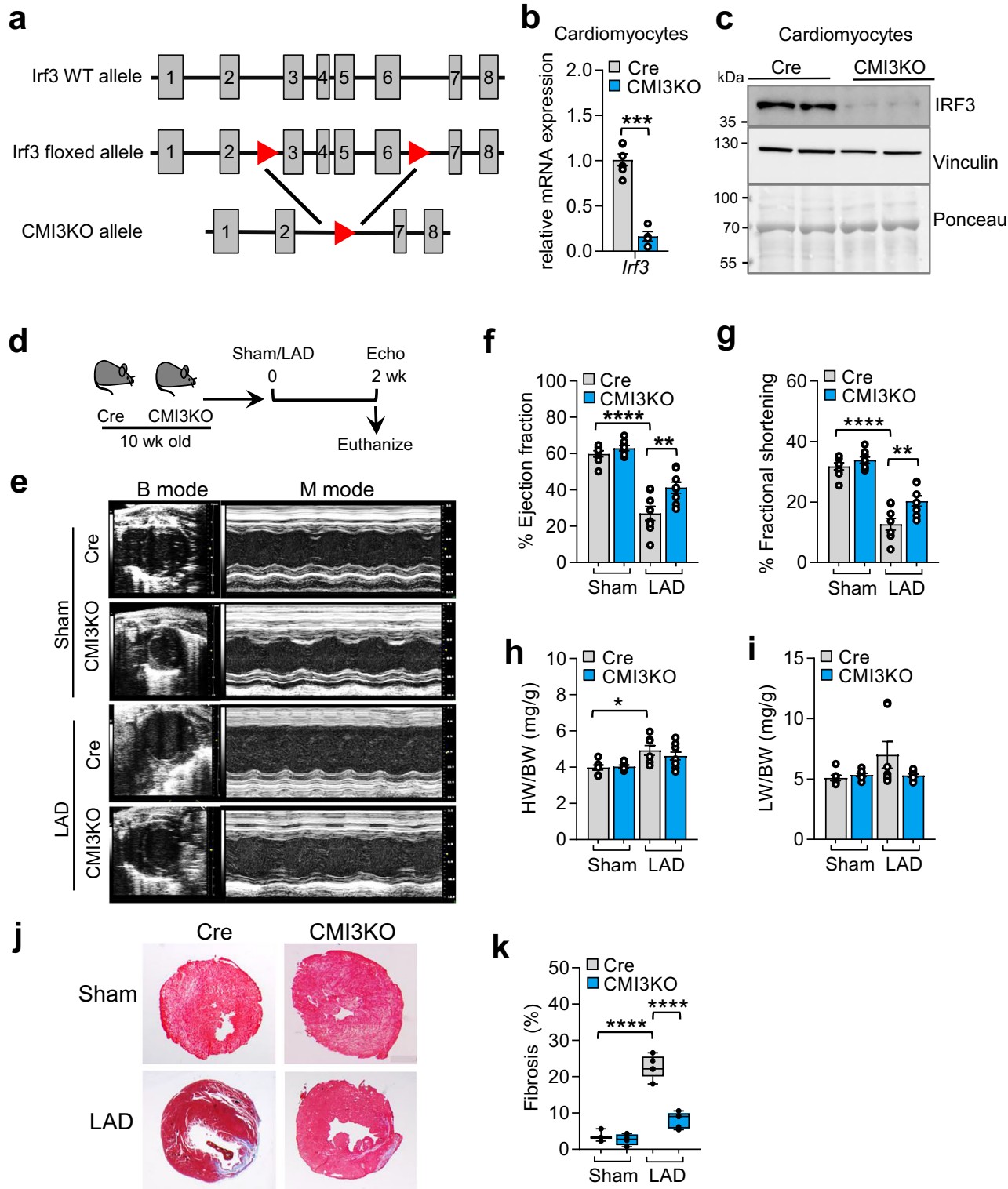

2D expressor mice, hereafter designated as CMI3OE. To minimize Cre expression-induced cardiotoxicity, we used oral gavage of low dose Tamoxifen (30 mg/kg) for 3 days in CMI3OE (Fig. 3b). Littermate αMHCMCM mice were used for comparison. Oral administration of tamoxifen has been earlier reported to be better tolerated in mice and lower doses have been associated with minimal adverse effects on heart function[43]. IRF3 activation in cardiomyocytes led to unexpected mortality following 5th day of IRF3-2D expression in CMI3OE mice (Fig. 3c, d). We found ~3 fold upregulation of IRF3 protein levels in

CMI3OE mice (Fig. 3e, f) compared to tamoxifen-gavaged αMHCMCM controls which is similar to the amount of phosphorylated IRF3 induction seen in the left ventricle of patients with ischemic cardio-myopathy (Fig. 1b). *Irf3* mRNA and interferon target genes *Ifnβ*, *Ifit1*, *Ifit2*, *Ifit3*, *Rsad2*, and *Ccl2* were significantly upregulated (Fig. 3g). Similarly, circulating cytokines levels of CCL2, IL-6, and TNFα was also significantly increased (Fig. 3h–j). CMI3OE mice did not show a sig-nificant change in body weight (Fig. 3k). Echocardiography showed impaired contractile function with reduction in left ventricular ejection

**Fig. 2 | Mice lacking IRF3 in cardiomyocytes are protected from ischemic cardiomyopathy. a** Schematic representation of cardiomyocyte-specific IRF3 deletion using mouse with floxed IRF3 allele crossed to αMHC-Cre (designated as Cre in this Figure). **b** Gene expression of *Irf3* mRNA levels in isolated cardiomyocytes from 12wk old αMHC-Cre and CMI3KO mice. $n = 6$ (αMHC-Cre) and $n = 4$ (CMI3KO), biological replicates. Significance was calculated by unpaired two-tailed Student's t test, ***$P = 1.8 \times 10^{-5}$. **c** Immunoblot showing IRF3 protein levels in isolated cardiomyocytes from 12wk old αMHC-Cre and CMI3KO mice, $n = 2$ per group, biological replicates. Experiment was repeated twice with similar results. **d** Schematic representation of experimental plan to determine cardiac function in CMI3KO mice upon LAD induced MI. αMHC-Cre served as control. **e** Representative images of echocardiography performed in 12wk old αMHC-Cre control and CMI3KO mice. **f-g** Cardiac function in CMI3KO mice compared to αMHC-Cre controls determined by echocardiography, $n = 8$ (αMHC-Cre), $n = 8$ (CMI3KO), biological replicates. Data analyzed by one-way analysis of variance (ANOVA) with Šídák's multiple comparison test, %EF Cre LAD vs Cre Sham: ****$P = < 0.0001$; %EF CMI3KO KO LAD vs Cre LAD: **$P = 0.0051$; %FS Cre LAD vs Cre Sham: ****$P = < 0.0001$; %EF CMI3KO KO LAD vs Cre LAD: **$P = 0.0072$. **h** Heart weight/body weight ratio in αMHC-Cre control and CMI3KO mice. Sham: $n = 7$ (αMHC-Cre), $n = 8$ (CMI3KO); LAD: $n = 7$ (αMHC-Cre), $n = 9$ (CMI3KO), biological replicates, data analyzed by one-way analysis of variance (ANOVA) with Šídák's multiple comparison test, *$P = 0.0145$. **i** Lung weight/body weight ratio in αMHC-Cre control and CMI3KO mice. Sham: $n = 7$ (αMHC-Cre), $n = 8$ (CMI3KO); LAD: $n = 7$ (αMHC-Cre), $n = 9$ (CMI3KO), biological replicates. **j** Representative images of Masson Trichrome staining in cardiac tissue of αMHC-Cre control and CMI3KO mice. **k** Quantification of Masson Trichrome staining in Fig. 2j and analysis using one-way analysis ANOVA with Šídák's multiple comparison test, ****$P = < 0.0001$. Box and whiskers plot showing all minimum to maximum points. $n = 5$ (Cre control LAD), $n = 5$ (CMI3KO LAD), $n = 3$ (Cre control Sham), $n = 4$ (CMI3KO Sham), biological replicates. Data in all panels are represented as mean ± SEM. Source data are provided as a Source data file.

fraction and fractional shortening (Fig. 3l-n) in CMI3OE mice compared to αMHCMCM control mice. CMI3OE mice also showed left ventricle dysfunction with increased left ventricle thickness and posterior wall thickness measured during diastole (Fig. 3o-q). The ratio of heart weight but not lung weight to tibia length was increased in CMI3OE mice (Fig. 3r, s).

Furthermore, in an additional study we determined the effect of IRF3 activation on cardiac function using Mck-Cre mouse (JAX: 006475), wherein Cre is expressed in skeletal muscle with a moderate expression in the cardiac tissue[44]. Floxed IRF3-2D mice were crossed with Mck-Cre mice to generate muscle-specific IRF3-2D transgenic mice, hereafter designated as MI3OE (Fig. S3a). To our surprise, body weight was significantly reduced in MI3OE mice compared to Mck-Cre control mice and the animals displayed early mortality from 4 weeks postnatal (Fig. S3b–S3d). Muscle-specific IRF3 activation resulted in ~2.5 fold and ~5 fold increased expression of *Irf3* in cardiac tissue and gastrocnemius muscle, respectively (Fig. S3e–S3g). This led to increased expression of IRF3 target genes in skeletal muscle as well as in the cardiac tissue of MI3OE mice (Fig. S3e, S3f). It is worth noting that plasma glucose and insulin levels were reduced whereas circulating CCL2 levels were robustly increased in MI3OE mice (Fig. S3h–S3j). Cardiac function determined by echocardiography in 4wk old MI3OE displayed severely impaired contractility with reduced ejection fraction and fractional shortening along with increased heart weight to body weight ratios (Fig. S3k–S3p). Taken together, these results show that aberrant activation of IRF3 in cardiomyocytes is a robust in vivo model recapitulating the potential of type I IFN signaling mediated sterile inflammation in promoting cardiac contractile dysfunction.

## Impaired PGC-1α activity and oxidative phosphorylation upon cardiomyocyte-specific IRF3 activation

We have earlier shown a role for the transcription factor IRF3 in regulating glucose and energy homeostasis in fat and liver tissues[32,33,41]. Little is known about how inflammatory signaling by activated IRF3 controls the transcriptional network in the heart regulating contractile and metabolic functions. Intrigued by the severe contractile dysfunction observed in CMI3OE mice, we performed bulk RNA-seq analysis to understand the impact of IRF3 activation on the cardiac transcriptome. We identified four major cluster of genes dysregulated in CMI3OE cardiac tissue upon IRF3 activation compared to αMHCMCM controls (Fig. 4a). Interestingly, we found the gene expression of oxidation-reduction processes severely downregulated with increased expression of pro-inflammatory and fibrotic markers (Fig. 4b, and Fig. S4a–S4d). Notably, expression of genes regulating chemokine signaling and extracellular matrix organization (*Ccl2, Ccl5, Col12a1, Col1a1, Col3a1, Itgax, Tgfb2* etc.) were increased suggesting metabolic and structural remodeling along with elevated inflammatory state in CMI3OE mice (Fig. 4a, b). Despite significantly elevated expression of

fibrotic markers, structural changes indicative of fibrosis were not observed suggesting an early phase of cellular adaptation in CMI3OE mice (Fig. S4e, S4f). Next, we determined the effect on OXPHOS proteins in the heart that generates oxidative energy and empowers the cellular energy signaling. We found that protein levels of NDUFB8, SDHB, CoI, and UQCRC2 were significantly reduced indicating severely impaired electron transport chain via reduced OXPHOS proteins in CMI3OE mice (Fig. 4c, d). Additionally, in the heart, the creatine kinase phosphotransfer circuit functions as an energy shuttle between mitochondria and cytosol[45]. Creatine kinase regulates cardiac muscle contractility which in turn is coupled to mitochondrial ETC (Fig. 4e). CMI3OE mice displayed increased myocardial cellular injury with increased creatine kinase (CK-MB) levels (Fig. 4f).

A profound decrease in mitochondrial aerobic respiration processes in CMI3OE mice was identified in Gene Ontology pathway analysis (Fig. S4d). The preponderance of genes regulating mitochondrial function in the transcriptomics analysis prompted us to assess the effect on mitochondrial DNA content and mitochondrial biogenesis markers upon IRF3 activation in CMI3OE mice. To assess an effect on mitochondrial DNA content, mitochondrial DNA copy number was quantified by determining the mitochondrial DNA/nuclear DNA ratio. Similar ratio suggests that mitochondrial function and not the quantity is altered in the left ventricle of CMI3OE mice (Fig. 4g). We identified robust downregulation of *Ppargc1α* at mRNA and protein level in the cardiac tissue of CMI3OE mice (Fig. 4h, and Fig. 5i).

PGC-1α is a transcriptional coactivator and master regulator of mitochondrial bioenergetics[46]. However, the transcriptional partners of PGC-1α modulating cardiac function are not completely known. In this regard, we utilized multiple approaches to understand the interaction between IRF3 and PGC-1α and transcriptional activity in cardiomyocytes. First, we performed co-immunoprecipitation assays in isolated cardiomyocytes and found that IRF3-2D and pIRF3 interacts with PGC-1α (Figs. 4i, j and Fig. S5a). Also, cell fractionation analysis in cardiomyocytes isolated from CMI3OE showed nuclear abundance of IRF3-2D in CMI3OE mice compared to Cre controls (Fig. S5b, S5c). Second, expression of IRF3-2D in isolated cardiomyocytes showed enhanced nuclear abundance (Fig. S5d–S5f) and led to reduced expression of *Ppargc1α* (Fig. 5a, and Fig. S5g). Concordant with these findings, expression of type I IFN stimulated genes (*Ifit3, Isg15, Oasll, and Rsad2*) was increased whereas mitochondrial marker genes expression was impaired in cardiomyocytes upon activation of IRF3 (Fig. S5h, S5i). Similarly, upregulation of *Ppargc1α* in cardiomyocytes downregulated *Irf3* mRNA levels (Fig. 5b). Contrary to the effect of IRF3 activation, *Ppargc1α* expression attenuated type I IFN stimulated genes expression and increased the mitochondrial marker genes expression in cardiomyocytes under basal as well as inflamed (LPS) or stressed (hypoxia) conditions (Fig. 5c, d; and Fig. S6a–S6d). Third, our results show that siRNA mediated knockdown of *Irf3* in cardiomyocytes increased *Ppargc1α* mRNA levels (Fig. 5e). Fourth, we utilized reporter

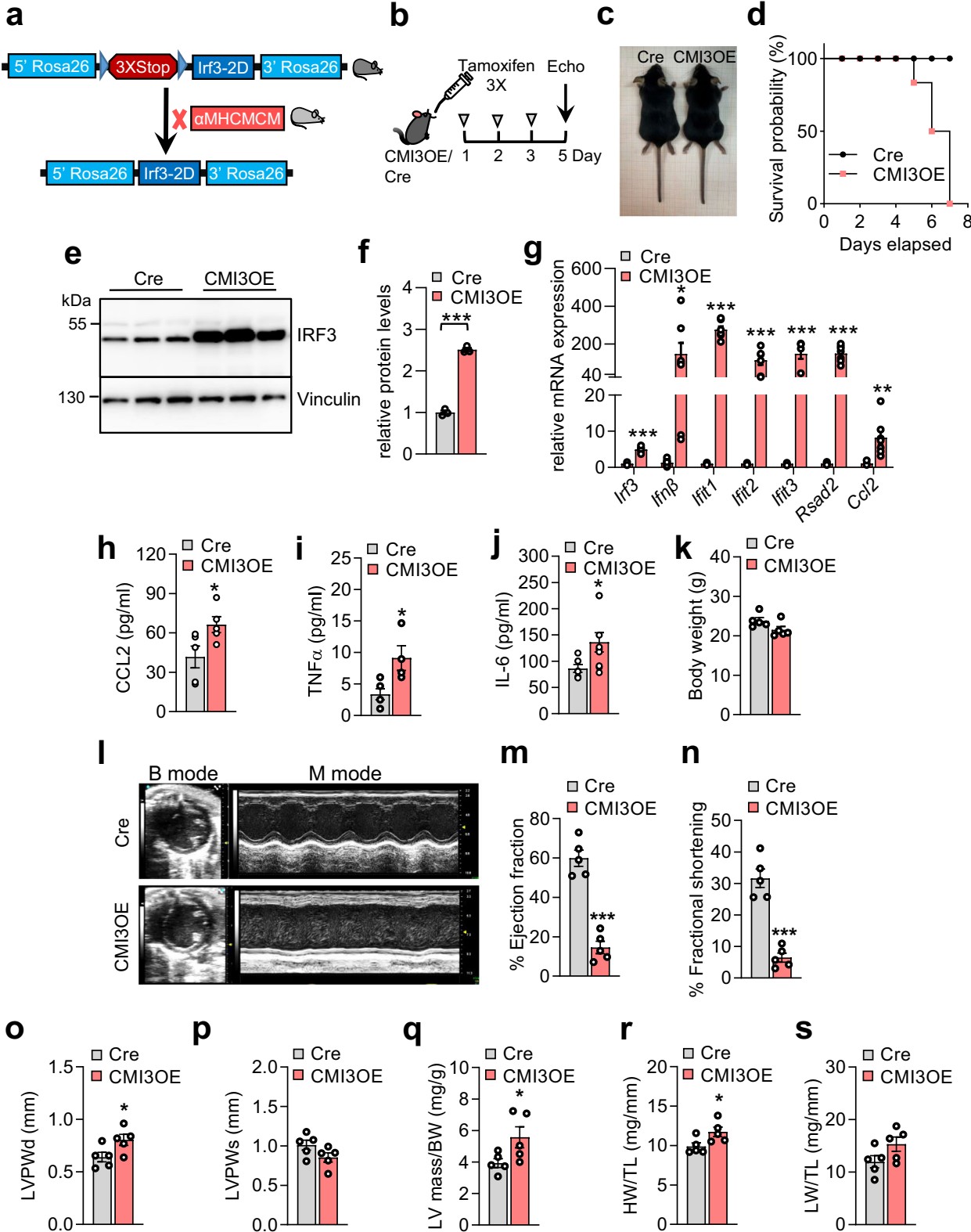

based assays to identify the impact of modulating IRF3 levels on PGC-1α transcriptional activity. To examine the effect of IRF3 activation on PGC-1α activity, we utilized a target reporter plasmid containing a PPARα response element (PPRE) derived from the peroxisomal acyl-CoA oxidase gene promoter multimerized upstream of the thymidine kinase minimal promoter [(PPRE)3TKLuc]. *Ppargc1α* and *Irf3-2D* were ectopically overexpressed alone or in combination in neonatal rat

cardiomyocytes (NRCMs). While *Ppargc1α* expression increased *PPARα* mediated activation of the reporter by 2-fold, expression of *Irf3-2D* inhibited this transcriptional activity of *Ppargc1α* in cardiomyocytes (Fig. 5f). On the other hand, siRNA mediated knockdown of IRF3 enhanced PGC-1α activity (Fig. 5g). In line with these findings, PPARα target genes (*Cd36, Acox1, Acadm, Acadvl,* and *Cpt1b*) were downregulated in CMI3OE mice (Fig. S6e). Fifth, to get a deeper insight into

**Fig. 3 | IRF3 activation in cardiomyocytes is sufficient to promote inflammation and leads to heart failure. a** Representative model showing strategy of CMI3OE transgenic mouse generation. *Irf3-2D* fl/fl mice were bred with αMHCMerCreMer (αMHCMCM) mice to generate tamoxifen inducible cardiomyocyte-specific CMI3OE (*Irf3-2D* fl/αMHCMCM) mice. **b** Schematic representation of cardiomyocyte-specific expression of murine IRF3-2D by oral gavage of low dose tamoxifen (30 mg/kg) to induce MerCreMer-mediated excision of cardiomyocyte-specific floxed cassette in 12wk old male mice. **c** Gross morphology of αMHCMCM (Cre) and CMI3OE male mice at 12wks of age. **d** Kaplan-Meier curve showing survival of CMI3OE compared to αMHCMCM (Cre) mice. **e** Western blot showing protein levels of IRF3 in the cardiac ventricular tissue, $n = 3$ per group, biological replicates. **f** Quantification of the Western blot shown in Fig. 3e, $n = 3$ per group, biological replicates. Data analyzed by unpaired two-tailed Student's t test ***$P = 2 \times 10^{-5}$. **g** mRNA expression of *Irf3* and its target genes in the cardiac ventricular tissue determined by qPCR, $n = 7$ per group, biological replicates. Data analyzed by unpaired two-tailed Student's t test, *Irf3*: ***$P = 1.8 \times 10^{-8}$; *Ifnβ*: *$P < 0.0280$; *Ifit1*: ***$P < 1.5 \times 10^{-9}$; *Ifit2*: ***$P < 0.0005$; *Ifit3*: ***$P < 0.0002$; *Rsad2*: ***$P < 9.9 \times 10^{-7}$; *Ccl2*: **$P < 0.0030$. **h-j** Plasma cytokine levels in CMI3OE mice determined by ELISA. Samples used are biological replicates. Data analyzed by unpaired two-tailed Student's t test, CCL2: *$P = 0.0430$ [$n = 5$ per group]; TNFα: *$P = 0.0229$ [$n = 5$ per group]; IL-6: *$P = 0.0532$ [$n = 6$ (Cre), $n = 9$ (CMI3OE)]. **k** Body weight measured in αMHCMCM (Cre) and CMI3OE mice, $n = 5$ per group. **l** Representative echocardiography images from αMHCMCM (Cre) and CMI3OE mice. **m–q** Cardiac function in αMHCMCM (Cre) and CMI3OE mice determined by echocardiography, n = 5 per group, biological replicates. Data analyzed by unpaired two-tailed Student's t test, %EF: ***$P = 2.3 \times 10^{-5}$; %FS: ***$P = 5.8 \times 10^{-5}$; LVPWd: *$P = 0.0448$; LVmass/BW: *$P = 0.0533$. Cardiac function was measured in three independent cohorts with similar results. **r** Heart weight to tibia length determined in αMHCMCM (Cre) and CMI3OE mice, $n = 5$ per group, data analyzed by unpaired two-tailed Student's t test, *$P = 0.0399$. **s** Lung weight to tibia length determined in αMHCMCM (Cre) and CMI3OE mice, $n = 5$ per group. Survival curve, gene expression analysis, and echocardiography experiments were performed using independent cohorts. Data in all panels are represented as mean ± SEM. Source data are provided as a Source data file.

the IRF3-PGC-1α axis, we analyzed the effect of IRF3 activation on transcriptional targets and post-translational regulators of PGC-1α in CMI3OE. In this regard, mRNA level of *Foxo1*, *Mef2a* and *Esrra* that are known PGC-1α transcriptional targets was reduced in the left ventricle of CMI3OE (Fig. 5h). Because PGC-1α expression can also be regulated by a panoply of posttranslational modifications such as phosphorylation and O-GlcNAcylation, we determined the effect on pAMPKα, p-p38MAPK and pAKT that are among the key kinases known to regulate PGC-1α activity[47]. We found reduced p-p38MAPK along with reduced expression of *Mapk14* (encoding p38MAPK) (Fig. 5i, k) in CMI3OE mice. There was no effect on pAMPKα levels, whereas both total AKT as well as pAKT was reduced in the myocardium of CMI3OE mice. O-GlcNAc transferase (OGT) is the sole enzyme that catalyzes O-GlcNAcylation by transfer of N-acetylglucosamine from UDP-N-acetylglucosamine to serine/threonine residues of proteins. Interestingly, we found reduced levels of OGT and an overall reduced O-GlcNAcylation in CMI3OE mice (Fig. 5i–k) indicating reduced activity of OGT. While increased OGT levels are detrimental for heart function, reduced OGT levels have also been identified to exacerbate heart failure[48,49]. Taken together, these results identify an important cardiomyocyte-specific IRF3/PGC-1α axis acting as a transcriptional control of sterile inflammation wherein activated IRF3 reduces mitochondrial oxidative capacity by downregulating *Ppargc1α* activity.

## Metabolic fuel alteration and impaired NAD metabolism upon cardiomyocyte-specific IRF3 activation

Cardiac function and metabolism are inextricably linked. By virtue of metabolic flexibility, the heart has the ability to adapt to an altered physiological environment. We sought to understand the alteration in the cardiac metabolic state upon IRF3 activation in cardiomyocytes. In the first approach, we performed metabolomics study and identified metabolic adaptations in fatty acid and glucose metabolism and impaired/alternate substrate utilization in the left ventricular tissue of CMI3OE mice (Fig. 6a–j, and Fig. S7a, S7b). Carnitine is essential for normal fatty acid oxidation and carnitine deficiency results in enhanced myocardial injury after ischemia[50–52]. Our results show reduced L-Carnitine, acetyl-Carnitine, and propionyl-Carnitine in the myocardium of CMI3OE mice (Fig. 6a–c). On the other hand, cardiac UDP-Glucose and UDP-Glucosamine levels (Fig. 6d, e) were increased in CMI3OE mice compared to αMHCMCM mice. Similarly, activation of IRF3 led to ~1.5 fold increase in glucose uptake in adult cardiomyocytes isolated from CMI3OE mice (Fig. S7c). Importantly, we found an increase in cardiac 3-hydroxybutyric acid (3OHB) (Fig. 6f) along with increased dimethylglycine levels (Fig. 6g), indicating increased ketogenesis and choline metabolism upon IRF3 activation in CMI3OE mice. Oxidation of 3OHB to acetoacetate by mitochondrial enzyme 3-hydroxybutyrate dehydrogenase 1 (BDH1) is essential for utilization in

the oxidative pathway of the TCA cycle[53–55]. However, reduced expression of *Bdh1* in CMI3OE mice suggests a failure in 3OHB oxidation to improve the energy-starved state in the myocardium (Fig. S8a). Isocitrate dehydrogenase 1 (IDH1) and succinate dehydrogenase (SDHA) play a key role in regulating the citrate and fumarate pool in the TCA cycle[56]. We found differential regulation of the TCA cycle metabolites with increased citric acid and reduced fumaric acid levels in CMI3OE mice (Fig. 6h, i). Similarly, *Idh1* and *Sdha* expression was reduced in the myocardium of CMI3OE mice (Fig. 6o, and Fig. S8b). Notably, fumarate-succinate ratio is a crucial indicator of SDH activity and succinate oxidation in the TCA cycle was found to be reduced in the ventricle of CMI3OE mice (Fig. 6k). Interestingly, our transcriptomics analysis showed a reduced expression of genes involved in the TCA cycle whereas the marker genes for upper glycolysis metabolism were increased (Fig. 6l, m). However, the marker genes for lower glycolytic pathway were downregulated in CMI3OE mice, indicating an imbalanced but not increased glycolysis. Inflammation driven increased ceramide accumulation functions as an immunological signal regulating cell survival along with glucose and lipid utilization[57,58]. In this regard, we found increased levels of ceramide 18:1/22:1 and ceramide 18:1/24:1 upon IRF3 activation in the myocardium of CMI3OE mice (Fig. 6n; and Fig. S7d, S7e). While plasma triglycerides and cholesterol was not altered, we found reduced free fatty acids in CMI3OE mice (Table S1).

In a second approach, we identified further alterations in the circulating lipid profile in CMI3OE mice by plasma lipidomic analysis using the Lipidyzer platform. Here, we found an increase in circulating sphingolipids such as hexosylceramide (HCER) and lactosylceramide (LCER) levels in the CMI3OE mice (Fig. S7f). Sphingolipids have been shown to regulate cellular structural integrity and act as effectors of cell stress response in myocardial pathology[59,60]. These results indicate that IRF3 activation in cardiomyocytes generates immunomodulatory impact with dysregulated ceramide/sphingolipid metabolism in CMI3OE mice.

Interestingly, in addition to the effect on fatty acid and glucose oxidation pathways, transcriptomic analysis of RNA-seq datasets revealed impaired NAD metabolism marker genes upon heightened inflammatory state (Fig. 6o; and Fig. S8c). Thus, in a third approach, we applied NAD metabolomics and assessed the effect on cardiac NAD homeostasis. IRF3 activation in cardiomyocytes of CMI3OE mice resulted in a significantly reduced cardiac NAD+ pool, increased NADH level and reduced NAD + /NADH ratio in comparison to αMHCMCM controls (Fig. 6p–r). Overall, these data suggest that IRF3 activation leads to impaired fatty acid and NAD metabolism in CMI3OE mice.

## IRF3 activation in cardiomyocytes rewires fuel metabolism

Cardiac dysfunction is accompanied by swift metabolic adaptations creating a complex interplay between cellular glycolysis, TCA cycle,

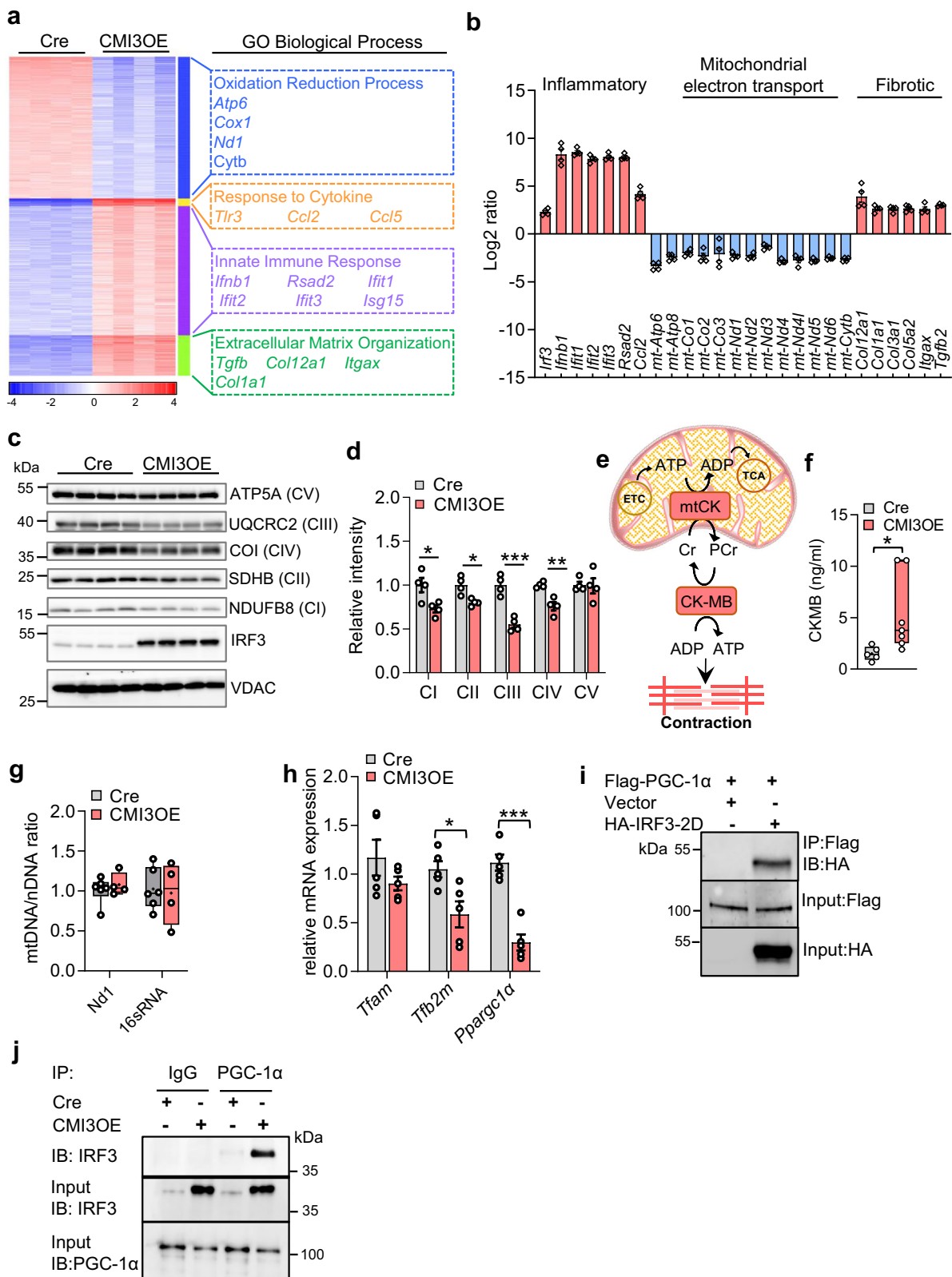

and PPP, wherein the flow of each pathway can be context dependent. To obtain a deeper insight into the effect of IRF3 activation on cardiomyocyte metabolism we performed stable-isotope labeling in cardiomyocytes isolated from CMI3OE and Cre control mice. Isolated cardiomyocytes were treated with 5.5 mM U-$^{13}$C$_6$-Glucose and incubated for 10 min and 120 min to determine the flux in glycolysis, PPP, and TCA cycle (Fig. 7a). While increased glucose uptake, UDP-N-

acetylglucosamine and expression of *Hk1* and *Gpi1* primarily indicated a transient increase in upper glycolysis pathway, there was a modest reduction in isotopologue distribution in glycolytic pathway, especially M + 6 isotopologue of Fructose-1-6-bisphosphate, overall indicating a rather reduced glycolytic flux at 10 min (Fig. 7b-e, Fig. S9a–S9e) in the cardiomyocytes of CMI3OE mice. Strikingly, our results from the isotopologue analysis showed increased flux from

**Fig. 4 | IRF3 activation in cardiomyocytes fails to maintain OXPHOS and PGC-1α levels. a** Heat map showing differential gene expression and GO biological processes altered in 12wk old male CMI3OE and Cre control mice. **b** Log2 ratio in the *y* axis is computed at *P* value less than 0.05 significance threshold from the bulk RNA-seq differential expression gene dataset showing expression of significantly altered inflammatory, mitochondrial and fibrotic marker genes in CMI3OE compared to Cre control mice. *n* = 4 per group, biological replicates. **c** Immunoblot showing cardiac OXPHOS protein levels [NDUFB8 (Complex I), SDHB (Complex II), CoI (Complex IV), UQCRC2 (Complex III), ATP5A (Complex V)] in the CMI3OE mice compared to Cre controls. *n* = 4 per group, biological replicates. **d** Quantification of the immunoblots shown in Fig. 4c, *n* = 4 per group. Data analyzed by unpaired two-tailed Student's t test, CI: *\*P* = 0.0241; CII: *\*\*P* = 0.0102; CIII: *\*\*\*P* = 0.0006; CIV: *\*\*P* = 0.0030. **e** Schematic representation of creatine kinase regulating phospho-creatine/creatine shuttle in cardiac cytosol and mitochondria. **f** Plasma levels of cardiac creatine kinase (CK-MB) levels in CMI3OE compared to Cre control mice determined using ELISA kit. Data analyzed by unpaired two-tailed Student's t test,

*\*P* = 0.0504, *n* = 5 (Cre) *n* = 7 (CMI3OE) per group, biological replicates. Box and whiskers plot showing all minimum to maximum points. **g** Relative quantification of mitochondrial DNA (mtDNA) to nuclear DNA (nDNA) was performed in mtDNA isolated from 12wk old CMI3OE and Cre control mice by amplification of *Nd1* and *16sRNA* belonging to the stable part of the mtDNA and normalized to hexokinase 2 (Hk2) gene. *n* = 6 (Cre), *n* = 4 (CMI3OE). Box and whiskers plot showing all minimum to maximum points. **h** Expression of mitochondrial transcription marker genes in the cardiac ventricle tissue of CMI3OE mice compared to Cre Control. Data analyzed by unpaired two-tailed Student's t test, *Tfb2m:* *\*P* = 0.0186; *Ppargc1a:* *\*\*\*P* = 0.0001, *n* = 5 per group, biological replicates. **i** Expression of Flag-PGC-1α and HA-IRF3-2D was obtained by transfection in neonatal rat cardiomyocytes. The immunoblot shows co-immunoprecipitation of Flag-PGC-1α with HA-IRF3-2D. **j** Cardiomyocytes were isolated from CMI3OE and Cre control mice using Langendorff-free method. The immunoblot shows co-immunoprecipitation of IRF3 with PGC-1α in cardiomyocytes. Data in all panels are represented as mean ± SEM. Source data are provided as a Source data file.

glucose to Sedoheptulose-7-phosphate, a product of the non-oxidative PPP (Fig. 7f). Additionally, our data from this analysis also demonstrated a substantially reduced flux in TCA cycle metabolites in CMI3OE cardiomyocytes (Fig. 7g–j, and Fig. S9f–S9h). These metabolic changes were also captured by the transcriptomic analysis that revealed an altered expression of genes regulating upper and lower glycolysis, TCA cycle, and PPP pathway with similar pattern in the left ventricle CMI3OE mice (Fig. 6l, m; and Fig. S9i). Overall, these data along with the results from the lipidomic analysis acclaim the unique flexibility of cardiomyocytes and suggests that IRF3 activation drives cardiac fuel adaptation towards increased PPP, impaired ketone body metabolism along with reduced oxidative phosphorylation and disrupted NAD metabolism in CMI3OE mice. Thus, our studies highlight a key role of IRF3 activation in reshaping myocardial metabolism during progression to heart failure in mice via switching cardiac fuel preferences (Fig. 7k).

## Cardiomyocyte-specific moderate expression of PGC-1α attenuates contractile dysfunction upon IRF3 activation

Compromised and damaged mitochondria can provoke inflammatory responses[16,61]. However, it remains unclear whether improved mitochondrial function can attenuate the deleterious effects of a type I IFN signaling driven inflammatory assault. Because IRF3 activation in cardiomyocytes diminished expression of PGC-1α and mitochondrial electron transport marker genes, we sought to test the potential of PGC-1α in regulating IRF3 driven type I IFN inflammatory response and mitochondrial oxidative dysfunction. Taking advantage of the recent finding that moderate expression of cardiac-specific *Ppargc1a* in young mice (3 month old) improves cardiac function[62], we evaluated the role of restoring cardiac *Ppargc1a* levels in preserving mitochondrial function, impact on inflammatory signaling and attenuating heart failure in CMI3OE mice. We utilized AAV9-mediated moderate expression of *Ppargc1a* in cardiomyocytes prior to induction of IRF3 expression (Fig. 8a). We injected AAV-PGC-1α (AAV9-TnT-PGC-1α) to achieve cardiomyocyte-specific expression of murine *Ppargc1a*, and AAV-EGFP (AAV9-TnT-EGFP) was used as control. CMI3OE mice with moderate expression of *Ppargc1a* in cardiac tissue were followed by *IRF3* activation using low dose tamoxifen oral gavage (Fig. 8a, b; and Fig. S10a). To note, we found that cardiomyocyte-specific expression of PGC-1α in CMI3OE mice improved cardiac systolic function with increased ejection fraction and fractional shortening compared to CMI3OE mice expressing EGFP (Fig. 8c-e). Furthermore, there was no significant effect of PGC-1α expression on heart weight/body weight ratio in CMI3OE or control mice (Fig. 8f, and Fig. S10b). While the end diastole left ventricle posterior wall thickness was unaltered, the end systole left ventricle posterior wall thickness showed a non-significant trend towards increase in CMI3OE mice as well as control mice expressing PGC-1α (Fig. S10c, S10d). These results unravel a previously

undescribed vital role of cardiomyocyte-specific PGC-1α expression in attenuating cardiac dysfunction caused by IRF3 activation driven sterile inflammation.

## Improved oxidative phosphorylation and metabolic state in CMI3OE mice upon cardiomyocyte-specific PGC-1α expression

We next sought to identify the role of PGC-1α expression in CMI3OE mice and investigated the effects on transcriptional profile of cellular processes and metabolic pathways in the left ventricle using RNA-sequencing datasets. We identified total 11,248 genes significantly differentially regulated, among these, PGC-1α expressing CMI3OE mice modulated 811 genes compared to CMI3OE mice expressing EGFP (Fig. S11a, S11b). Heat map analysis showed four major clusters of biological processes differentially altered (Fig. 9a). Importantly, among these, two clusters were related to downregulated pathways, namely, immune response and response to external stimuli. CMI3OE mice expressing PGC-1α showed significantly reduced pro-inflammatory and fibrotic marker genes compared to EGFP expressing CMI3OE (Fig. 9b, and Fig. S11c). Among the downregulated inflammatory genes were *Irf3* and several type I IFN stimulating genes such as *Ifnb1, Ifit2, Rsad2*, etc. indicating downregulation of pro-inflammatory signaling upon *Ppargc1α* expression. The two other clusters identified pathways upregulated corresponding to regulation of heart contraction and ion transmembrane transport. Importantly, PGC-1α expression increased mitochondrial marker gene expression in CMI3OE mice (Fig. 9c; and Fig. S11d). Similarly, mitochondrial OXPHOS protein levels (NDUFB8, SDHB, CoI, UQCRC2, ATP5A) were increased in CMI3OE mice expressing PGC-1α (Fig. 9d, e). We also found augmented expression of NAD metabolism marker genes in CMI3OE mice with PGC-1α expression (Fig. 9f; Fig. S11e). To corroborate our findings, we measured NAD+ and NADH levels upon expression of IRF3-2D and PGC-1α in isolated cardiomyocytes. Our data showed that IRF3-2D and PGC-1α expression regulated NAD+ and NADH levels oppositely, wherein IRF3-2D expression resulted in reduced NAD+ and increased NADH levels (Fig. 9g–i). On the other hand, PGC-1α expression increased NAD+ levels and reduced NADH levels. Interestingly, cardiomyocytes co-expressing PGC-1α and IRF3-2D showed an increased NAD + /NADH ratio. Taken together, these data highlight that upregulating mitochondrial electron transport function via cardiomyocyte-specific PGC-1α expression downregulates IRF3-dependent pro-inflammatory pathways and restores NAD + /NADH homeostasis.

Reactome analysis provided further insight into the metabolic state of the myocardium upon cardiomyocyte-specific PGC-1α expression in CMI3OE mice, where pathways regulating cardiac contractile function were upregulated (Fig. 10a). Additionally, we also found pathways regulating fatty acid and lipid metabolism upregulated in PGC-1α expressing CMI3OE mice. Pathway enrichment analysis identified increased expression of genes regulating fatty acid

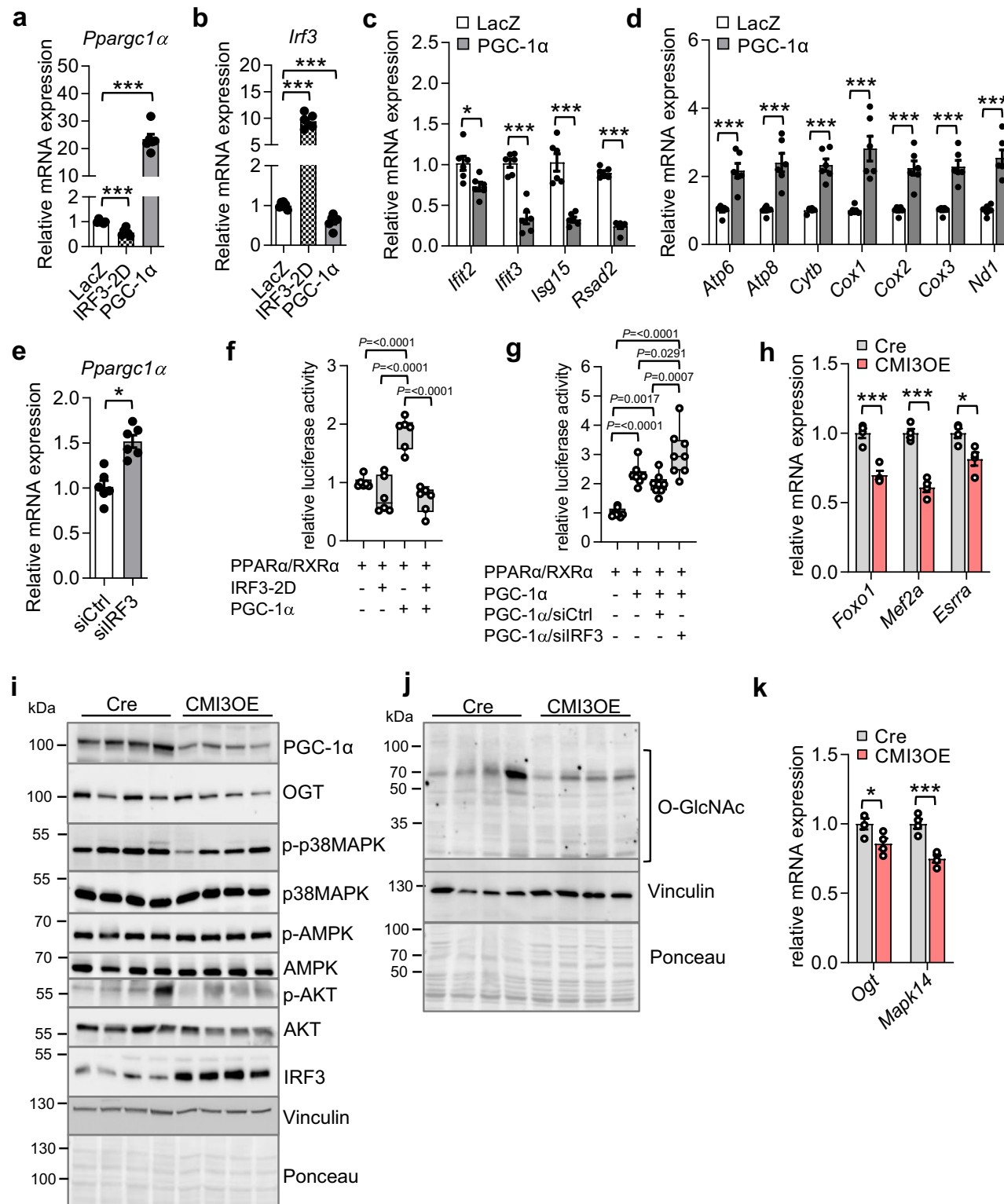

oxidation (*Cpt1b, Cpt2, Acad10, Acad11*, etc.), fatty acid metabolism (*Scd4, Acot, Gpam, lipin, Dgat2*, etc.) and TCA cycle (*Idh3a, Idh3b, Mdh1, Sdhb*, etc.) in the Reactome (Fig. 10b, c). These findings are consistent with the proposed role of PGC-1α in transcriptional control of genes inducing mitochondrial fatty acid oxidation[63,64]. Concordantly, fuel flex assays in isolated cardiomyocytes identified increased fatty acid dependency compared to glucose in the cells expressing PGC-1α and IRF3-2D (Fig. 10d, e). Thus, our studies establish CMI3OE mice as an effective model to study cardiomyocyte-specific sterile inflammation

via downregulation of PGC-1α and mitochondrial OXPHOS function ultimately leading to cardiac dysfunction (Fig. 10f). Moreover, our data indicate that elevated PGC-1α levels in cardiomyocytes may attenuate IRF3 driven contractile dysfunction by preserving the cardiac fuel utilization from FA metabolism.

## Discussion
An exuberant state of the inflammatory response is a hallmark of cardiac ischemic injury followed by altered fuel metabolism and

**Fig. 5 | IRF3 activation in cardiomyocytes reduces PGC-1α activity. a** Effect on *Ppargc1α* mRNA levels was determined by qPCR in neonatal rat cardiomyocytes transduced with murine PGC-1α and murine IRF3-2D. $n = 6$ replicates per group. Data analyzed by unpaired two-tailed Student's t test, IRF3-2D: ***$P = 3.9 \times 10^{-5}$; PGC-1α: ***$P = 2.2 \times 10^{-7}$. **b** Effect on *Irf3* mRNA levels was determined by qPCR in neonatal rat cardiomyocytes transduced with murine PGC-1α and murine IRF3-2D. $n = 6$ replicates per group. Data analyzed by unpaired two-tailed Student's t test, IRF3-2D: ***$P = 2.2 \times 10^{-8}$; PGC-1α: ***$P = 0.0003$. **c** Effect on IRF3 target genes upon *Ppargc1α* expression in neonatal rat cardiomyocytes was determined by qPCR, $n = 6$ replicates per group. Data analyzed by unpaired two-tailed Student's t test, *Ifit2*: *$P = 0.0163$; *Ifit3*: ***$P = 1.9 \times 10^{-5}$; *Isg15*: ***$P = 8.5 \times 10^{-5}$; *Rsad2*: ***$P = 7.3 \times 10^{-9}$. **d** Effect on mitochondrial marker genes upon *Ppargc1α* expression in neonatal rat cardio-myocytes was determined by qPCR. $n = 6$ replicates per group. Data analyzed by unpaired two-tailed Student's t test, *Atp6* ***$P = 0.0004$; *Atp8*: ***$P = 0005$; *Cytb*: ***$P = 2.1 \times 10^{-5}$; *Cox1*: ***$P = 0.0006$; Cox2: ***$P = 0.0002$; *Cox3*: ***$P = 5.1 \times 10^{-5}$; *Nd1*: ***$P = 8.3 \times 10^{-5}$. **e** Effect on *Ppargc1α* mRNA levels upon siRNA mediated *Irf3* knockdown in neonatal rat cardiomyocytes was determined by qPCR. $n = 6$ replicates per group. Data analyzed by unpaired two-tailed Student's t test, *$P = 0.0004$. **f** Luciferase reporter assay was performed in cardiomyocytes isolated from neonatal rats upon transient transfection with PPRE-X3-TK-luc luciferase reporter plasmid. Co-transfection of PPARα, RXRα, Renilla, PGC-1α, and IRF3 expression vector was performed as indicated in the experimental section. Luciferase activity was normalized to *Renilla* (internal control), $n = 6$ replicates per group. Box and whiskers plot showing all minimum to maximum points. Data analyzed by unpaired two-tailed Student's t test, $P$ values are included in the box plot. **g** Luciferase reporter assay was performed in cardiomyocytes isolated from neonatal rats upon transient transfection with siRNA targeting rat IRF3. Co-transfection of PPRE-X3-TK-luc luciferase reporter, PPARα, RXRα, Renilla, and PGC-1α expression vectors was performed as indicated in the experimental section. Luciferase activity was normalized to *Renilla* (internal control), $n = 8$ replicates per group. Box and whiskers plot showing all minimum to maximum points. Data analyzed by unpaired two-tailed Student's t test, $P$ values are included in the box plot. **h** Relative mRNA levels of *Foxo1*, *Mef2a*, and *Esrra* in the left ventricle of CMI3OE mice. $n = 4$ per group, biological replicates. Data analyzed by unpaired two-tailed Student's t test, *Foxo1*: ***$P = 0.0007$, *Mef2a*: ***$P = 0.0001$; Foxo1: *$P = 0.0194$. **i** Immunoblot showing protein levels of PGC-1α, OGT, p-p38MAPK, p38MAPK, pAMPKα, AMPKα, pAKT, and AKT in CMI3OE mice compared to Cre control. $n = 4$ per group, biological replicates. **j** Immunoblot showing O-GlcNAcylation of proteins in the left ventricle of CMI3OE mice compared to Cre control. $n = 4$ per group, biological replicates. **k** Relative mRNA levels of *Ogt* and *Mapk14* in the left ventricle of CMI3OE mice. $n = 4$ per group, biological replicates. Data analyzed by unpaired two-tailed Student's t test, *Ogt*: *$P = 0.0398$, *Mapk14*: ***$P = 0.0011$. Experiments in NRCMs were repeated three times with similar results. Data in all panels are represented as mean ± SEM. Source data are provided as a Source data file.

mitochondrial inefficiency leading towards HF. While most prior studies have focused on myeloid cells as effector of immune response, our understanding of transcriptional control of innate immune activation within cardiomyocytes and its impact on cardiac oxidative metabolism remains unclear. Here we present transcription factor IRF3 phosphomimetic transgene as an effective model of cardiac sterile inflammation to dissect the role of cardiomyocyte-specific innate immune activation and altered cardiac immuno-metabolism during progression to HF. We show a bidirectional transcriptional regulation between IFN signaling response and mitochondrial oxidative phosphorylation altering cardiac contractile function and cellular energy metabolism by modulating IRF3 and PGC-1α levels in cardiomyocytes.

Compelling evidence over the years, both in humans and experimental models of ischemia, have shown that damaged or dying host cells provoke innate immune responses to delimit myocardium injury. IRF3 driven type I IFN signaling is known to regulate innate immunity and has largely been studied in immune cells in the context of viral myocarditis, hypertension, systemic lupus erythematosus, and in response to various pathogens or infections in humans[30]. There has been renewed interest in IFN-related pathways due to their involvement in Severe Acute Respiratory Syndrome Coronavirus 2 (SARS-CoV-2) infection and its cardiac complications. Interestingly, SARS-CoV-2 escapes the IFN response at the early stage of infection by inhibiting phosphorylation and nuclear localization of IRF3 via its nucleocapsid protein, however, at higher doses, i.e., at late stage, SARS-CoV-2 nucleocapsid hyperactivates IRF3 to cause so called "cytokine storm"[65]. IKKε and TANK-binding-kinase 1 (TBK1) are the two major phosphorylation kinases of IRF3[66], whereas IRF3 can be deactivated by dephosphorylation mediated by serine threonine phosphatase PP2A and its adapter protein RACK1[67]. While these studies and vast literature in pathogenic infections epitomizes the diverse effects of immune cells on type I IFN signaling, the potential role of cardiomyocytes in regulating sterile inflammation driven cardiac pathophysiology remains poorly defined. We identified a robust upregulation of phospho-IRF3 (Ser396/Ser398) in the ischemic myocardium of mice and humans. Upon phosphorylation, IRF3 translocate to the nucleus to regulate gene transcription. Our experiments highlight that determining phosphorylated IRF3 levels is critical to assess activation of transcription factor IRF3. Of note, earlier studies have shown that relative IRF3 mRNA levels do not change upon IFN treatment or viral induction and thus do not represent an activated state of this transcription factor[68]. Interestingly, we detected IRF3 phosphorylation in the human

myocardium at basal level confirming its importance in normal cardiac physiology. Recent studies have identified a role for cellular mtDNA content and circulating mtDNA acting as a DAMP in the disease pathology of heart failure patients[36,37,69]. Our data indeed show that mtDNA introduction in primary cardiomyocytes elicits a robust increase in phospho-IRF3 level along with increased type I IFN signaling.

Our finding that cardiomyocyte-specific IRF3 deficiency in CMI3KO mice attenuated contractile dysfunction and fibrosis post MI indicates that IRF3 deficiency in surviving cardiomyocytes confers partial protection against impaired cardiac function by suppressing deleterious inflammatory responses provoked by cellular damage and aberrant oxygen deprivation. Results from these studies highlight a critical role for IRF3 in cardiomyocytes to mitigate inflammatory damage on contractile function upon ischemic trigger. Our findings align with an elegant study from Kevin King lab, showing cardiomyocytes lacking IRF3 led to diminished IFN induced cells at the border zone in MI whereas IRF3 deficiency in fibroblasts, macrophages, endothelial cells or neutrophils did not[70]. The extant literature highlights the role of unresolved inflammation with cytokines (such as TNFα, IL6) and chemokine (CCL2) driven development of cardiac remodeling and progression towards cardiac failure[71]. Despite advances, reverting pathogenic cardiac remodeling upon ischemic injury remains challenging. An alternative strategy is targeting downstream transcriptional regulators of inflammatory signaling and their effectors to delay the pathological structural remodeling processes which may turn out to be of therapeutic potential.

Mitochondrial oxidative phosphorylation is the major contributor towards high energy phosphates required for cardiac contraction in a healthy heart[72]. Activation of IRF3 in CMI3OE highlights the potential of cardiomyocyte-selective type I IFN signaling in driving OXPHOS dysfunction by reducing transcription of genes involved in ETC. In addition to the transcriptional alterations, sustained pro-inflammatory stimulus upon IRF3 activation showed profound effect on cellular metabolic reserve and fuel transition in our study. FAs and carbohydrates metabolism account for 90-95% production of reduced intermediates of mitochondrial electron transport and ATP generation in a healthy heart. The inherent dynamic nature of the heart adapts to metabolize a wide range of energy substrates in order to drive OXPHOS machinery and prevent myocardial insufficiency. This characteristic to adapt towards inconstant energy demands makes the heart a 'metabolic omnivore'.

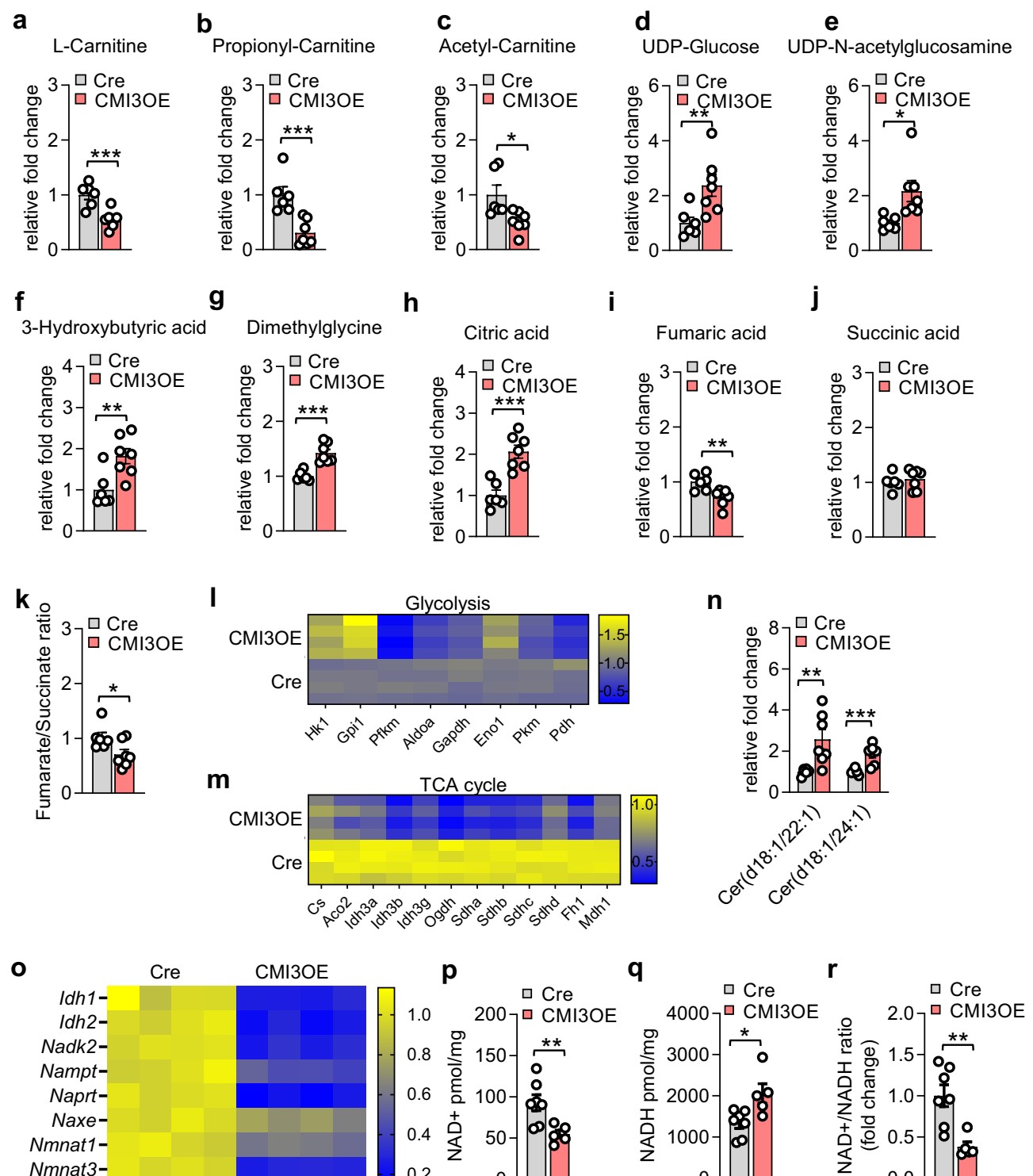

Previous studies have shown major alterations in cardiac energy metabolism contributing towards the severity of cardiac injury that works as a driving force to deteriorate metabolic environment in the myocardium[73]. However, cardiac metabolic changes upon type I IFN signaling activation by IRF3 in the heart have not been investigated so far. IRF3 mediated inflammation led to increased metabolic flux into PPP and reduced flux into TCA cycle highlighting the unique metabolic flexibility of cardiomyocytes towards fuel adaptation under stress conditions. Along with this, a wide spectrum of metabolic changes comprised of reduced FA oxidation and impaired ketone bodies oxidation represents a hallmark of metabolic fuel rewiring by

IRF3 activation in the cardiomyocytes of CMI3OE mice. Carnitine, an amino acid derivative, is essential for transfer of FAs across inner mitochondrial membrane and subsequent β-oxidation which is one of the major pathways fueling TCA cycle in healthy heart[74]. Additionally, ketone bodies have been suggested as "superfuel" oxidized by the failing heart of both humans and rodents in preference to FA and glucose[75,76]. Of note, there is lack of consensus as to whether these metabolic changes are adaptive or non-adaptive in ischemic failing heart. Perhaps, these altered metabolites in CMI3OE mice function as adaptive in the early stage and turn maladaptive when not metabolized efficiently in the late failing heart.

**Fig. 6 | Altered metabolic state and NAD metabolism upon IRF3 activation in cardiomyocytes. a**–**j** Relative fold change of significantly altered metabolites analyzed in the left ventricle tissue of CMI3OE mice compared to αMHCMCM controls determined by LC-MS. $n$ = 6 (Cre) and $n$ = 7 (CMI3OE), biological replicates. Data analyzed by unpaired two-tailed Student's t test, L-Carnitine: \*\*\**P* = 0.0010; Propionyl-Carnitine: \*\*\**P* = 0.0013; Acetyl-Carnitine: \**P* = 0.0202; UDP-Glucose: \*\**P* = 0.0139; UDP-N-acetylglucosamine: \**P* = 0.0201; 3-Hydroxybutyric acid: \*\**P* = 0.0082; Dimethylglycine: \*\*\**P* = 0.0003; Citric acid: \*\*\**P* = 0.0003; Fumaric acid: \*\**P* = 0.0067. **k** Fumarate-succinate ratio in the left ventricle of CMI3OE mice. n = 6 (Cre) and $n$ = 7 (CMI3OE), biological replicates. Data analyzed by unpaired two-tailed Student's t test, \**P* = 0.0344. **l** Heatmap showing differential expression of genes regulating glycolysis pathway in the left ventricle of CMI3OE mice.

**m** Heatmap showing differential expression of genes regulating TCA cycle in the left ventricle of CMI3OE mice. **n** Relative fold change of ceramide species altered in the left ventricle of CMI3OE mice determined by LC-MS. $n$ = 6 (Cre) and $n$ = 7 (CMI3OE), biological replicates. Data analyzed by unpaired two-tailed Student's t test, Cer(d18:1/22:1): \*\**P* = 0.0113; Cer(d18:1/24:1): \*\*\**P* < 0.0013. **o** Heatmap showing differential expression of genes regulating NAD metabolism in the left ventricle of CMI3OE mice. **p**–**r** NAD + , NADH levels in the ice-cold PBS perfused left ventricle of CMI3OE mice was analyzed by NAD metabolomics in an independent cohort. Statistical significance was assessed by unpaired two tailed Student's t test, Cre ($n$ = 7) and CMI3OE ($n$ = 5), biological replicates, NAD + : \*\**P* = 0.0120, NADH: \**P* = 0.0149, NAD + /NADH: \*\**P* = 0.0039. Data in all panels are represented as mean ± SEM. Source data are provided as a Source data file.

PGC-1α is a master regulator of mitochondrial bioenergetics and is a major transcriptional coactivator expressed in cells containing abundant mitochondria and active oxidative metabolism such as cardiomyocytes and brown adipocytes[77]. PGC-1α lacks a DNA binding domain and elucidate nuclear receptor signaling effects by docking to transcription factors[47,78]. Despite a well-known role of PGC-1α upon stress conditions such as fasting, cold and exercise, transcriptional regulators of PGC-1α in an inflamed heart remain poorly defined. Our discovery of IRF3 as a transcriptional regulator of PGC-1α in cardiomyocytes provides a plausible mechanism for the type I IFN driven mitochondrial OXPHOS dysfunction via downregulation of PGC-1α. Within cardiomyocytes, phosphorylated IRF3 and PGC-1α levels exist in inverse correlation. In a rescue experiment, restoring PGC-1α levels partially rescued contractile dysfunction in CMI3OE mice. Most likely, PGC-1α role in elevating mitochondrial OXPHOS and metabolic switch to FA oxidation improved the cardiac function. Overall, our findings markedly enhance our understanding via a) identifying the potential of cardiomyocyte in mediating pro-inflammatory effect of IRF3 on mitochondrial OXPHOS machinery, b) identifying IRF3 as a negative transcriptional regulator of PGC-1α levels in inflamed cardiomyocytes, c) identifying bidirectional role of IRF3-PGC-1α axis in altering cardiac inflammation and oxidative metabolism, and d) demonstrating the potential of restoring PGC-1α levels as mode of partial rescue of a failing heart from IRF3 mediated IFN activation and reverting cardiac metabolic state.

Numerous independent studies have shown a role for PGC-1α in cardiac dysfunction using gain- and -loss of function models. Reduced *Ppargc1α* expression has been reported in animal model of ischemia following coronary artery ligation[79,80] and PPAR-γ agonist Pioglitazone (TZD class of drug) has been shown to activate PGC-1α signaling and attenuate myocardial ischemia reperfusion injury[81–84]. Activating PGC-1α can have potential side effects beyond mitochondrial functions due to coactivating many targets[85]. Results from PGC-1α KO and PGC-1α OE mice suggests the importance of PGC-1α dose and context dependence on the outcome. In this regard, homeostasis in mitochondrial and cardiac function can be attained by fine tuning PGC-1α levels during aging. Furthermore, a moderate increase in PGC-1α expression reported in endurance exercise studies have beneficial effect on heart contractile function[86–88]. In our study the improved cardiac function in CMI3OE with PGC-1α intervention recapitulated an animal model where disrupted metabolic processes in mitochondria are partially restored.

Averting HF in patients with MI is a challenge wherein up to 1 in 4 patients succumb to HF within one year[1,89]. Evidence from preclinical and clinical studies indicate that mitochondrial bioenergetics insufficiency in HF and improving mitochondrial oxidative function may improve or restore cardiac function, respectively[17,90]. Furthermore, NAD+ is an essential cofactor involved in several redox reactions related to ETC. While reduced NAD+ and PGC-1α levels in the liver, skeletal muscle and white adipose tissue have been reported in diet induced obesity[91–93], a role for type I IFN signaling in NAD+ metabolism remains poorly defined. Our studies show that IRF3 activation impaired NAD+ metabolism, whereas PGC-1α expression improved cellular homeostasis in maintaining NAD+ levels, shifted the fuel dependency to FA and reduced IRF3 driven inflammatory signaling in the myocardium. This suggests that PGC-1α mediated increased FA oxidation and NAD + /NADH pool may attenuate the deleterious effects of IRF3 mediated pro-inflammatory milieu in the affected myocardium.

Limited studies including ours highlight the importance of inflammation and cardiac energy deficit working in parallel. However, the *kinetics* of inflammation and metabolic changes may not be parallel in each context[94] and with lack of this knowledge an attempt to prioritize one or the other pathway may lead to a causality dilemma. In summary, the IRF3/PGC-1α axis represents key transcriptional regulators coordinating inflammatory and metabolic circuits modulating mitochondrial oxidative phosphorylation in cardiomyocytes.

## Limitations
Our study has certain limitations. We have determined cardiac contractile function in CMI3OE and CMI3KO male mice. Young female mice have higher survival than male mice without ruptures or HF upon myocardial infarction, and thus, the results in female mice may differ[95,96]. Similarly, gender-specific effects may differently modulate cardiac NAD+ levels which are essential for energy metabolism and DNA repair regulating cellular metabolic state. PGC-1α is an essential coordinator of many vital cellular events, including mitochondrial oxidative function. Due to high expression of PGC-1α in cardiomyocytes, skeletal muscle and hepatocytes, most of the research has focused on its potential to modulate mitochondrial oxidative phosphorylation and mitochondrial biogenesis. Multiple factors regulate PGC-1α level and it is beyond the scope of this manuscript to determine the mechanisms of severely low PGC-1α levels in the failing heart and how IRF3 interacts with PGC-1α. Also, our findings do not rule out the potential role of other transcription factors or cytokines regulating type I IFN signaling in ischemic cardiomyopathy. Importantly, increased chemokine signaling suggests further research required to unveil the potential of IRF3 to modulate autocrine or paracrine processes regulating cardiac function. Importantly, future experiments that pharmacological target interferon signaling via IRF3 may unveil the therapeutic potential in controlling cardiac dysfunction.

## Methods
### Animal Experiments
Animal experiments were approved by the Animal Welfare Officers at University Medical Center Hamburg-Eppendorf, Behörde für Gesundheit und Verbraucherschutz (BGV) Hamburg and Ministerium für Landwirtschaft, ländliche Räume, Europa und Verbraucherschutz (MLLEV) of the state of Schleswig-Holstein, and were carried out in accordance with institutional ethical guidelines. Mice had *ad libitum* access to standard chow diet (Altromin, 1329 P) and water. Mice were housed under a regular 12 h light/12 h dark cycle at constant temperature (23 °C). Strains purchased include: C57BL/6 J (JAX: 000664), αMHCMCM (JAX: 005657), Mck-Cre (JAX: 006475), αMHC-Cre (JAX: 011038).

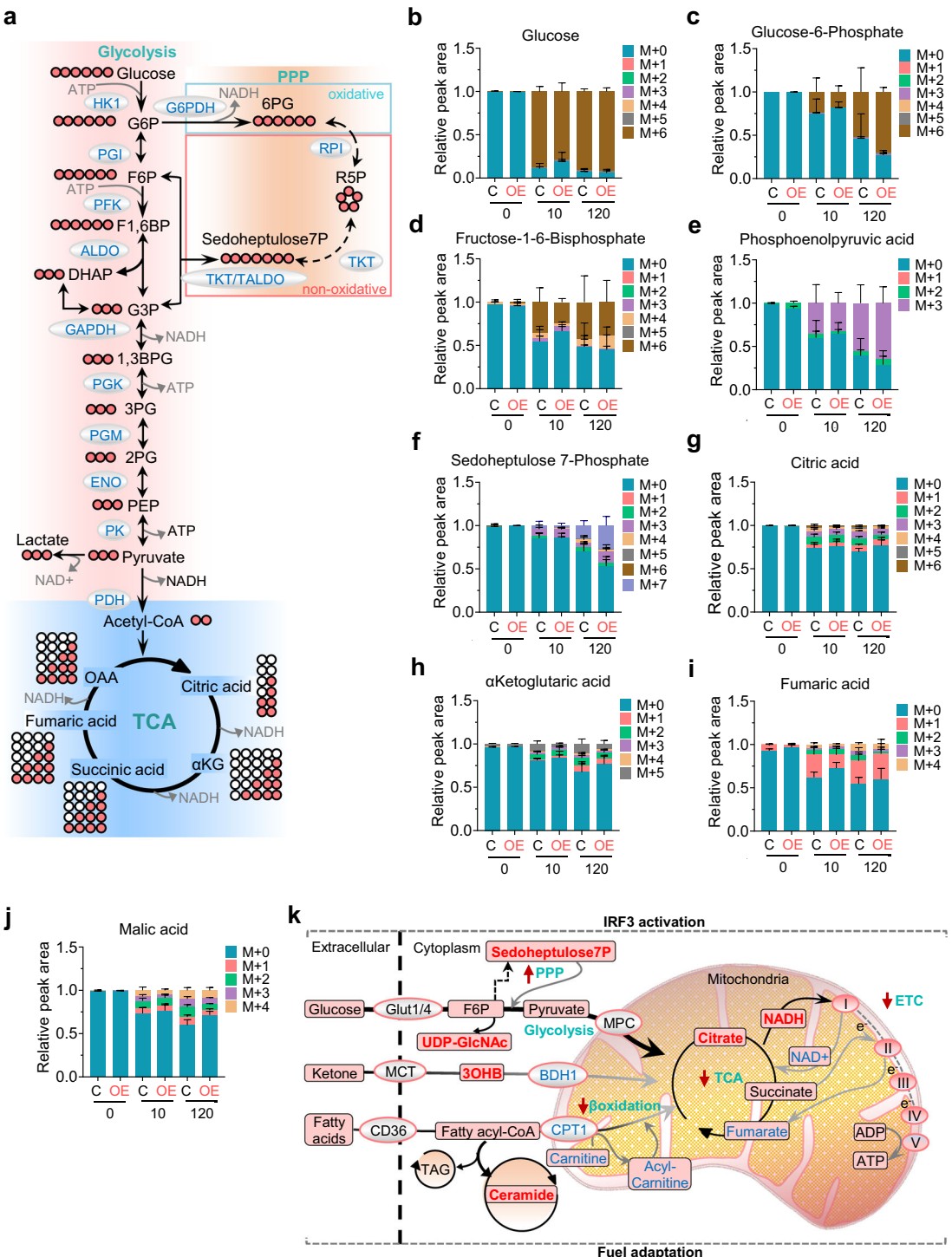

**Fig. 7 | Altered metabolic flux into PPP and TCA upon IRF3 activation in cardiomyocytes. a** Schematic representation of the stable isotope labeling using U-$^{13}C_6$-glucose into glycolysis, PPP and TCA cycle. **b**–**j** Cardiomyocytes isolated from each CMI3OE and Cre control was plated in 6 cm dish at a density of ~1 × 10⁶ cells/dish for 3 h. Cardiomyocytes were washed with PBS and cultured in medium containing 5.5 mM U-$^{13}C_6$-glucose for 10 min and 120 min. Metabolites were extracted and analyzed by LC-MS. Graphs show relative peak area of the iso-topologue from glycolysis, PPP and TCA cycle pathways. Samples group, 0 min: Cre

($n = 4$), CMI3OE ($n = 4$); 10 min: Cre ($n = 5$), CMI3OE ($n = 6$); 120 min: Cre ($n = 5$), CMI3OE ($n = 4$), independent biological replicates. **k** Schematic representation of the fuel adaptation in the left ventricle tissue of CMI3OE mice. Altered metabolite levels indicate an increase in glycolysis and ketone bodies oxidation whereas β-oxidation, carnitine and NAD metabolism is downregulated upon IRF3 activation in CMI3OE mice. Furthermore, this was associated with overall impaired ETC machinery in CMI3OE mice. Data in all panels are represented as mean ± SEM. Source data are provided as a Source data file.

## Generation of IRF3 deficient mice

The IRF3 floxed mice were generated by using a gene targeting vector designed to place LoxP sites flanking exons 3-6 in collaboration with Texas A&M Institute of Genomic Medicine (TIGM). Murine C57BL/6 embryonic stem cells were transfected, and integration of the genomic construct at the predicted site within the Irf3 locus was confirmed by long distance PCR using primers external to the targeted region (3′ homologous arm and 5′ homologous arm).

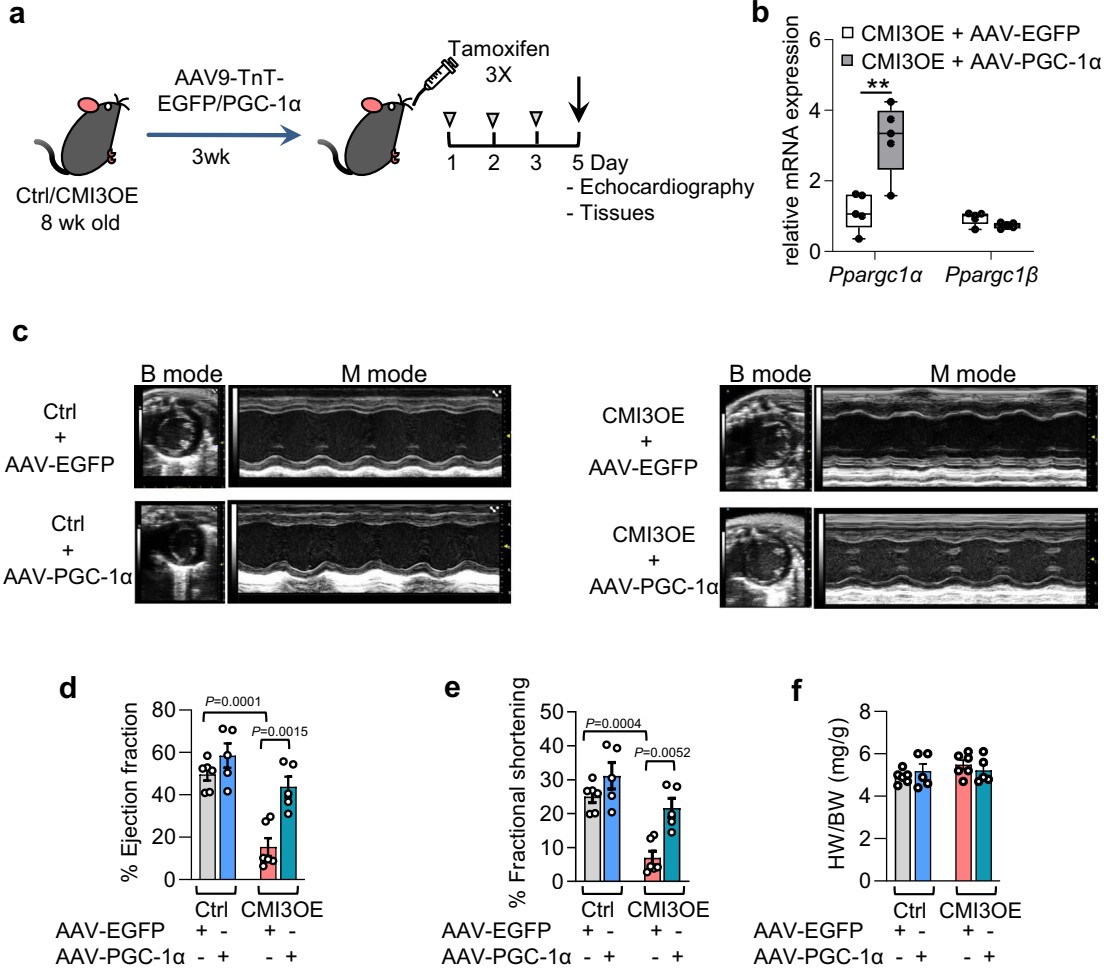

**Fig. 8 | Cardiomyocyte-specific moderate PGC-1α expression attenuates cardiac dysfunction upon IRF3 activation in adult mice. a** Schematic representation of cardiomyocyte-specific expression of PGC-1α and EGFP using AAV9-TnT-PGC-1α and AAV9-TnT-EGFP and IRF3 activation with low dose tamoxifen gavage in CMI3OE mice. **b** Gene expression of *Ppargc1α* and *Ppargc1β* determined in the left ventricle of CMI3OE mice. *n* = 5 per group, biological replicates. Box and whiskers plot showing all minimum to maximum points. Statistical significance was assessed by unpaired two tailed Student's t test, **P = 0.0035. **c** Representative echo-cardiography images from CMI3OE mice treated with AAV9-TnT-PGC-1α and AAV9-TnT-EGFP compared to respective controls. **d**, **e** Cardiac function in CMI3OE mice

treated with AAV9-TnT-PGC-1α and AAV9-TnT-EGFP was determined by echo-cardiography, *n* = 6 (Ctrl-AAV-EGFP), *n* = 5 (Ctrl-AAV-PGC-1α), *n* = 6 (CMI3OE-AAV-EGFP), *n* = 5 (CMI3OE-AAV-PGC-1α), biological replicates. Data analyzed by one-way ANOVA with Šídák's multiple comparison test, *P* values are shown in the graph. **f** Heart weight to body weight ratio determined in CMI3OE mice treated with AAV9-TnT-PGC-1α and AAV9-TnT-EGFP, *n* = 6 (Ctrl-AAV-EGFP), *n* = 5 (Ctrl-AAV-PGC-1α), *n* = 6 (CMI3OE-AAV-EGFP), *n* = 5 (CMI3OE-AAV-PGC-1α), biological replicates. Statistical significance was assessed by unpaired two tailed Student's t test. Data in all panels are represented as mean ± SEM. Source data are provided as a Source data file.

Following ES cell expansion and karyotyping, selected clones were microinjected into C57BL/6 blastocysts and implanted into pseudo-pregnant female mice. Male germ line chimeras carrying an Irf3 floxed allele were backcrossed with wild-type C57BL/6 N females. Heterozygous F1 progeny were crossed to Flpe line (B6.Cg-Tg(ACTFLPe)9205Dym/J) to remove the selection cassette and obtain the floxed allele only. Use of mice was in accordance with protocols approved by the TAMU Institutional Animal Care and Use Committee and following the National Institutes of Health guidelines for laboratory animal use. This mouse line is available from the Jackson Laboratory: *Irf3flox* (JAX: 036260).

### Generation of IRF3 transgenic mice

To generate IRF3-2D mice, the 3XFlag-IRF3-2D transgene was introduced into the ROSA26 locus downstream of a "STOP" cassette consisting of multiple polyadenylation signals flanked by loxP sites. The IRF3-2D coding sequence was amplified using murine 3XFlag-IRF3-2D-pCDH-CMV-MCS-EF1-puro construct[32] with the addition of MluI and

NsiI restriction sites and ligated into the BirA-RanGap-TRAP vector[97]. Primers used for cloning are: forward 5'-GACACGCGTACCATGGATTACAAGGATGACGACGATAA-GATGGAAACCCCGAAACCG-3'

and reverse 5'-GACATGCATTCAGATATTTCCAGTGGCCTG-3'. The vector was linearized by KpnI digestion and the purified product was introduced into C57BL/6 N ES cells. Correctly targeted ES cell clones were identified by long-range PCR screen[98]. The following primers were used to screen the 5' insertion site: 5'-GCCAAGTGGGCAGTT-TACCG-3' (outside of the 5'-arm) and 5'-TAGGTAGGGGATCGG-GACTCT-3' (in the CAG). For the 3' insertion site, the primers used were: 5'-GCCAGCTCATTCCTCCCACTC-3' and 5'-GGCATGG-CAATGTTCAAGCAG-3' (outside of 3'-arm). Chimeric mice were generated by microinjection of ES cells into blastocysts, and germ-line transmission was confirmed by testing the presence of ROSA26 transgene DNA into genomic DNA by PCR using two pairs of the following primers: forward 5'-GCAGCCCAAGCTAGATCGAAT-3', reverse 5'-TTGACACGTCCGGCTTATCC-3' for the transgene and

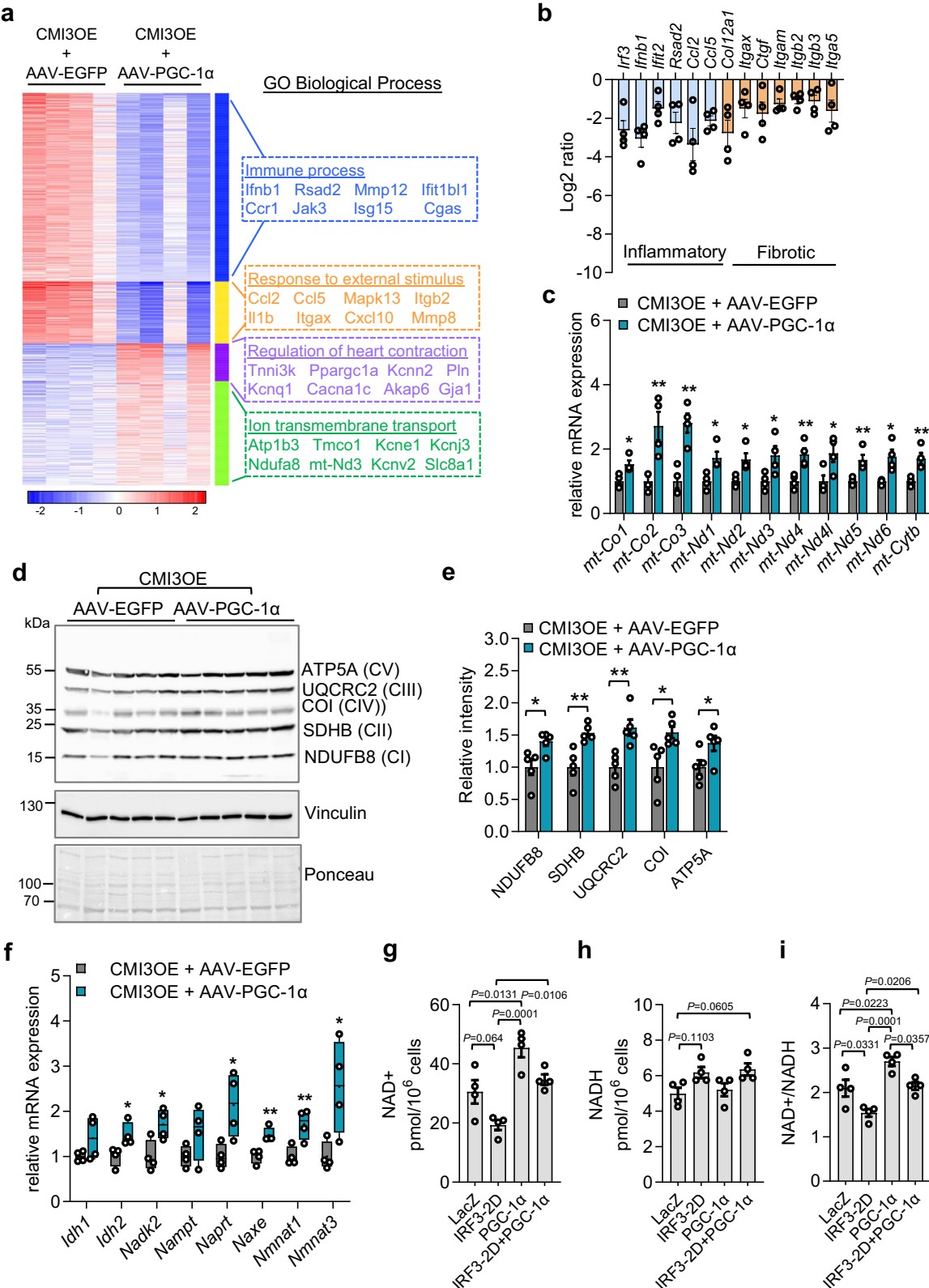

forward 5′-CCTAGCTGTCACCAACCCTTT-3′, reverse 5′-GACGAA-GAGCATCACAAGGAG-3′ for the wild-type. This mouse line is available from the Jackson Laboratory: IRF3-2D (JAX: 036261).

## Echocardiography

Echocardiography was performed either in anesthetized ~12 week old (αMHCMCM, CMI3OE, αMHC-Cre, CMI3KO) with 1.5-2.0% Isofluorane or in awake ~4 week old (Mck-Cre and MI3OE) mice using the Vevo 3100 System (VisualSonics) equipped with a 30 MHz-transducer (MX400 Transducer). Left ventricular structural measurements were obtained in the short-axis view using 2D B-mode and M-mode images including left ventricular posterior wall diameter during diastole and systole (LVPWd, LWPWs). Left ventricular structural measurements were used to obtain systolic indices, ejection fraction (EF) and fractional shortening (FS).

**Fig. 9 | Cardiomyocyte-specific moderate PGC-1α expression attenuates cardiac inflammation and upregulates oxidative metabolism in CMI3OE mice.** **a** Heatmap showing differential gene expression between CMI3OE expressing EGFP or PGC-1α using CMI3OE-AAV9-TnT-EGFP or CMI3OE-AAV9-TnT-PGC-1α, respectively. **b** Log2 ratio in the $y$ axis is computed at $P$ value less than 0.05 significance threshold from the bulk RNA-seq differential expression gene dataset showing gene expression of inflammatory and fibrotic marker genes in CMI3OE-AAV-PGC-1α mice compared to CMI3OE-AAV-EGFP. N = 4 per group, biological replicates. **c** Mitochondrial transcription marker genes expression by qPCR in the cardiac ventricle tissue of CMI3OE-AAV-PGC-1α mice compared to CMI3OE-AAV-EGFP. n = 4 per group, biological replicates. Statistical significance was assessed by unpaired two tailed Student's t test, *mt-Co1*: *$P$ = 0.0160; *mt-Co2*: **$P$ = 0.0100; *mt-Co3*: **$P$ = 0.0027; *mt-Nd1*: *$P$ = 0.0212; *mt-Nd2*: *$P$ = 0.0189; *mt-Nd3*: *$P$ = 0.0455; *mt-Nd4*: **$P$ = 0.0094; *mt-Nd4l*: *$P$ = 0.0372; *mt-Nd5*: *$P$ = 0.0091; *mt-Nd6*: *$P$ = 0.0154; *mt-Cytb*: **$P$ = 0.0095. **d** Immunoblot showing cardiac OXPHOS protein levels of NADH: ubiquinone oxidoreductase (NDUFB8), Succinate dehydrogenase (SDHB), Cytochrome c oxidase I (CoI), Ubiquinol Cytochrome c oxidoreductase (UQCRC2), ATP

Synthase (ATP5A) determined in the CMI3OE-AAV-PGC-1α mice compared to CMI3OE-AAV-EGFP. n = 5 per group, biological replicates. **e** Quantification of the immunoblot shown in Fig. 9d, n = 5 per group, biological replicates. Statistical significance was assessed by unpaired two tailed Student's t test, NDUFB8: *$P$ = 0.0199; SDHB: **$P$ = 0.0040; UQCRC2: **$P$ = 0.0050; COI: *$P$ = 0.0207; ATP5A: *$P$ = 0.0476. **f** Relative expression of genes regulating NAD metabolism in the left ventricle of CMI3OE-AAV-PGC-1α compared to CMI3OE-AAV-EGFP mice by Reactome analysis. N = 4 per group, biological replicates. Box and whiskers plot showing all minimum to maximum points. Significance was assessed by unpaired two tailed Student's t test, *Idh2*: *$P$ = 0.0289; *Nadk2*: *$P$ = 0.0248; *Naprt*: *$P$ = 0.0246; *Naxe*: **$P$ = 0.0047; *Nmnat1*: **$P$ = 0.0106; *Nmnat3*: *$P$ = 0.0293. **g–i** Determination of intracellular NAD + , NADH, NAD + /NADH ratio in neonatal rat cardiomyocytes upon expression of LacZ, IRF3-2D and PGC-1α by adenovirus mediated transduction using NAD + /NADH Quantification kit. $n$ = 4 per replicates group. Data analyzed by one-way ANOVA with Tukey's multiple comparison test, $P$ values are shown in the graph. Data in all panels are represented as mean ± SEM. Source data are provided as a Source data file.

## AAV9-mediated cardiomyocyte-specific PGC-1α overexpression in mice

To attain cardiomyocyte-specific expression of murine PGC-1α in adult mice we obtained ultra-purified AAV9-TnT-mPGC-1α ($1.36 \times 10^{14}$ GC/ml) and AAV9-TnT-EGFP ($5.2 \times 10^{13}$ GC/ml) from Vector Builder Inc, USA. Male C57BL/6 J, 8 week old mice were injected with various doses of AAV9-TnT-mPGC-1α to identify the optimum dose for PGC-1α expression. After screening, $1 \times 10^{12}$ was selected as the appropriate dose to obtain moderate gene expression ( ~ 3 fold) of PGC-1α in the left ventricle tissue of CMI3OE mice compared to EGFP controls.

## Analysis of human cardiac tissue

The human studies were approved by Ethics Committee of the Medical Association Hamburg, File No. 523/116/9.7.1991. Non-failing specimens were obtained between 1991 and 1999, where, in rare cases, incompatibilities prevented transplantation. Patients provided written informed consent. Human left ventricle tissue samples from non-failing and ischemic patients were processed for RNA isolation using RNeasy Universal Mini kit (Cat. No. 73404, Qiagen) combined with genomic DNA elimination following the manufacturer instructions. To obtain protein lysates Kranias buffer (30 mM Tris pH8.8, 5 mM EDTA, 30 mM NaF, 3% SDS, 10% Glycerol, 1 mM DTT) was used. Protease inhibitor cocktail (Roche), phosphatase inhibitor cocktail (Thermo Fisher), 1 mM $Na_3VO_4$ and 1 mM PMSF was added fresh, right before tissue homogenization. For immunoblotting, tissue samples were homogenized in Qiagen Tissue Lyser II and immunoblotted as mentioned below.

## Reagents

Tamoxifen, lipopolysaccharide (LPS), polyinosinic-polycytidylic acid (poly I:C), Insulin, cytochalasin B, glucose, and 2-deoxyglucose were purchased from Sigma-Aldrich. 2-Deoxy-D-glucose, [1,2-3H(N)]- (1 mCi/ml, Cat. MT911) was purchased from Hartmann Analytic.

## Tamoxifen gavage

Body weight was measured before and on third day of Tamoxifen gavage. For inducing gene expression, 12-week old CMI3OE mice were treated with tamoxifen by oral gavage of 30 mg/kg tamoxifen dissolved in corn oil on 3 consecutive days.

## Isolation of adult cardiomyocytes

Adult mouse ventricle cardiomyocytes were isolated by enzymatic digestion via the Langendorff perfusion as previously described[99]. In brief, 12 week old mice were euthanized, hearts were quickly excised and transferred into a chamber filled with ice-cold PBS, aorta was connected to a 21 G cannular. The heart was perfused with the perfusion buffer (NaCl 113 mM, KCl 4.7 mM, $KH_2PO_4$ 0.6 mM,

$Na_2HPO_4.2H_2O$ 0.6 mM, $MgSO_4.7H_2O$ 1.2 mM, $NaHCO_3$ 12 mM, $KHCO_3$ 10 mM, HEPES 10 mM, Taurine 30 mM, 2,3-butanedione-monoxime 10 mM, glucose 5.5 mM, pH 7.4) for 3 min at 37 °C. The heart was digested with 30 ml digestion buffer containing perfusion buffer with liberase DH (0.04 mg/ml, Roche), trypsin (0.025%, Gibco) and $CaCl_2$ (12.5 μM, Sigma) at 37 °C. The atria were excised and discarded, and the ventricle tissue digestion was stopped using 2.5 ml stop buffer I (perfusion buffer containing 1% BSA (Sigma, Germany) and 50 μM $CaCl_2$). The digested heart was cut into small pieces, minced with 1 ml syringe for 3 min and filtered through a 200 μm nylon mesh. Cardiomyocytes were allowed to sediment for 10 min and transferred to stop buffer II (perfusion buffer containing 0.5% BSA and 37.5 μM $CaCl_2$) settled by subjecting to increasing recalcification up to 1 μM Ca2+ concentration. The supernatant was aspirated and the isolated cardiomyocyte pellet was resuspended in 20 ml myocyte culture medium. Isolated adult cardiomyocytes were plated in laminin coated 24 well plates. After 2 h the culture media was changed and cells were used for glucose uptake assay or harvested for protein or RNA isolation.

## Glucose uptake in isolated adult cardiomyocytes

Glucose uptake assay was performed in isolated cardiomyocytes from CMI3OE and αMHCMCM mice as described earlier[32]. Briefly, adult cardiomyocytes were isolated as described above and incubated in serum-free DMEM (11-965-118, Gibco) for 15 min. Cells were treated with 0.5μCi/ml of 2-Deoxy-D-glucose, [1,2-3H(N)]- (1 mCi/ml, Cat. MT911, Hartmann Analytic) for 15 min. Glucose uptake was stopped quickly with media containing 10 μM cytochalasin B (Sigma) on ice. Cardiomyocytes were solubilized in 0.1 N NaOH for 20 min, and radioactivity was measured by scintillation counting. Total protein was measured by BCA protein assay kit (Pierce) and the glucose uptake was normalized to the protein content.

## Isolation of mouse fibroblasts

Fibroblasts from CMI3KO and Cre control were isolated following the protocol from Sahadevan P and Allen BG[100] with slight modifications. Briefly, the heart was perfused with ice-cold perfusion buffer (PBS containing 2% FBS) by inserting a 27 G needle into the left ventricle apex at a rate of ~3 ml/min for 3 min. The heart was excised and rinsed twice with ice-cold perfusion buffer. The atria were removed and the ventricle was minced into small pieces using scissors in 50 μl digestion buffer (DMEM/F12 (1:1) containing 2 mg/ml Collagenase II, 1% FBS, 1% penicillin/streptomycin). The minced heart was transferred into a 15 ml falcon tube containing digestion buffer (2 ml per heart) and incubated at 37 °C for 20 min with gentle rotation and cardiac fibroblasts were isolated as described in the protocol. Isolated cardiac fibroblasts were cultured on collagen pre-coated plates in DMEM/F12 (1:1) containing

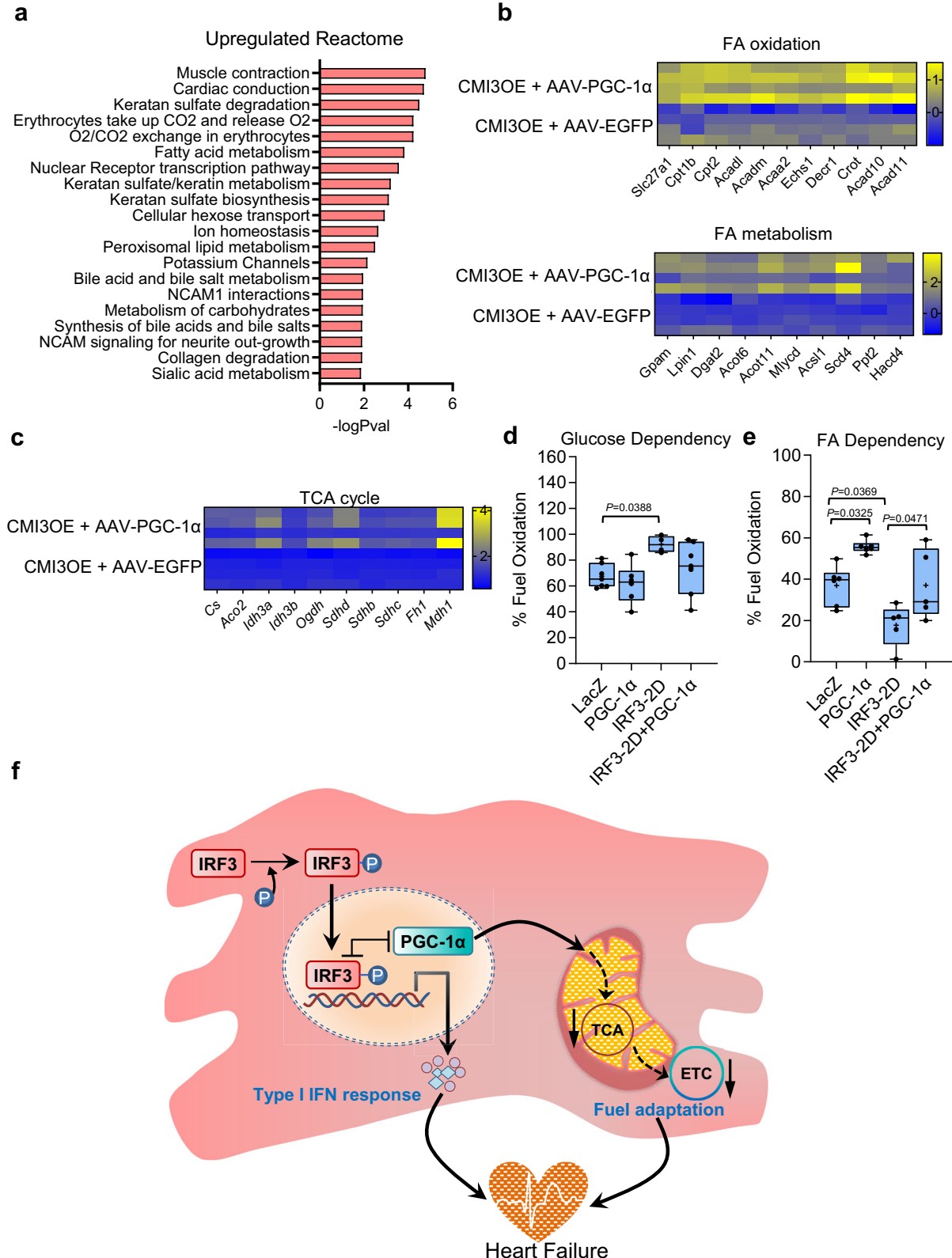

10% FCS and 1% penicillin/streptomycin. After 2 h, cells were washed twice with PBS and fresh media was added. Cardiac fibroblasts at ~70% confluency were harvested for RNA isolation.

### Isolation of mouse endothelial cells

Endothelial cells from CMI3KO and Cre control was isolated following the protocol from Sokoi et al.[101] with slight modifications. Briefly,

transcardial perfusion via the left ventricle was performed with ice-cold PBS at a perfusion rate of ~3 ml/min for 3 min. Heart was harvested, atria were removed and cut into tiny pieces with scissor, pooled (from two mice) and transferred in a 50 ml falcon tube with 5 ml digestion buffer (KnockOut DMEM (ThermoFisher) containing Pencillin/Streptomycin, Antibiotic-Antimycotic, sodium pyruvate, MEM non-essential amino acids, endothelial cell growth supplement, Collagenase II, and DNAse I).

**Fig. 10 | Cardiomyocyte-specific moderate PGC-1α expression upregulates fatty acid metabolism in CMI3OE mice. a** Upregulated pathways in the left ventricle of CMI3OE mice expressing PGC-1α or EGFP (CMI3OE-AAV-PGC-1α vs CMI3OE-AAV-EGFP). **b-c** Heatmap showing relative expression of genes regulating fatty acid oxidation, fatty acid metabolism and TCA cycle identified by Reactome enrichment analysis in the left ventricle of CMI3OE-AAV-PGC-1α compared to CMI3OE-AAV-EGFP mice. **d, e** Fuel flex assay using Seahorse analyzer to determine glucose and fatty acid dependency in NRCMs expressing IRF3-2D and PGC-1α compared to LacZ control. For glucose dependency: LacZ ($n = 7$), PGC-1α ($n = 6$), IRF3-2D ($n = 4$), IRF3-

2D + PGC-1α ($n = 7$). For FA dependency: LacZ ($n = 6$), PGC-1α ($n = 6$), IRF3-2D ($n = 5$), IRF3-2D + PGC-1α ($n = 5$). Box and whiskers plot showing all minimum to maximum points. Data analyzed by one-way ANOVA with Tukey's multiple comparison test, *P* values are shown in the graph. **f** Schematic representation of the effect of IRF3 activation in cardiomyocytes leading to cardiac dysfunction by downregulation of PGC-1α and mitochondrial OXPHOS function. The figure also shows p-IRF3 and PGC-1α levels exist in inverse correlation within cardiomyocytes. Data in all panels are represented as mean ± SEM. Source data are provided as a Source data file.

Samples were digested in 37 °C water bath for 25 min. Tubes were shaken vigorously by hand ever 5-10 min for faster tissue dissociation. The mix was transferred onto a Gentle MACS C tube (Miltenyi Biotech #130-093-237), placed on a gentleMACS™ Octo Dissociator and Protocol C was run. Collected samples were washed and pelleted by sequential steps as described in the protocol. The pellet was resuspended in 90 μl wash buffer (PBS containing 0.5% BSA and 2 mM EDTA) and 10 μl of CD31 Microbeads (Miltenyi Biotech #130-097-418) for $1 \times 10^7$ cells was added and incubated for 15 min at 4 C. Cells were washed with 3 ml wash buffer and centrifuged at 300 g for 5 min at 4 °C. The pellet was resuspended in 0.5 ml wash buffer, the cell suspension was loaded onto the LD column (Miltenyi Biotech #130-042-901) and CD31 positive cells were collected as described in the protocol. The isolated endothelial cells were seeded in 12well plates in culturing media (KnockOut DMEM (ThermoFisher) containing Penicillin/Streptomycin, Antibiotic-Antimycotic, sodium pyruvate, MEM non-essential amino acids, and endothelial cell growth supplement) and the media was changed after 3 h. Endothelial cells were harvested after 24 h for RNA isolation.

### Cloning of IRF3, IRF3-2D mutant or PGC-1α

For expression in primary cardiomyocytes, the PCR product encoding murine PGC-1α, WT murine IRF3 or IRF3-2D mutant (described earlier[32]) or was recombined into the pDonR221 plasmid (Invitrogen) using the Gateway technology and was subsequently shuttled into the p-DEST (Invitrogen) vector to generate HA-tagged WT murine IRF3 or IRF3-2D mutant, Flag-tagged murine PGC-1α following manufacturer's instructions. For adenoviral expression, vector pAd/CMV/V5-DEST (Invitrogen) encoding cDNA for murine PGC-1α, WT murine IRF3 or IRF3-2D mutant was generated as described earlier[32]. Briefly, the PCR product encoding murine PGC-1α, WT murine IRF3 or IRF3-2D was recombined into the pDonR221 plasmid using the Gateway technology and was subsequently shuttled into the pAd/CMV/V5-DEST vector to generate adenoviruses as described below. A β-galactosidase-V5–encoding adenovirus (Ad-LacZ, Life Technologies) served as a control.

### Adenovirus production and titer for cell culture

HEK293A cells were transfected with *PacI* restriction-digested adenovirus vector pAd/CMV/V5-DEST encoding cDNA for WT murine IRF3 or IRF3-2D or PGC-1α using Lipofectamine 2000 (Life Technologies). Adenovirus titer was determined by staining transduced HEK293A cells with fluorescent anti-Hexon antibody. A β-galactosidase-V5-encoding adenovirus (pAd-LacZ, Life Technologies) served as a control.

### Isolation and treatment of neonatal rat cardiomyocytes

NRCMs were isolated as previously described[102]. In brief, left ventricles from 1–2-days old Wistar rats (Charles River, Germany) were harvested and chopped in ADS buffer containing 120 mM NaCl, 20 mM HEPES, 8 mM $NaH_2PO_4$, 6 mM glucose, 5 mM KCl, and 0.8 mM $MgSO_4$, pH 7.4. About five to six enzymatic digestion steps were performed at 37 °C with 0.6 mg/ml pancreatin (Sigma-Aldrich) and 0.5 mg/ml collagenase type II (Worthington Biochemical Corporation) in sterile ADS buffer. The homogeneous cell suspension was passed through a cell strainer followed by the addition of newborn calf serum to stop the enzymatic digestion. A Percoll (GE Healthcare) gradient centrifugation was used to separate cardiomyocytes from fibroblasts and cultured in DMEM

containing 10% FCS, penicillin/streptomycin (Life Technologies), and L-glutamine (Life Technologies). About 80% confluent NRCMS were treated with 0.5 μg/ml poly I:C for 6 h and harvested for RNA isolation. Overexpression and knockdown studies for gene expression and immunoblotting was achieved using adenovirus-mediated transduction for 48 h. For hypoxia treatments, 70-80% confluent NRCMs were transduced with adenovirus for 24 h and the media was changed to DMEM supplemented with 1% penicillin/streptomycin and placed into the hypoxic incubator (1% $O_2$, 5% $CO_2$, 94% $N_2$). The media was equilibrated overnight to respective oxygen levels before use. After 24 h, cells were briefly washed with DBPS and terminated with TRIzol™ (Invitrogen #15596018) for RNA isolation.

### Reporter gene assays

The reporter gene assays were performed in 70% confluent NRCMs cultured in 24 well plates. For overexpression, NRCMs were transfected with HA-IRF3-2D, Flag-PGC-1α or empty vector (pcDNA3.1); for knock-down experiments, NRCMs were transfected with siRNA against IRF3 or *Silencer™* negative control siRNA (siCtrl) from Silencer®Select (siIRF3: Cat AM16704, ID 217531, Ambion; *Silencer* negative control: Cat AM4635, Ambion) along with plasmid constructs encoding Renilla luciferase, PPRE-luciferase reporter (PPRE X3-TK-Luc), PPARα, RXRα using Lipofectamine 3000. After 48 h of transfection, chemiluminescence was measured using a Dual-luciferase® Reporter Assay kit (E1960 Promega) on TECAN Spark multimode reader with injector module following manufacturer's instructions. Luciferase activity was normalized for transfection efficiency against Renilla activity that was used as an internal control.

### RNA isolation and gene expression analysis by quantitative PCR

Total RNA was isolated from mice cardiac ventricle tissue using the RNeasy Universal Mini kit (Cat. No. 73404, Qiagen) combined with genomic DNA elimination following the manufacturer instructions. From cultured cells, total RNA was isolated using TRIzol reagent (Invitrogen). RNA was reverse transcribed using MMLV reverse transcriptase. Gene expression was assessed by quantitative PCR (qPCR) using synthesized cDNA with SYBR Green qPCR Master Mix (Thermo Fisher) and specific primers using a 7900HT Fast Real Time PCR System (Applied Biosystems). Primer sequences used in this manuscript are listed in Source data file_ Supplemental Table 2. The relative abundance of mRNA was normalized to 36B4 mRNA as the invariant control using the $\Delta\Delta C_t$ method (where $C_t$ is the threshold cycle)[103].

### RNA-seq

mRNA was purified from total RNA using poly-T oligo-attached magnetic beads. After fragmentation, the first strand cDNA was synthesized using random hexamer primers, followed by the second strand cDNA synthesis. Libraries prepared using Novogene NGS RNA Library Prep Set (PT042) were sequenced on Illumina NovaSeq 6000 platform S4 flow cell. RNA-seq data are aligned using Hisat2 v2.0.5. Reads are assigned to transcripts using featureCounts[104]. Differential expression analysis of the data was performed using edgeR[105]. The *P* values were adjusted using the Benjamini & Hochberg method. Local version of the gene set enrichment analysis tool (http://www.broadinstitute.org/gsea/index.jsp) was used, GO, KEGG (Kyoto

Encyclopedia of genes and Genomens) and Reactome were used for GSEA independently[106,107].

## Single-cell RNA-Seq data analysis

The analysis and visualization of the data obtained from GEO accession number GSE109816 and GSE121893[35] was performed using Python v3.12.4 with the following modules: pandas (v2.2.2), matplotlib (v3.8.4), scanpy (v1.10.3) and anndata (v0.10.0).

## Mitochondrial DNA isolation from the left ventricle of rats

Mitochondrial DNA of high purity for transfection was isolated from the left ventricle tissue of 16wk old Wistar rats using mtDNA Extractor CT kit from WAKO (#291-55301) following the manual instructions. Isolated mtDNA was dissolved in TE buffer and treated with 10 μg/ml RNaseA (ThermoFisher) for 30 min at RT. About 80% confluent NRCMs were transfected with freshly isolated mtDNA (1 μg/ml) using Lipofectamine 3000 (Life Technologies) for RNA and protein analysis.

## Mitochondrial DNA content measurement by mtDNA/nDNA ratio

To access the mitochondrial DNA content, mitochondrial DNA to nuclear DNA ratio was estimated from mitochondrial DNA isolated from the left ventricle of CMI3OE and Cre controls using a method established in our lab following the detailed protocol published by Quiros et al.[108]. The primer sequences used to quantify mtDNA/nDNA ratio by accessing Nd1, 16sRNA and HK2 expression are provided in the referred article.

## Cell-Fractionation

For cell fractionation, adult cardiomyocytes were isolated from CMI3OE and Cre control mice following the Langendorff-free method from Roger Foo lab[109]. Cells were fractionated into cytoplasm, membrane bound and nuclear fractions following the protocol published by Baghirova et al.[110]. Protein in each fraction was quantified by BCA method and immunoblotting was performed with GAPDH, VDAC and Histone H3 antibody as a marker for cytoplasmic, membrane bound and nuclear protein fraction.

## Co-Immunoprecipitation assay

About 70% confluent primary cardiomyocytes were transfected with HA-IRF3-2D and Flag-PGC-1α in DMEM containing 10% FCS. Cells transfected with empty vector were used as negative control. After 6 h, media was changed and cells were culture for another 48 h in DMEM containing Pen/Strep and 10% FCS. Cells were harvested after 48 h and lysed in Cell lysis buffer (Cell signaling #9803). Protein was determined by BCA protein assay kit and 1 mg protein was used for co-immunoprecipitation using anti-Flag Dynabeads (Sigma) following manufacturer's instructions. Eluted sample (20 μl) was loaded on 4%–12% SDS-PAGE gel, transferred to nitrocellulose membrane and immunoblotted with anti-Flag and anti-HA antibody.

Similarly, co-immunoprecipitation assay in the cardiomyocytes isolated from the left ventricular tissue of CMI3OE and Cre control mice was performed using 1 mg of the protein lysate with anti-PGC-1α antibody pull down followed by incubation with Dynabeads Protein G (Invitrogen #10004D). In another assay, primary cardiomyocytes seeded in 10 cm dishes were treated with Poly I:C (2 μg/ml) for 6 h, cells were lysed in Cell lysis buffer and co-immunoprecipitation was performed using 1 mg total protein lysate as mentioned above. Immunoblots were blocked using Easy blocker (GTX425858) and incubated with primary antibody overnight. Monoclonal mouse anti-rabbit IgG HRP (Cell signaling #5127) was used as secondary antibody.

## Immunostaining and confocal imaging

For immunostaining, NRCMs seeded in 4 well μ-slides (Ibidi, Cat. #80426) were transduced with adenovirus for 48 h. Cells were washed with PBS and fixed in 4% paraformaldehyde (PFA) for 30 min at room temperature (RT). Following three PBS washes, cells were incubated in blocking solution (1x PBS containing 2.5% bovine serum albumin and 0.3% Triton X-100) for 1 h at RT. Primary antibodies, diluted 1:200 in blocking solution, were applied overnight at 4 °C. After three PBS washes, cells were incubated with secondary antibodies diluted 1:400 in blocking solution for 90 min at RT. Cells were washed in PBS and incubated with DAPI (MilliporeSigma, Cat. #10236276001), diluted 1:1000 in blocking solution, for 6 min at RT. Primary antibodies used were anti-IRF3 (Cell Signaling Technology, Cat. #4302) and anti-α-actinin (Sigma-Aldrich, Cat. #A7732). Secondary antibodies were Alexa Fluor 488-conjugated anti-mouse IgG and Alexa Fluor 647-conjugated anti-rabbit (Thermo Fisher Scientific). Confocal imaging was performed using a Leica Mica confocal microscope with a 63x HC PL APO objective.

## Immunoblotting

Tissue lysates (20-40 μg protein), cell lysates (10 μg protein) prepared in RIPA buffer (9806 Cell signaling) with appropriate protease and phosphatase inhibitors were separated by 4%-12% gradient SDS-PAGE and transferred to nitrocellulose membrane (Millipore). Blots were incubated with primary and HRP-conjugated secondary antibodies. Detection was performed with enhanced chemiluminescence (GE healthcare) and imaged on Amersham Imager 600 (GE healthcare). All bots were quantified using ImageJ software (NIH, Bethesda, MD). To view unedited blots please refer to the Source data_unedited blots.

## Antibodies

Antibodies were purchased from Cell Signaling (IRF3, 4302; pIRF3 [S396], 29047; Vinculin, 4650; HA-Tag, 3724; VDAC [D73D12], 4661; Histone H3 [D1H2], 4499; OGT [D1D8Q], 24083; p-p38MAPK [Thr180/Tyr182] 9211; p38MAPK, 9212; pAMPKα [Thr172], 2535; AMPKα, 2532; pAKT [S473], 4051; AKT, 9272; O-GlcNAc [CTD110.6], 9875), Abcam (OXPHOS, ab110413), MerckMillipore (PGC-1α, ST1202), and Sigma (Flag M2 [F1804], GAPDH [G8795]). Secondary antibodies used were: HRP goat anti rabbit (Cat. 111-035-144, Jackson ImmunoResearch Labs) and HRP goat anti-mouse (Cat. 115-035-003, Jackson ImmunoResearch Labs).

## Analysis of plasma parameters

Blood glucose levels were measured using a portable glucometer (Accu-Chek). Blood samples were collected from fed (*ad libitum*) animals in EDTA-coated blood collection tubes. Plasma insulin levels were quantified using an ELISA kit (Crystal Chem) following the manufacturer's instructions.

## Measurement of plasms free fatty acids, triglyceride and cholesterol

Plasma free fatty acids were determined using the NEFA HR (2) Assay (FUJIFILM). Plasma cholesterol and triglycerides were measured using commercial colorimetric kits (Roche, Mannheim, Germany) that were adapted to 96-well microtiter plates with Precipath as standard.

## Measurement of plasma cytokines

The concentration of TNF-α (#DY410), CCL2 (#DY479), IL-1β (#DY401), and IL-6 (#DY406) was measured in the mouse plasma samples using the equivalent DuoSet ELISA kits (R&D Systems, Minneapolis, MN, USA) following the manufacturer's protocols. Coating of Nunc-immuno modules (#469949, Thermo Scientific, Waltham, MA, USA) with the respective capture antibodies was performed overnight and 1% Bovine Serum Albumin, Fraktion V (#9048-46-8, Carl Roth, Karlsruhe, Germany) in PBS, pH 7.4 was used as a reagent diluent. To detect HRP-conjugated antibodies KPL TMB Microwell Peroxidase Substrate System (#5120-0047, Seracare, Milford, MA, USA) was used as a substrate and 1 N Sulfuric acid (#339741, Merck, Darmstadt, Germany) was

used as a stop solution. Finally, absorbance was measured at 450 nm using a Multiskan GO Microplate Spectrophotometer (Thermo Scientific, Waltham, MA, USA).

## Measurement of CK-MB

Quantitative determination of CK-MB levels was performed in plasma samples using colorimetric ELISA kit (#NBP2-75312, Novus Biologicals) following manufacturer's instructions.

## Metabolome and lipidome sample processing and data analysis

**Sample collection and processing for metabolome and lipidome.** To perform metabolome and lipidome profiling, mice were perfused with PBS, cardiac ventricle tissue samples were harvested, snap frozen in liquid nitrogen and stored at -80 °C before extraction.

**Metabolites extraction (water-soluble metabolites and lipids).** The samples were homogenized with Mixer Mill (Retsch) and steel beads at maximum frequency for 3 min in pre-cooled racks after adding ice-cold methanol/$H_2O$ (4:1, v/v, 500 μL per 10 mg tissue) with internal standards (D4-glutaric acid, and D8-phenylalanine, 4 μl Splash Lipidomix per 10 mg tissue). A total of 500 μL of homogenate was then collected and extracted by applying 120 μL 0.2 M HCl, 800 μL chloroform, and 400 μL $H_2O$ consecutively with vortex. The extracts were spun down at 16000 g for 10 min, and the upper phase (water-soluble metabolites) was transferred to a 1.5 ml vial and evaporated for 15 min at 35 °C under nitrogen and dried in SpeedVac (Eppendorf) at 15 °C overnight. The lower phase (lipids) was evaporated to dryness at 45 °C under nitrogen. The interphase was used to determine the protein concentration with the BCA assay. Samples were stored at −80 °C until processed.

**LC-MS analysis of water-soluble metabolites.** Water-soluble metabolites extract was dissolved in 100 μl 5 mM ammonium acetate (in 75% acetonitrile (v/v)) before loading to LC/MS. LC-MS analysis was performed on an Ultimate 3000 HPLC system (Thermo Fisher Scientific) coupled with a Q Exactive Plus MS (Thermo Fisher Scientific) in both ESI positive and negative mode. The analytical gradients were carried out using an Accucore 150-Amide-HILIC column (2.6 μm, 2.1 mm × 100 mm, Thermo Fisher Scientific) with solvent A (5 mM ammonium acetate in 5% acetonitrile) and solvent B (5 mM ammonium acetate in 95% acetonitrile). A total of 3 μl sample was applied to the Amide-HILIC column at 30 °C, and the analytical gradient lasted 20 min. During this time, 98% of solvent B was applied for 1 min, followed by a linear decrease to 40% within 5 min and maintained for 13 min before returning to 98% in 1 min and appended with a 5-min equilibration step. The flow rate was maintained at 350 μL/min. The eluents were analyzed with MS in ESI positive/negative mode with ddMS2. The full scan at 70k resolution (69-1000 m/z scan range, 1e6 AGC-Target, 50 ms maximum Injection Time (maxIT)) was followed by a ddMS2 at 17.5k resolution (1e5 AGC target, 50 ms maxIT, 1 loop count, 0.1 s to 10 s apex trigger, 2e3 minimum AGC target, 20 s dynamic exclusion). The HESI source parameters were set as 30 sheath gas flow rate, 10 auxiliary gas flow rate, 0 sweep gas flow rate, spray voltage: 3.6 kV in positive mode, 2.5 kV in negative mode, 320 °C capillary temperature, and the heater temperature of auxiliary gas was 120 °C. The annotation of the metabolites was performed using the El-Maven software (Elucidata, https://www.elucidata.io/el-maven) with an offset of ± 15ppm.

**LC-MS/MS analysis of lipids.** The lipids extract was dissolved in 100 μl of isopropylalcohol (iPrOH) before loading. The analytical gradients were carried out using an Accucore C8 column (2.6 μm, 2.1 mm × 50 mm, Thermo Fisher Scientific) with solvent A (acetonitrile/$H_2O$/formic acid (10/89.9/0.1, v/v/v)) and solvent B (acetonitrile/iPrOH/$H_2O$/formic acid (45/45/9.9/0.1, v/v/v/v)). 3 μl sample was applied to the C8 column at 40 °C, and the analytical gradient lasted for 35 min. During this time, 20% of solvent B was

applied for 2 min, followed by a linear increase to 99.5% within 5 min and maintained for 27 min before returning to 20% in 1 min and appended with a 5-min equilibration step. The flow rate was maintained at 350 μl/min. The full scan and ddMS2 parameters were the same as the analysis of the water-soluble metabolites, except the scan range were adjusted to 200–1600 m/z. The HESI source parameters were also adapted with a 3-sweep gas flow rate and a 3.2 kV spray voltage in positive and 3.0 kV in negative mode. Peaks corresponding to the calculated lipid masses (± 5 ppm) were integrated using El-Maven software.

## Stable-isotope tracing for metabolic flux measurement

**Sample isolation, treatment and collection.** Cardiomyocytes were isolated from 12-14wk old CMI3OE and Cre controls following the Langendorff-free method from Roger Foo lab[109]. About $1 \times 10^6$ cells were plated on laminin (5 μg/ml) pre-coated 6 cm dishes for 3 h in plating media [M199 (Sigma 4530) containing 5% FBS, 10 mM BDM and 1X Pen/Strep]. Cardiomyocytes were washed twice with PBS and treated with 5.5 mM U-$^{13}C_6$-Glucose (Merck #389374) in DMEM media (Gibco™ 11966025) for 10 min and 120 min to measure metabolic flux into glycolysis, PPP and TCA cycle. Cells were washed quickly with PBS, flash frozen in liq. $N_2$ and stored at -80 °C until analysis.

**Polar Metabolites extraction and LC-MS analysis.** Cells were scraped off in 1 ml ice-cold methanol:H2O:acetonitrile (50:20:30 v/v) containing internal standards (MSK-CAA-1, DLM-7654, DLM-9476, DLM-9045, DLM-9071, DLM-6068, DLM-831 and DLM-3487, Cambridge Isotope Laboratories). Polar metabolites were extracted using a C18 8B-S001-DAK solid phase column, as previously described[111]. Dried metabolite extracts were dissolved in 100 μl acetonitrile/water (25:75, v/v) before analysis. Water-soluble metabolites were analyzed by LC-MS/MS using a Q-Exactive Plus quadrupole-Orbitrap mass spectrometer (Thermo) interfaced with an ultra high-performance liquid chromatography (Ultimate 3000, Thermo). The analytical gradient was carried out as previously described[112] using a SeQuant ZIC-pHILIC column (5 μm, 2.1 mm × 150 mm, Millipore Sigma) with solvent A (20 mM ammonium carbonate + 0.1% ammonium hydroxide in water) and solvent B (acetonitrile). The samples (3 μL) were injected into the column at 25 °C, and the analytical gradient lasted 30 min, as follows: linear gradient from 80% to 20% within 20 min before returning to 80% in 0.5 min. and appended with a 7.5-min equilibration step. The flow rate was maintained at 150 μL/min. The eluents were analyzed with MS in ESI positive/negative mode with ddMS2. The full scan at 70k resolution (69-1000 m/z scan range, x 106 AGC-Target, 50 ms maximum injection time (maxIT)) was followed by a ddMS2 at 17.5k resolution (1 × 105 AGC target, 20 ms maxIT, 1 loop count, 2 s to 10 s apex trigger, 2 × 103 minimum AGC target, 20 s dynamic exclusion). The HESI source parameters were set as 40 sheath gas flow rate, 15 auxiliary gas flow rate, 1 sweep gas flow rate, spray voltage: 3.0 kV in positive mode, 3.1 kV in negative mode, 320 °C capillary temperature, and the heater temperature of auxiliary gas was 120 °C. Peaks were integrated using the El-MAVEN software and the R package IsoCorrectoR[113] was used to correct for natural $^{13}C$ abundance. The peak area of each target was normalized to the peak area of internal standards (positive/negative standards for metabolites) and protein concentration.

## NAD metabolomics

**Preparation of murine heart tissue for NAD metabolomics.** The heart of an anaesthetized mouse was perfused with ice-cold phosphate-buffered saline (PBS), quickly excised, and placed in a dish containing ice-cold PBS. Subsequently, the atria were removed, and the ventricle was excised by cross-section cut. ~20–25 mg of the apex was weighed and promptly transferred into a pre-chilled tube. The collected tissue sample was snap-frozen in liquid nitrogen and stored at -80 °C until further processing.

**Determination of energy carrier metabolites.** Frozen murine heart tissues (15-25 mg) were processed following an adjusted protocol targeting energy carriers such as NAD/NADH and NADP/NADPH[114]. Briefly, on ice to the frozen murine heart tissues were added ice-cooled steel balls with 250 μl cooled extraction buffer (Acetonitrile: Methanol: 15 mM ammonium acetate in $H_2O$ (3:1:1), pH 10). Subsequently, frozen material was grinded in a Retsch© Mill MM400 (1× 30 sec.; frequency 25/sec.). Afterwards, samples were centrifuged for 15 min at 4 °C and 13,000 g, and the resulting supernatant was transferred to a new LC-MS grade autosampler vial for measuring.

For metabolite separation and detection, an ACQUITY I-class PLUS UPLC system (Waters) coupled to a QTRAP 6500+ (AB SCIEX) mass spectrometer with electrospray ionization (ESI) source was used. In detail, metabolites were separated on an ACQUITY Premier BEH Amide Vanguard Fit column (100 mm × 2.1 mm, 1.7 μm, Waters) with constant column temperature of 35 °C. Separation of NAD/NADH and NADP/NADPH was achieved by the following LC gradient scheme (Table S2) using mobile phase A (50/50; Acetonitrile / Water with 5 mM ammonium acetate + 0.05% (v/v) ammonium hydroxide, pH 10) and mobile phase B (90/10; Acetonitrile: Water with 5 mM ammonium acetate + 0.05% (v/v) ammonium hydroxide, pH 10). Data acquisition was performed using Analyst 1.7.2 (AB SCIEX) and processed using the OS software suite 2.0.0 (AB SCIEX).

**Intracellular NAD + /NADH measurement in neonatal rat cardiomyocytes.** Cardiomyocytes isolated from neonatal rat ventricles were transduced with adenovirus expressing IRF3-2D or PGC-1α alone or in combination. LacZ transduced cardiomyocytes were used as a control. Intracellular nucleotides NADH, NAD and their ratio measured using NAD + /NADH Quantification Kit, (Cat. # K337-100, BioVision) following the manufacturer's instructions.

**Plasma Lipidomic analysis.** Plasma lipidomic analysis was performed using the Lipidyzer™ Platform from SCIEX. In brief, plasma samples were spiked with Lipidyzer™ Internal Standards and lipid extraction was done using an adjusted MTBE/methanol extraction protocol[115]. Briefly, plasma samples were mixed with the MTBE/methanol/water in the ratio 10:3:2.5 (v/v/v). Samples were homogenized and centrifuged at 10,000 g, 4 °C for 10 min. The upper phase containing lipids was separated, concentrated and reconstituted in a mixture of dichloromethane (50): methanol (50) containing 10 mM ammonium acetate. Separation and targeted profiling of lipid species was performed combining differential mobility spectrometry and a QTRAP® system (QTRAP® 5500; SCIEX). Lipids were quantified by the Lipidyzer™ software (Lipidomics Workflow Manager software version 1.0.5.0; SCIEX) employing specific multiple reaction monitoring transitions.

**Measurement of fuel dependency with Fuel-Flex assay using Agilent Seahorse XF.** Cardiomyocytes were isolated from neonatal rats and seeded in 96-well plate at the density of 5000 cells per well for Seahorse assays (Agilent Seahorse XFE96 Analyzer). Cardiomyocytes were transduced with adenovirus expressing LacZ (25 ifu), IRF3-2D (10 ifu) and PGC-1α (25 ifu) for 48 h. Fuel dependency was determined using Agilent Seahorse XF Mito Fuel Flex Test kit (Cat. #103270) following manufacturer guidelines. Glucose oxidation, glutamine oxidation, and fatty acid oxidation was measured using UK5099 (2 μM), BPTES (3 μM), and Etomoxir (4 μM) final concentration to the cells. Fuel dependency percentage was calculated using formula [(Baseline OCR-Target inhibitor OCR)/ (Baseline OCR-All inhibitor OCR)]*100.

**Tissue histology and Masson's-Trichrome staining.** Heart tissues were frozen in Tissue-Tek O.C.T and were cryosectioned using Cryotom Leica CM 3050S to generate heart cross-sections of 6 μM thickness. Cryosections were stained for fibrosis using Masson's-Trichrome

staining method. Imaging was done with the Leica Microsystems CMS Microscope (Cat no: DFC310 FX). The fraction of fibrotic area stained in blue was quantified using Image J software.

**Statistics.** All Biochemical and physiological data are presented as mean ± SEM. Statistical significance between two groups was measured using an unpaired two-tailed Student's t test. A one-way ANOVA followed by Šídák's multiple comparison test was performed for more than two groups containing one factor. No exclusion or inclusion criteria were used to analyze the data. RNA-seq studies were analyzed as described above. GraphPad Prism 9.0 was used for statistical calculations. P value less than 0.05 was considered statistically significant and the statistical parameters can be found in each Figure legends.

### Reporting summary
Further information on research design is available in the Nature Portfolio Reporting Summary linked to this article.

## Data availability
All data supporting the findings are available in the manuscript, supplemental file containing Lipidomics checklist, and **Source data** file. The raw RNA-seq dataset generated in this paper is submitted to NCBI Gene Expression Omnibus (GEO) and can be accessed through accession number GSE283352, GSE283353). The Metabolomics and Lipidomics data have been deposited to MetaboLights repository with the study identifier MTBLS13783. **Source data** file contains the raw numbers for graphs and boxplots, raw immunoblot images, metabolite levels (shown in Fig. 6, Fig. 7, Figure S7 and Figure S9) associated with this manuscript. Primer sequences utilized for qPCR in this manuscript are included as a Supplemental file. Source data are provided with this paper.

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

## Acknowledgements

This research work was supported by DFG grants to M.K. (KU 4356/1-1), to A.Y.R. (RA 2717/5-1). N.F., A.S. and N.P.E. acknowledge funding from DFG-SFB1550/1. The research project was initiated with Alexander von Humboldt Postdoctoral Research Fellowship to M.K. and NIH R01 DK102170 to E.D.R. supported the research tools generation. A.S., A.B.C.F. and R.C.F.S. acknowledge funding by the German Ministry of Education and Research (BMBF) within the LiSyM-Cancer networks SMART-NAFLD [031L0256A]. For the publication fee we acknowledge financial support by Heidelberg University. We thank Sandra Ehret, Birgit Henkel, Eva Maria Azizi, Alexandra Rosskopf, Walter Tauscher and Christian Heselmaier for excellent technical support. We thank Sebastian Graute, Esther Verkade, Paul Pertzborn, Laura Ehlen, and Markus Heine for helping with mice organ harvest, immunohistochemistry and AAV administration.

## Author contributions

M.K. conceived the project, analyzed data, and wrote the manuscript. T.E. procured human cardiac tissue samples. M.K., K.D.J., I.E., A.D., and N.S. carried out the experiments with help from T.T., L.L., H.C.R., C.L., and M.B.; M.K. generated transgenic mice in E.D.R. laboratory; J.H., L.S., E.D.R., A.Y.R., and M.K. wrote and edited animal applications; N.P.E performed sc-RNA-seq computational analysis; A.D. performed experiments in NRVCM's; K.D.J. and N.O.V. performed Echocardiography in CMI3OE and MI3OE mice; N.S. performed LAD ligation and Echocardiography in CMI3KO mice; A.B.C.F., R.C.F.S., G.F.K. and M.M.F. performed metabolomics study; J.J.B. performed immunofluorescence and hypoxia study. M.K., K.J., N.S., A.D., and A.Y.R. analyzed data. M.K., A.Y.R., L.S., A.S., M.F., N.F. and J.H. supervised all experiments. A.Y.R., E.D.R., N.F. and J.H. edited the manuscript. N.F., E.D.R., A.S., A.Y.R., and M.K. acquired funding.

## Funding

## Competing interests

The authors declare no competing interests.

## Additional information

¹Department of Cardiology, Angiology and Pneumology, Internal Medicine III, University Hospital Heidelberg, Heidelberg, Germany. ²Department of Biochemistry and Molecular Cell Biology, University Medical Center Hamburg-Eppendorf, Hamburg, Germany. ³DZHK (German Centre for Cardiovascular

Research), partner site Heidelberg/Mannheim, Heidelberg, Germany. [4]Institute of Experimental Cardiovascular Research, University Medical Center Hamburg-Eppendorf, Hamburg, Germany. [5]Department of Internal Medicine III, University Medical Center Schleswig-Holstein, Kiel, Germany. [6]Institute for Computational Biomedicine, University Hospital Heidelberg, Heidelberg, Germany. [7]German Cancer Research Center, Division of Tumor Metabolism and Microenvironment, Heidelberg, Germany. [8]Metabolomics Core Technology Platform, Centre for Organismal Studies (COS), Heidelberg University, Heidelberg, Germany. [9]Department of Biochemistry and Molecular Biology, Indiana University School of Medicine, Indianapolis, IN, USA. [10]Division of Pharmacology, University Hospital Heidelberg, Heidelberg, Germany. [11]Department of Cardiology, University Hospital Zurich, Zurich, Switzerland. [12]Institute of Experimental Pharmacology and Toxicology, University Medical Center Hamburg-Eppendorf, Hamburg, Germany. [13]Division of Endocrinology, Diabetes and Metabolism, Beth Israel Deaconess Medical Center, Boston, MA, USA. [14]Broad Institute of Harvard and MIT, Cambridge, MA, USA. [15]These authors contributed equally: Manju Kumari, Joerg Heeren, Norbert Frey. ✉e-mail: manju.kumari@med.uni-heidelberg.de; heeren@uke.de; norbert.frey@med.uni-heidelberg.de

