## [Transparent Peer Review file · Nature Communications]

Activation of IRF3 in cardiomyocytes impairs mitochondrial oxidative function through PGC-1 α inhibition and drives heart failure

Corresponding Author: Dr Manju Kumari

Version 0:

Reviewer comments:

Reviewer #1

(Remarks to the Author)

There is currently a plethora of published data implicating IRF3 in cardiac fibrosis and myocardial infarction. The manuscript by Kumari and colleagues extends our understanding of how IRF3 may contribute mechanistically to these conditions within cardiomyocytes. The authors used a variety of strategies including knockout, knockdown, cell-specific knockout, inducible overexpression, etc, that provides for an overall rigorous analysis of IRF3 function. Noteworthy findings are the 1) identification of PGC1a as a transcriptional target of IRF3 within cardiomyocytes, 2) regulation of metabolic mediators by IRF3 that contribute to heart failure, and 3) reversal of IRF3-driven disease by AAV overexpression of PGC1a. Although the two factors appear to be linked by negative regulation and/or interaction, there are some discrepancies in the function of unphosphorylated versus phosphorylated IRF3 that could be addressed with additional experimentation, as well as clarification of cell-specific function. Below are the major comments related to IRF3 studies.

1. While the authors show elevated IRF3 transcript levels and pIRF3 levels in both human and mouse models of ischemic cardiomyopathy, it is never addressed what is causing this upregulation. Mechanistically, this seems important to understand disease pathogenesis and to understand 1, what is driving IRF3 expression and 2, what is inducing IRF3 activation in models of heart failure. Further, data in SF1c-d suggests a differential mechanism of IRF3 induction by LPS and poly IC in cardiomyocytes than that seen in the models (TLR ligands appear more robust). This may shed light on the requirements for IRF3-mediated PGC1a expression in cardiomyocytes.
2. Alterations in transcript expression shown throughout the manuscript of a variety of genes are somewhat correlative with IRF3 expression rather than mechanistically proven. This is particularly relevant for mitochondrial and metabolic genes.
3. Specificity of CMI3KO mice was only shown in cardiomyocytes (Fig. 2b) versus organs not relevant to the heart (SF2a). Given findings of IRF3 expression within other cell types in the heart (SF1b), it would be more beneficial to examine IRF3 expression within ECs, FBs, MPs and SMCs of CMI3KO mice to exclude functional contributions.
4. Relevant to comment #3, as shown in Fig. 2, CMI3KO mice are only partially protected.
5. Data shown in SF2f is a bit confusing as it seems to represent IRF3-mediated transcriptional regulation by non-activated (non-pIRF3) IRF3 since no stimuli appears to be included. These data then suggest that pIRF3 may not be the driver and/or not necessary for transcriptional regulation. Is this instead an issue of siRNA specificity? In addition, did the authors ever examine cellular localization of IRF3 to confirm activity?
6. Fig. 4i co-immunoprecipitation experiment was done by overexpression. Why not look at interaction of endogenous proteins within the model, which would be more physiologic?
7. The mechanism(s) by which IRF3 and PGC1a regulate each other is not entirely clear. Moreover, the relevance of the interaction is not clear nor is the data shown in Fig. 4k suggesting that PGC1a somehow regulates IRF3 transcription by overexpression.

Reviewer #2

(Remarks to the Author)

In this study, Manju Kumari et al. demonstrated that phosphorylation levels of Ser396/Ser398 in the transcription factor IRF3 are significantly elevated in the left ventricles of both patients with ischemic cardiomyopathy and mouse models. The phosphorylation of IRF3 suppresses the expression of Ppargc1 α , resulting in impaired mitochondrial oxidative

phosphorylation, dysregulated NAD metabolism, and an excessive type I IFN-driven inflammatory response, collectively contributing to the development of heart failure. Notably, in IRF3-2D-overexpressing mice, restoration of Ppargc1 α expression alleviated contractile dysfunction by promoting a metabolic shift toward fatty acid oxidation and attenuating inflammatory fibrotic responses. These findings provide novel insights into the molecular mechanisms underlying heart failure, yet several critical questions remain to be addressed.

1. The authors' explanation of how IRF3 inhibits PGC-1 α function through their interaction is not completely clear. In Fig.4i, the authors described the interaction between IRF3 and PGC-1 α . Actually, the author used IRF3-2D (phosphomimic mutant) to investigate its interaction with PGC-1 α . What are the differences in the interaction strength between PGC-1 α and phosphorylated/non-phosphorylated IRF3? Was the localization of p-IRF3 and PGC-1 α examined in cardiomyocytes of Cre and CMI3OE mice using immunofluorescence? Additionally, does p-IRF3 compete with PPAR(or other PGC-1 α related transcript factors) for binding to PGC-1 α ?
2. The authors should provide the extent of cardiac fibrosis in CMI3OE mice following tamoxifen treatment. If severe fibrosis has already developed in the heart, its cellular composition would significantly differ from that of a normal heart. Therefore, it is important to determine whether the observed metabolic differences may be caused by changes in cellular composition.
3. PGC-1 α plays a role in promoting mitochondrial biogenesis. Therefore, it is necessary to compare mitochondrial mass and quantity between the Cre group and the CMI3OE group to determine whether the observed metabolic changes are caused by alterations in mitochondrial function or changes in mitochondrial quantity.
4. An increase in 3OHB alone cannot definitively indicate increased ketogenesis metabolism, as it could also be explained by reduced BOH1 and NAD⁺ levels leading to its accumulation. Additionally, it is necessary to analyze the fumarate-to-succinate ratio to assess the status of succinate oxidation, rather than relying solely on a decrease in fumarate.
5. Although the authors observed increased glucose uptake using Deoxy-D-glucose, it can be seen from Fig.S5c that both fructose-1,6-bisphosphate and phosphoenolpyruvate significantly decrease in glycolysis. This suggests that glycolysis may not actually be increased in CMI3OE group. The increased glucose may also be utilized through the pentose phosphate pathway to generate reducing power to counteract ROS. It may not necessarily be that energy metabolism shifted from FA oxidation to glycolysis in CMI3OE group (Fig.5o). Therefore, further analysis of metabolites in the PPP pathway is needed, as well as an examination of whether the transcriptomics of the glycolytic pathway reveals changes in the expression of glycolytic enzymes. It is especially important to use U-13C-glucose in vivo isotope labeling to analyze changes in the glycolysis-TCA cycle metabolic flux.

Reviewer #3

(Remarks to the Author)

This is an interesting manuscript that presents new evidence for an axis between IRF3/PGC-1 α in promoting ischemic cardiomyopathy. The work uses state of the art approaches and rigorous data analysis. These efforts make a strong case that IRF-3 activation contributes to cardiomyopathy and that PGC-1 α overexpression is protective. However, several major questions remain unresolved and should be addressed to understand the mechanisms underlying the proposed axis and maximize the value of these findings to the field.

Specific points

1. Could repair of energetics in cardiomyocytes, independent of IFN response signaling, be sufficient to protect against cardiomyopathy? Is PGC-1 α over expression protection due to metabolic repair exclusively (i.e. where inflammatory signals are not important) or a combination of metabolic and inflammatory repair? These studies make a convincing case for a connection between excessive IRF-3 activation, type I IFN response and fuel switch/defective energetics in cardiomyocytes. However, it is not clear how much of the cardiomyopathy phenotype is caused by defective energetics versus the type I IFN response. The role of inflammation in contributing to cardiomyopathy in the context of myocytes is not proven in these studies; the current studies do show augmentation of IFN response in response to injury in cardiomyocytes but stop short of proving the IFN response drives cardiomyopathy. The current studies do not measure or distinguish the individual contributions nor make a convincing case that the IFN response alone is important in these models, despite the schematic that suggests both arms are essential.
2. Separating the IFN from metabolic responses to injury/IRF3 activation is important, particularly given the introductory narrative making the point that inflammation plays context and time dependent beneficial and detrimental roles. Perhaps this critical confusion can be resolved by a focus on energetics alone: can preservation of Ox-phos sustain injury resistance despite an augmented IFN response? It seems like this may be the case because Ppargc1 α overexpression is protective against genetically augmented IRF-3 activity.
3. Is Ppargc1 α over expression sufficient to protect NRCMs from inflammatory (LPS) or ischemic injury responses?
4. Does IRF-3 binding to PGC-1 α sequester PGC-1 α and IRF-3 from their transcriptional tasks or are there other mechanisms at play for the apparently reciprocal transcriptional responses? How does the IRF-3/PCG-1 α axis in cardiomyocytes work?
5. Do the current findings exclude a role of myeloid cells in the INF-3-PGC-1-metabolism pathway? Why or why not?
6. What kinases/phosphatases are necessary for setting the balance of IRF3 phosphorylation?

Reviewer comments:

Reviewer #1

(Remarks to the Author)

The authors have done a nice job of addressing the majority of this Reviewers' comments. However, there remains concern (that was brought up by two of the Reviewers) over whether pIRF3 corresponds to nuclear/functional IRF3 in the physiologic model. The authors included a Figure 1 Revision image within the Response Letter that is not compelling as the majority of hIRF3-2D is cytoplasmic and the comparative control should be overexpressed wild-type unphosphorylated IRF3 rather than just showing minimal IRF3 expression. This image is not shown in the revised manuscript. In efforts to further address the concern, the authors performed cellular fractionation and provided a new Fig. S5b showing cytoplasmic, membrane and nuclear expression of (p)IRF3 in CMI3OE cardiomyocytes. Indeed, they show 'more' constitutive pIRF3 in both the cytoplasm and nucleus of these cells as compared to Cre mice; however, this may simply be due to overexpression of pIRF3 than a response of IRF3 to ischemic cardiomyopathy. It seems more relevant to examine IRF3 cellular localization in LAD mice (Fig. 1f). It is also concerning in this blot (new Fig. S5b) that there appears to be no 'strong' affect on PGC-1a expression. This is different than what is shown in Fig. 5i, whereby elevated IRF3 in CMI3OE mice corresponds to decreased PGC-1a expression. These apparent differences in the same cells need to be explained. Densitometry analysis could help.

Last, it would be helpful to include molecular weight markers on all uncut blots as locations for individual proteins on the blots change quite dramatically between blots, particularly for IRF3/pIRF3.

Reviewer #2

(Remarks to the Author)

The authors have addressed all my concerns and made the corresponding revisions. There are only the following minor issues remaining:

1. Figure 7a: The labeling pattern for Sedoheptulose-7-phosphate (S7P) requires correction.
2. Figure S9g: The acetyl-CoA labeling should be verified, as species above M+2 should not be present.

Reviewer #3

(Remarks to the Author)
congratulations

Version 2:

Reviewer comments:

Reviewer #1

(Remarks to the Author)

The authors have provided additional data in support of their conclusions.

RESPONSE TO REVIEWERS

We express our sincere gratitude to all the reviewers and are thankful for their thoughtful and constructive comments. We have performed several additional experiments and also added relevant text in the discussion section to address the concerns raised. We believe that the manuscript research findings have overall gained more strength. The reviewer comments are marked below in bold and our response in plain text.

Reviewer #1 (Remarks to the Author):

There is currently a plethora of published data implicating IRF3 in cardiac fibrosis and myocardial infarction. The manuscript by Kumari and colleagues extends our understanding of how IRF3 may contribute mechanistically to these conditions within cardiomyocytes. The authors used a variety of strategies including knockout, knockdown, cell-specific knockout, inducible overexpression, etc, that provides for an overall rigorous analysis of IRF3 function. Noteworthy findings are the 1) identification of PGC1a as a transcriptional target of IRF3 within cardiomyocytes, 2) regulation of metabolic mediators by IRF3 that contribute to heart failure, and 3) reversal of IRF3-driven disease by AAV overexpression of PGC1a. Although the two factors appear to be linked by negative regulation and/or interaction, there are some discrepancies in the function of unphosphorylated versus phosphorylated IRF3 that could be addressed with additional experimentation, as well as clarification of cell-specific function. Below are the major comments related to IRF3 studies.

We appreciate that the reviewer finds our study noteworthy for the research field.

1. While the authors show elevated IRF3 transcript levels and pIRF3 levels in both human and mouse models of ischemic cardiomyopathy, it is never addressed what is causing this upregulation. Mechanistically, this seems important to understand disease pathogenesis and to understand 1, what is driving IRF3 expression and 2, what is inducing IRF3 activation in models of heart failure. Further, data in SF1c-d suggests a differential mechanism of IRF3 induction by LPS and poly IC in cardiomyocytes than that seen in the models (TLR ligands appear more robust). This may shed light on the requirements for IRF3-mediated PGC1a expression in cardiomyocytes.

We agree that this is an important issue and have thus addressed the role of damage-associated molecular patterns (DAMPs) generated by sterile inflammation within the myocardium in aggravating innate immune response in the discussion section of the revised manuscript (page 22) and in the manuscript text of Fig. 1k,l. IRF3 is a transcriptional regulator of innate immune response via type I interferon signaling, expressed in various tissues and mostly localized to the cytosol under resting state indicating a role in maintaining basal cellular function. Data in Fig. S1c-d indeed show a robust innate response to TLR ligands in neonatal cardiomyocytes and this likely is not only because of the TLR ligands being robust but also due to mammalian immune response differences between the neonate and adults resulting

heightened yet uncontrolled (immature) innate immune response of neonatal cardiomyocytes compared to adult cardiomyocytes (PMID 18490660 and studies from our lab).

Several DAMPs such as mitochondrial DNA (mtDNA), dsRNA, ssRNA, heat shock proteins, S100 proteins etc. have been reported to provoke innate immune signaling in the injured myocardium (PMID 27340270). Relevant to the focus of the current study, mtDNA is a widely studied DAMP, also known to trigger the inflammatory response during cardiac injury. In the revised manuscript we show that mtDNA introduction in primary cardiomyocytes elicits an increase in type I IFN signaling along with a robust increase in phospho-IRF3 (Fig. 1k). Recent studies have identified a role for cellular mtDNA content and circulating mtDNA in disease pathology of heart failure patients (PMID 20339121, PMID 30632383, PMID 28049543). Thus, it is plausible that mtDNA released from damaged cells in myocardial infarction provoke type I IFN response in the surviving cells including cardiomyocytes resulting in IRF3 activation by phosphorylation and translocation to the nucleus. PGC-1 α is also localized in the nucleus and under stress conditions its transcriptional activity is regulated by multiple post-translational alterations (PMID 31406949, PMID 29388552). Although it is not known whether PGC-1 α is a direct transcriptional target of IRF3 in cardiomyocytes, in the revised manuscript we investigated the effect of IRF3 activation on potential factors that are known to regulate PGC-1 α posttranslational state. This is also further described in our response to the next comment below.

2. Alterations in transcript expression shown throughout the manuscript of a variety of genes are somewhat correlative with IRF3 expression rather than mechanistically proven. This is particularly relevant for mitochondrial and metabolic genes.

We agree to the reviewer that the effects on inflammatory and mitochondrial oxidative phosphorylation marker genes are correlated to IRF3 expression. Our findings in the cardiomyocyte-specific IRF3 deficient and IRF3 activation model together with effect on primary cardiomyocytes indeed suggest a mechanistic link and are in accordance of IRF3 as a transcriptional nexus between inflammation and oxidative function within cardiomyocytes via regulation of PGC-1 α . We have shown in Fig. 4g and Fig. 5i that PGC-1 α is downregulated upon IRF3 activation. Furthermore, we have shown that cardiomyocyte specific expression of AAV9-TnT-PGC-1 α could partially rescue the systolic dysfunction in CMI3OE mice (Fig. 7d-e). Our data demonstrates effect on PGC-1 α target genes in CMI3OE and also in AAV9-TnT-PGC-1 α expressing CMI3OE mice that confirms the relevance of IRF3-PGC-1 α nexus for cardiomyocyte function but certainly does not claim as the only mechanism affecting mitochondrial and metabolic genes. In the revised manuscript we show the potential of PGC-1 α in protecting cardiomyocytes from type I IFN (LPS) and hypoxia driven side effects by downregulating inflammation and mitochondrial damage in cardiomyocytes (Fig S6a-d). Overall, these studies provide mechanistic insight into the bidirectional role of IRF3-PGC-1 α axis regulating cardiac sterile inflammation and metabolism.

3. Specificity of CMI3KO mice was only shown in cardiomyocytes (Fig. 2b) versus organs

not relevant to the heart (SF2a). Given findings of IRF3 expression within other cell types in the heart (SF1b), it would be more beneficial to examine IRF3 expression within ECs, FBs, MPs and SMCs of CMI3KO mice to exclude functional contributions.

We have utilized α MHC-Cre mouse to generate cardiomyocyte-specific *Irf3*KO (CMI3KO) that resulted in a target allele deletion when crossed to a mouse with a floxed *Irf3* transgene. α MHC-Cre has been previously shown to be cardiac specific with no expression in other tissues (lung, liver, skeletal muscle, spleen) (PMID 9202069). To rule out any functional contribution from other cell types, in addition to cardiomyocytes we isolated other cell types of the heart (cardiac fibroblasts, cardiac endothelial cells and other pooled cardiac cells [after separating cardiomyocytes and fibroblasts]) to examine IRF3 expression in adult CMI3KO mice (Fig. S2b). We found no effect on *Irf3* levels in these cell types by α MHC-mediated IRF3 deletion.

4. Relevant to comment #3, as shown in Fig. 2, CMI3KO mice are only partially protected.

We agree with the reviewer that despite IRF3 deficiency in cardiomyocytes, basal IRF3 levels in other cell types is likely the cause of the partial protection of CMI3KO mice from myocardial infarction induced systolic dysfunction. In the revised manuscript, we have included this info in the text of Fig. 2e-g.

5. Data shown in SF2f is a bit confusing as it seems to represent IRF3-mediated transcriptional regulation by non-activated (non-pIRF3) IRF3 since no stimuli appears to be included. These data then suggest that pIRF3 may not be the driver and/or not necessary for transcriptional regulation. Is this instead an issue of siRNA specificity? In addition, did the authors ever examine cellular localization of IRF3 to confirm activity?

This is an important point raised by the reviewer. We show that the siRNA mediated knock down of IRF3 reduces the overall inflammatory gene expression in NRCMs in the absence of a stimulus (Fig. S2e,f, h). There are two potential reasons for this effect on inflammatory markers. First, we and others have noted an elevated and rapid inflammatory response in NRVCN compared to adult cardiomyocytes (PMID 18490660). Second, both siCtrl and siIRF3 is introduced into cardiomyocytes by transfection using Lipofectamine reagent. Our results and previous reports (PMID 9858316) have shown that liposome based reagents and control siRNA per se may result in significant but generalized induction of interferons and ISG (interferon stimulated genes) compared to un-transfected cells possibly due to cationic lipid composition of the Lipofectamine reagent and the mere presence of a non-specific siRNA (short dsRNA) as a DAMP. Thus, the effect seen at basal level due to siIRF3 is against transfection stress of siCtrl RNA. Additionally, we have also done siIRF3 KD in presence of LPS and in the revised manuscript we now show that LPS mediated inflammatory response was severely impaired with IRF3 KD in primary cardiomyocytes (Fig. S2h). Furthermore, we have tested siIRF3 specificity and with the exception of IRF7 no other IRF is modulated by siIRF3 knock-down in primary cardiomyocytes in this study (Fig. S2g). It has been known that IRF7 is a transcriptional target of IRF3 and IRF3-IRF7 can dimerize under stress conditions where the primary effect is driven by IRF3 (PMID 33205822). Thus, siIRF3 effect on IRF7 mRNA level is not surprising. We have

added this info in the revised manuscript text on page 9. Cellular localization of IRF3 has been asked for by reviewer 2 (point 1) as well. In the revised manuscript we thus performed cell fractionation from the isolated cardiomyocytes of CMI3OE mice and show enriched nuclear expression of IRF3-2D in CMI3OE mice compared to Cre controls (Fig. S5b).

6. Fig. 4i co-immunoprecipitation experiment was done by overexpression. Why not look at interaction of endogenous proteins within the model, which would be more physiologic?

As per the reviewer's suggestion to assess the physiological interaction within the model, we have now included an endogenous Co-IP in CMI3OE mouse model (Figure 4j). These results indeed confirm that IRF3-2D interacts with PGC-1 α .

7. The mechanism(s) by which IRF3 and PGC1a regulate each other is not entirely clear. Moreover, the relevance of the interaction is not clear nor is the data shown in Fig. 4k suggesting that PGC1a somehow regulates IRF3 transcription by overexpression.

We thank the reviewer for this comment and agree that how IRF3 regulates PGC-1 α gene expression is not completely clear. We had also noted this in the manuscript under "Limitations of the study". Yet, it is worth noting that PGC-1 α expression is low in pre- and -neonatal heart where innate inflammatory response can be paradoxically high. In the revised manuscript, we have shown that PGC-1 α expression is sufficient to suppress inflammatory effects alone and this characteristic is maintained even in the presence of extravagant IFN response induced by LPS or under hypoxia condition in NRCMs (Fig. 5c, d and Fig. S6a-d). Thus, effect of IRF3 and PGC-1 α on each other's levels is highly relevant.

We have not been successful with IRF3 ChIP in cardiomyocytes per se and this makes differentiating direct and indirect effect of IRF3-PGC-1 α interaction exceedingly difficult. We understand that interaction studies using Co-IP apparently speaks of the close proximity of two factors but does not provide insight into their function. Due to multiple interacting partners of PGC-1 α the relationship between IRF3 and PGC-1 α is perhaps not a linear pathway but rather a complex network. However, the below mentioned new findings in the revised manuscript provide further insight into indirect regulation of PGC-1 α levels by IRF3 activation and enhances our understanding of how IRF3 and PGC-1 α may influence each other's function:

- 1) PGC-1 α is known to interact with its cognate transcription factors and thereby influence a broad range of metabolic pathways (FAO and glucose metabolism). *Foxo1*, *Mef2*, and *Esrra*, are well-known and important transcriptional targets of *Ppargc1a* regulating mitochondrial function, metabolism and β oxidation (PMID 16511594). We now show that IRF3 activation downregulates *Esrra*, *Foxo1* and *Mef2* gene expression in the myocardium of CMI3OE mice (Fig. 5h).
- 2) Using Luciferase assay we have shown that IRF3 activation impairs PGC-1 α activity on PPAR α reporter. We also found reduced expression of PPAR α target genes (*Cd36*, *Cpt1b*, *Acox1*, *Acadm*, *Acadvl*) involved in FA transport and mitochondrial β oxidation

upon IRF3 activation in CMI3OE (Fig. S6e). Similarly, PGC-1 α expression reduced IRF3 direct target gene expression in cardiomyocytes as shown in the new Fig. 5c.

- 3) Apart from direct transcriptional control, PGC-1 α expression can also be regulated by panoply of posttranslational modifications such as phosphorylation and O-GlcNAc modification (PMID 31406949, PMID 27646831, PMID 19103600). AMPK, p38MAPK and AKT are among the key kinases that are known to regulate PGC-1 α levels. We found reduced p-p38MAPK levels in CMI3OE mice along with reduced expression of *Mapk14* (encoding p38MAPK) (Fig. 5i,k). There was no effect on pAMPK levels, whereas both total AKT as well as pAKT was reduced in the myocardium of CMI3OE mice. O-GlcNAcylation is another novel pathway to regulate PGC-1 α stability. O-GlcNAc transferase (OGT) is the sole enzyme that catalyzes O-GlcNAcylation by transfer of N-acetylglucosamine from UDP-N-acetylglucosamine to serine/threonine residues of cytosolic and nuclear proteins. Interestingly, we found increased levels of UDP-N-acetylglucosamine in CMI3OE mice (Fig. 6e) indicating reduced activity of OGT. While increased OGT levels are detrimental for heart function, reduced OGT levels have also been shown to exacerbate heart failure (PMID 20876116). These data together suggest metabolic and transcriptional adaptation upon IRF3 activation in the myocardium.

Overall, our above mentioned findings suggest that the IRF3 and PGC-1 α interaction may divert them from their transcriptional task by direct or indirect interactive pathways.

Of note, IRF3 has been traditionally studied with inflammatory perspective, on the other hand most of the studies highlight PGC-1 α role in mitochondrial function. Here we show bidirectional effect of IRF3-PGC-1 α expression wherein IRF3 activation downregulates PGC-1 α expression (now Fig. 5a, earlier Fig. 4j) along with downregulation of mitochondrial oxidative phosphorylation marker genes (Fig. S5g, in the revised manuscript) whereas PGC-1 α overexpression downregulates inflammatory marker gene expression (Fig. 5b, c in the revised manuscript) including IRF3 (Fig. 5b, earlier Fig. 4k). We believe that these studies contributes to an improved understanding by extending IRF3 and PGC-1 α function beyond their classical boundaries and highlight a broader role in cardiac immuno-metabolism.

Reviewer #2 (Remarks to the Author):

In this study, Manju Kumari et al. demonstrated that phosphorylation levels of Ser396/Ser398 in the transcription factor IRF3 are significantly elevated in the left ventricles of both patients with ischemic cardiomyopathy and mouse models. The phosphorylation of IRF3 suppresses the expression of Ppargc1 α , resulting in impaired mitochondrial oxidative phosphorylation, dysregulated NAD metabolism, and an excessive type I IFN-driven inflammatory response, collectively contributing to the development of heart failure. Notably, in IRF3-2D-overexpressing mice, restoration of Ppargc1 α expression alleviated contractile dysfunction by promoting a metabolic shift toward fatty acid oxidation and attenuating inflammatory fibrotic responses. These findings provide novel insights into the molecular mechanisms underlying heart failure, yet several critical questions remain to be addressed.

We are thankful to the reviewer for appreciating the novel findings in this article and overall positive comments.

1. The authors' explanation of how IRF3 inhibits PGC-1 α function through their interaction is not completely clear. In Fig.4i, the authors described the interaction between IRF3 and PGC-1 α . Actually, the author used IRF3-2D (phosphomimic mutant) to investigate its interaction with PGC-1 α . What are the differences in the interaction strength between PGC-1 α and phosphorylated/non-phosphorylated IRF3? Was the localization of p-IRF3 and PGC-1 α examined in cardiomyocytes of Cre and CMI3OE mice using immunofluorescence? Additionally, does p-IRF3 compete with PPAR(or other PGC-1 α related transcript factors) for binding to PGC-1 α ?

Due to lower endogenous expression level and antibody specificity concerns related to one or both of the interacting factors/proteins, co-immunoprecipitation (Co-IP) assays are often performed with tagged proteins or overexpression system. For these reasons, we had used NRCMs overexpressing Flag-tagged PGC-1 α and IRF3-2D to show interaction by Co-IP. In the revised manuscript, to investigate the physiological interaction between PGC-1 α and IRF3-2D in the model we have used to study cardiac function, we now performed endogenous Co-IP and confirmed the interaction between PGC-1 α and IRF3-2D in the ventricular tissue of CMI3OE mice compared to healthy Cre controls (Fig. 4j). This also addresses a similar concern of reviewer 1 (point 6). Furthermore, we also performed Co-IP assays in NRCMs treated with Poly I:C to induce IRF3 phosphorylation and show that PGC-1 α interacts with phosphorylated IRF3 and not wild type IRF3 most likely due to nuclear localization of p-IRF3 (Fig. S5a).

Due to lack of optimal antibodies for the immunofluorescence (IF) studies in mouse cardiomyocytes, we instead performed cell fractionation of isolated adult cardiomyocytes. Nuclear localization of IRF3-2D and PGC-1 α in the cardiomyocytes of CMI3OE mice can clearly be seen with appropriate controls (Fig. S5b). Although IF in mouse samples for IRF3 are not optimized yet in our lab (likely due to epitope conformation/accessibility differences), we have successfully seen nuclear localization of human IRF3-2D in hiPSC-cardiomyocytes transduced with adenovirus expressing human IRF3-2D compared to LacZ as shown here in Fig. 1Revision.

Fig. 1Revision: Immunofluorescence in hiPSC-cardiomyocytes transduced with LacZ control or human IRF3-2D adenovirus at 10⁶ ifu for 48 h showing nuclear localization of human IRF3-2D.

We have not been successful with IRF3 ChIP in cardiomyocytes per se and this makes identifying direct effects of IRF3-PGC-1 α interaction exceedingly difficult. Nevertheless, using

Luciferase reporter assays, we show that IRF3-2D expression attenuated PGC-1 α activity on PPRE Luc (Fig. 5f, g). In the revised manuscript we furthermore show that IRF3 activation resulted in reduced expression of *Ppara* target genes (*Cd36*, *Acox1*, *Acadm*, *Acadvl*, *Cpt1b*) (Fig. S5g) and *Ppargc1a* transcriptional targets (*Esrra*, *Mef2a*, and *Foxo1*) (Fig. 5h). Furthermore, apart from direct transcriptional control, PGC-1 α expression can also be regulated by posttranslational modifications such as phosphorylation and O-GlcNAcylation. In the revised manuscript we show reduced p-p38MAPK and OGT levels in the myocardium of CMI3OE mice providing additional insight into the effect of IRF3-PGC-1 α axis within cardiomyocytes (Fig. 5i,j).

2. The authors should provide the extent of cardiac fibrosis in CMI3OE mice following tamoxifen treatment. If severe fibrosis has already developed in the heart, its cellular composition would significantly differ from that of a normal heart. Therefore, it is important to determine whether the observed metabolic differences may be caused by changes in cellular composition.

We thank the reviewer for raising this important point. We have now included Masson's Trichrome stained ventricular IHC images of CMI3OE mice in the revised manuscript (Fig. S4e). We could not detect increased fibrosis by IHC indicating obvious structural changes with severe fibrosis have not developed. However, at mRNA level, the expression of fibrotic markers are significantly elevated upon IRF3 activation in CMI3OE mice (Fig. 4b) and can also be seen with α SMA protein levels included in the new Fig. S4f. CMI3OE were treated with low dose Tamoxifen by gavage and the cardiac tissue was analyzed on 5th day after the start of the treatment. While severe fibrosis is not set in during this period, we believe that IRF3 activation does provoke pro-fibrotic signaling with potential to change cellular metabolism and cellular composition. Our laboratory is actively investigating the crucial role of IRF3 in modulating cardiomyocyte microenvironment which is part of a follow up manuscript-in-preparation. We have addressed this in the revised manuscript Results section (para 1, page 12) where we wrote "Despite significantly elevated expression of fibrotic markers, structural changes indicative of fibrosis were not observed suggesting an early phase of cellular adaptation in CMI3OE mice (Fig. S4e, S4f)".

3. PGC1- α plays a role in promoting mitochondrial biogenesis. Therefore, it is necessary to compare mitochondrial mass and quantity between the Cre group and the CMI3OE group to determine whether the observed metabolic changes are caused by alterations in mitochondrial function or changes in mitochondrial quantity.

In the revised manuscript we have quantified mitochondrial DNA copy number by determining the mitochondrial DNA/nuclear DNA ratio in ventricular tissue of CMI3OE mice compared to Cre controls to assess an effect on mitochondrial DNA content/mass (Fig. 4g). These results show that indeed mitochondrial function and not the quantity is altered in the heart of CMI3OE mice.

4. An increase in 3OHB alone cannot definitively indicate increased ketogenesis metabolism, as it could also be explained by reduced BOH1 and NAD⁺ levels leading to

its accumulation. Additionally, it is necessary to analyze the fumarate-to-succinate ratio to assess the status of succinate oxidation, rather than relying solely on a decrease in fumarate.

We agree and thank the reviewer for highlighting the importance of fumarate to succinate ratio that is an indicator of SDH activity and succinate oxidation in TCA cycle. We have analyzed and included the fumarate-succinate ratio to assess succinate oxidation in the revised manuscript (Fig. 6k). The analysis showed reduced succinate oxidation in the myocardium of CMI3OE mice.

5. Although the authors observed increased glucose uptake using Deoxy-D-glucose, it can be seen from Fig.S5c that both fructose-1,6-bisphosphate and phosphoenolpyruvate significantly decrease in glycolysis. This suggests that glycolysis may not actually be increased in CMI3OE group. The increased glucose may also be utilized through the pentose phosphate pathway to generate reducing power to counteract ROS. It may not necessarily be that energy metabolism shifted from FA oxidation to glycolysis in CMI3OE group (Fig.5o). Therefore, further analysis of metabolites in the PPP pathway is needed, as well as an examination of whether the transcriptomics of the glycolytic pathway reveals changes in the expression of glycolytic enzymes. It is especially important to use U-13C-glucose in vivo isotope labeling to analyze changes in the glycolysis-TCA cycle metabolic flux.

We sincerely thank the reviewer for this detailed observation and thoughtful comment. This prompted us to perform stable isotope labeling assays and we could identify a unique capacity of cardiomyocyte to adapt towards PPP for fuel utilization upon IRF3 activation. Cardiac injury is followed by swift metabolic adaptations creating a complex interplay between cellular glycolysis, TCA cycle, and PPP wherein the flow of each pathway is indeed context dependent. In our lipidomic study, the metabolic changes in the ventricular tissue reflect the static metabolic state but may not represent the cardiomyocyte intrinsic effects driven by IRF3 activation. Stable isotope labelling represents the true metabolic flux status, thus, we investigated metabolic flux in glycolysis, TCA cycle and PPP in adult cardiomyocytes isolated from Cre controls and CMI3OE mice utilizing U-¹³C₆-Glucose. Our studies in CMI3OE mice highlight the previously unknown potential of the cardiomyocyte's response towards sterile inflammation. Strikingly, we found an increase in Sedoheptulose-7-phosphate (at 120 min) which represents increased flux in PPP in cardiomyocytes with IRF3 activation (Fig. 7f). Additionally, we found decreased flux in TCA cycle in cardiomyocytes (Fig. 7g-j, Fig. S9f) at 120 min. These metabolic changes were also captured in the transcriptomics analysis and we show that expression of genes regulating upper glycolysis, TCA cycle and PPP pathway were altered in the same pattern in CMI3OE mice (Fig. 6l,m; Fig.7k). While increase in glucose uptake, increased UDP-Glucosamine and increased expression of *Hk1* and *Gpi1* primarily indicated a transient increase in upper glycolysis pathway, there was no major effect on glycolysis flux at 10 min (Fig. 7b-e, Fig. S9a-e).

We have also modified the Fig. 5o (now Fig. 7k) and updated the text in the manuscript Results section to accommodate these new findings. Taken together, alterations in the metabolic flux in

PPP and TCA cycle define the unique flexibility of cardiomyocytes and strengthen the cell-autonomous effect of IRF3 activation more precisely in maintaining metabolic homeostasis.

Reviewer #3 (Remarks to the Author):

This is an interesting manuscript that presents new evidence for an axis between IRF3/PGC-1 α in promoting ischemic cardiomyopathy. The work uses state of the art approaches and rigorous data analysis. These efforts make a strong case that IRF-3 activation contributes to cardiomyopathy and that PGC-1 α overexpression is protective. However, several major questions remain unresolved and should be addressed to understand the mechanisms underlying the proposed axis and maximize the value of these findings to the field.

We thank the reviewer for appreciating our work and providing thoughtful comments.

Specific points

1. Could repair of energetics in cardiomyocytes, independent of IFN response signaling, be sufficient to protect against cardiomyopathy? Is PGC-1 α over expression protection due to metabolic repair exclusively (i.e. where inflammatory signals are not important) or a combination of metabolic and inflammatory repair? These studies make a convincing case for a connection between excessive IRF-3 activation, type I IFN response and fuel switch/defective energetics in cardiomyocytes. However, it is not clear how much of the cardiomyopathy phenotype is caused by defective energetics versus the type I IFN response. The role of inflammation in contributing to cardiomyopathy in the context of myocytes is not proven in these studies; the current studies do show augmentation of IFN response in response to injury in cardiomyocytes but stop short of proving the IFN response drives cardiomyopathy. The current studies do not measure or distinguish the individual contributions nor make a convincing case that the IFN response alone is important in these models, despite the schematic that suggests both arms are essential.

We agree with the reviewer's comment that the individual contribution of IFN response vs maladaptive cardiac metabolism is not segregated in CMI3OE model. Earlier studies from King et al (PMID 29106401) and Ninh et al (PMID 39198639) have proven the role of IRF3 driven inflammation in myocardial infarction. Findings from these studies support the conclusion that IRF3 driven IFN signaling has the potential to drive cardiomyopathy. However, these studies focused on the role of IRF3 upregulation in myeloid cells towards cardiac inflammation or identification of IFN induced cells in myocardial infarction model and did not address a role for IRF3 on cardiac metabolism or cardiomyocytes in those inflamed hearts. Our data in CMI3OE, CMI3KO and cell-autonomous studies in primary cardiomyocytes indeed suggests a bidirectional role of IRF3 and PGC-1 α in regulating inflammation and cardiac metabolism which in our view is novel and fascinating. However, this bidirectional effect apparently comes with a challenge to deconvolute the contribution of their respective functions. Our stable isotope

labeling studies highlights that swift metabolic adaptations occur in the presence of IRF3 mediated sterile inflammation and are essential to track earlier changes leading to an energy deficit. PGC-1 α expression partially rescued cardiac dysfunction by reducing cardiac inflammation and increasing FA oxidation pathways. IRF3-PGC-1 α axis is indeed one of the mechanisms but certainly does not claim as the only mechanism affecting mitochondrial and metabolic genes. Overall “limited studies including ours highlight the importance of inflammation and cardiac energy deficit working in parallel. However, the *kinetics* of inflammation and metabolic changes may not be parallel in each context (PMID 37086246) and with lack of this knowledge an attempt to prioritize one or the other pathway may lead to a causality dilemma”. To emphasize this fundamental aspect we have included this text in the last paragraph of the revised manuscript (page 25).

2. Separating the IFN from metabolic responses to injury/IRF3 activation is important, particularly given the introductory narrative making the point that inflammation plays context and time dependent beneficial and detrimental roles. Perhaps this critical confusion can be resolved by a focus on energetics alone: can preservation of Ox-phos sustain injury resistance despite an augmented IFN response? It seems like this may be the case because Ppargc1a overexpression is protective against genetically augmented IRF-3 activity.

Because the beneficial or detrimental outcome of inflammation is both context and time dependent, our data showing that PGC-1 α expression rescues cardiac systolic dysfunction in CMI3OE mice indeed emphasizes that the prompt control of oxidative metabolic homeostasis at the onset of inflammation occurring during the early phase of cardiac injury appears beneficial. Both extreme upregulation and downregulation of PGC-1 α has been associated with heart failure (PMID 31412728). To note, we have shown that apart from improved OXPHOS levels, moderate PGC-1 α expression also reduced overall cardiac inflammation in CMI3OE (Fig. 9b). We believe that reduced inflammation due to PGC-1 α (Fig. 9b, Fig. 5c) plays a critical role in improving cardiac function apart from the effect on energetics alone. Taken together, our results suggest that both restored OXPHOS and attenuated but not diminished cardiac inflammation are required to maintain cardiac function. Furthermore, in response to this comment as well as the next (see below), our findings from cardiomyocytes expressing PGC-1 α highlight that the anti-inflammatory properties of PGC-1 α are indeed maintained in the presence of type I IFN (LPS) stimuli in vitro (Fig. S6a, b). However, despite amelioration of acute cardiac dysfunction, the long term effect of persistent inflammation and the role of improving oxidative function in cardiac muscle remains unknown. This is being actively pursued in our lab using other HF models due to limited survival of CMI3OE model.

3. Is Ppargc1a over expression sufficient to protect NRCMs from inflammatory (LPS) or ischemic injury responses?

We sincerely thank the reviewer for prompting us to perform this interesting experiment. Indeed *Ppargc1a* expression in NRVCm protects the cells from LPS induced inflammatory response

(Fig. S6a-d). PGC-1 α expression also led to increased expression of mitochondrial marker genes in a hypoxia model. These results also suggest that *Ppargc1a* exerts stronger effect in an inflammation inducing model. Overall, this again confirmed the bidirectional role of PGC-1 α in improving mitochondrial function as well as reducing IFN signaling in cardiomyocytes.

4. Does IRF-3 binding to PGC-1 α sequester PGC-1 α and IRF-3 from their transcriptional tasks or are there other mechanisms at play for the apparently reciprocal transcriptional responses? How does the IRF-3/PGC-1 α axis in cardiomyocytes work?

In the revised manuscript we provide further mechanistic insight into the reciprocal regulation of IRF3-PGC-1 α levels in cardiomyocytes as a similar comment has been raised by another reviewer. We show activated IRF3 (IRF3-2D/pIRF3) and PGC-1 α interaction by native co-immunoprecipitation assay (Fig. 4j). PGC-1 α does not directly bind to DNA but influence other transcription factor's activity. In neonatal heart, innate inflammatory response is paradoxically high whereas PGC-1 α expression is low. On the other hand, PGC-1 α expression increases in adult heart along with the establishment of adaptive immune response and oxidative metabolism. Thus, the effect of IRF3 and PGC-1 α on each other inflammatory and metabolic roles in cardiomyocytes is highly relevant.

PGC-1 α has multiple transcriptional targets in different tissues. We found reduced expression of *Esrra*, *Mef2* and *Foxo1* in the myocardium of CMI3OE indicating reduced PGC-1 α transcriptional activity (Fig. 5h). Similarly, *lfits* (*lfit1*, *lfit2* and *lfit3*) and *Rsad2* are recognized as IRF3 targets and PGC-1 α expression reduced their expression in cardiomyocytes (Fig. 5c). PGC-1 α levels and activity can also be controlled by various post-translational regulations such as phosphorylation and O-GlcNAcylation. We now show that p-P38MAPK and OGT (O-linked N-acetylglucosamine transferase) levels that are known to regulate PGC-1 α levels by phosphorylation and O-GlcNAcylation were downregulated at mRNA and protein level in the myocardium of CMI3OE mice (Fig. 5i,k). pAMPK and pAKT also play an important role in PGC-1 α regulation. However, there was no effect on pAMPK α levels, whereas both total AKT as well as pAKT was reduced in the myocardium of CMI3OE mice indicating a tissue specific effect of IRF3 activation. OGT is the sole enzyme that catalyzes O-GlcNAcylation by transfer of N-acetylglucosamine from UDP-N-acetylglucosamine to serine/threonine residues of cytosolic and nuclear proteins. This O-GlcNAcylation is strictly dependent on the UDP-N-acetylglucosamine level which is a substrate for OGT activity. Interestingly, we found increased levels of UDP-N-acetylglucosamine in CMI3OE mice (Fig. 6e) indicating reduced activity of OGT. While increased OGT levels are detrimental for heart function, reduced OGT levels have also been shown to exacerbate heart failure (PMID 20876116). Taken together, these results identify a context specific role of IRF3 activation in the myocardium.

5. Do the current findings exclude a role of myeloid cells in the INF-3-PGC-1-metabolism pathway? Why or why not?

Because IRF3 upregulation has been noted in the myeloid cells upon ischemic injury (PMID 29106401) and PGC-1 α has been shown to influence macrophage polarization towards M2

state (PMID 32215168), our findings do not exclude a role of myeloid cells per se in IRF3-PGC-1 α metabolic regulation. In addition, IRF3 activation in cardiomyocytes altered circulating chemokine levels (Fig. 3h), thus, we are aware that paracrine effects likely can also alter the function of other cell-types including myeloid cells and overall modulate cardiac metabolism. In this regard, we had included a text in the Limitations of this study stating “Importantly, increased chemokine signaling suggests further research required to unveil the potential of IRF3 to modulate autocrine or paracrine processes regulating cardiac function”.

Of note, in the revised manuscript, our data from the metabolic flux assay using U-¹³C₆-Glucose in isolated cardiomyocytes from CMI3OE and Cre control mice shows that IRF3 activation increased flux into PPP along with a reduced flux in TCA cycle. These findings from metabolic flux assays in adult cardiomyocytes and cardiomyocyte-specific IRF3 activation model highlights that cardiomyocytes possess the potential to initiate type I IFN inflammatory response and influence cardiac metabolism to a greater extent than previously anticipated. However, these studies do not exclude contribution of other cell types within myocardium towards immuno-metabolism.

6. What kinases/phosphatases are necessary for setting the balance of IRF3 phosphorylation?

The kinases and phosphatases necessary for maintaining pIRF3/IRF3 balance have been mostly studied in immune cells. We have included this info in the discussion of the revised manuscript (page 21) stating that “IKK ϵ and TANK-binding-kinase 1 (TBK1) are the two major phosphorylation kinases of IRF3 (PMID 39384770), whereas IRF3 can be deactivated by dephosphorylation mediated by serine threonine phosphatase PP2A and its adaptor protein RACK1 (PMID 24726876).

RESPONSE TO REVIEWERS

We are again thankful to all the reviewers for their thoughtful comments. We believe that in the revised manuscript we have now addressed all the remaining concerns from the reviewers by performing required experiments or additional analyses. The reviewer comments are marked below in bold and our response in plain text.

Reviewer #1 (Remarks to the Author):

The authors have done a nice job of addressing the majority of this Reviewers' comments. However, there remains concern (that was brought up by two of the Reviewers) over whether pIRF3 corresponds to nuclear/functional IRF3 in the physiologic model. The authors included a Figure 1 Revision image within the Response Letter that is not compelling as the majority of hIRF3-2D is cytoplasmic and the comparative control should be overexpressed wild-type unphosphorylated IRF3 rather than just showing minimal IRF3 expression. This image is not shown in the revised manuscript. In efforts to further address the concern, the authors performed cellular fractionation and provided a new Fig. S5b showing cytoplasmic, membrane and nuclear expression of (p)IRF3 in CMI3OE cardiomyocytes. Indeed, they show 'more' constitutive pIRF3 in both the cytoplasm and nucleus of these cells as compared to Cre mice; however, this may simply be due to overexpression of pIRF3 than a response of IRF3 to ischemic cardiomyopathy. It seems more relevant to examine IRF3 cellular localization in LAD mice (Fig. 1f). It is also concerning in this blot (new Fig. S5b) that there appears to be no 'strong' affect on PGC-1a expression. This is different than what is shown in Fig. 5i, whereby elevated IRF3 in CMI3OE mice corresponds to decreased PGC-1a expression. These apparent differences in the same cells need to be explained. Densitometry analysis could help.

Last, it would be helpful to include molecular weight markers on all uncut blots as locations for individual proteins on the blots change quite dramatically between blots, particularly for IRF3/pIRF3.

We thank the reviewer for appreciating our work done to address the concerns raised. We agree with the reviewer that indeed the nuclear expression of IRF3 might be influenced by high expression in the transgenic CMI3OE model. In order to determine IRF3 activation by phosphorylation in a physiological model, we isolated adult cardiomyocytes from mice subjected to LAD-induced myocardial infarction and sham controls. Cell fractionation of these cardiomyocytes showed robust increase in nuclear pIRF3 levels in ischemic mice compared to sham controls (Fig. S1c). We have now added the following text on page 7 of the revised manuscript to include this finding, "Furthermore, immunoblot analysis showed higher levels of phosphorylated IRF3 in the nuclear fraction of cardiomyocytes isolated from mice with myocardial infarction suggesting physiological activation of endogenous IRF3 (Fig. S1c)".

Of note, in vitro experiments in this manuscript are carried out in cardiomyocytes isolated from neonatal rats or adult mouse heart. Thus, immunofluorescence images from hiPS-cardiomyocytes were not included in the manuscript but shown as a response to the reviewer in the previous revision. Meanwhile, we have taken multiple efforts to optimize the immunofluorescence (IF) in neonatal rat cardiomyocytes since the last revision. In the revised manuscript, we show that wild type IRF3 resides predominantly in the cytoplasm whereas IRF3-2D showed higher nuclear abundance compared to LacZ and wild-type IRF3 (Fig. S5f). We also added the following text on page 13, “Second, expression of IRF3-2D in isolated cardiomyocytes showed enhanced nuclear abundance (Fig. S5d-S5f)”. We appreciate the reviewer’s feedback and do acknowledge observing IRF3-2D expression as a halo around nucleus in cultured cardiomyocytes, in both neonatal rat cardiomyocytes as well as hiPS-cardiomyocytes. We believe that there can be multiple reasons for this observation in IF images such as: **a)** technical limitations of the antibody permeability to nucleus, or **b)** interplay between phosphatases vs kinases in regulating IRF3 levels in the cytosol in response to IRF3 activation to maintain cellular homeostasis (PMID 24726876, PMID 39384770), or **c)** binding to unknown nuclear envelop proteins/associated organelles upon IRF3-2D expression, or **d)** requirement of further optimization of blocking, permeability and antibody dilution conditions.

To address the next concern, we performed densitometry analysis and quantified PGC-1 α expression in the cell fractions of adult cardiomyocytes isolated from CMI3OE mice and normalized to the respective loading controls of cell fractions. In the revised manuscript, Figure S5c shows significant downregulation of ~30% for PGC-1 α level in cardiomyocytes of CMI3OE mice. We agree that this downregulation is lower than the Figure 5i (probably due to cell-fractionation processing), nevertheless PGC-1 α downregulation remains significant in multiple cohorts we have studied.

Finally, we ofcourse agree with the reviewer’s suggestion and have added molecular weight to all the original uncut blots.

Reviewer #2 (Remarks to the Author):

The authors have addressed all my concerns and made the corresponding revisions.

There are only the following minor issues remaining:

- 1. Figure 7a: The labeling pattern for Sedoheptulose-7-phosphate (S7P) requires correction.**
- 2. Figure S9g: The acetyl-CoA labeling should be verified, as species above M+2 should not be present.**

We thank the reviewer for these careful observations. We have corrected the isotope labeling of Sedoheptulose7P in Fig 7a.

For the acetyl-CoA labeling in Fig. S9g, the isotopologues starting from M+3 to M+23 were excluded, as they do not get any label from glucose.

Reviewer #3 (Remarks to the Author):

congratulations

Thank you!